# Membrane-associated effluxosomes coordinate multi-metal resistance in *Mycobacterium tuberculosis*

Pierre Dupuy [iD][1,✉], Yves-Marie Boudehen [iD][1,7], Marion Faucher[1,8], John A Buglino[2], Allison Fay [iD][2], Sylvain Cantaloube[3], Yasmina Grimoire [iD][1], Julien Marcoux[1,4], Florian Levet[5,6], Laetitia Bettarel[5], Bertille Voisin[1], Jérôme Rech[3], Jean-Yves Bouet [iD][3], Olivier Saurel[1], Jean-Baptiste Sibarita [iD][5], Michael Glickman[2], Claude Gutierrez [iD][1] & Olivier Neyrolles [iD][1,✉]

## Abstract

Bacterial pathogens must withstand metal-induced stress during infection, yet the mechanisms by which they sense and respond to toxic metal ions remain incompletely understood. Here, we uncover a previously unrecognized mechanism in *Mycobacterium tuberculosis*, the causative agent of tuberculosis, which assembles dynamic, membrane-associated platforms organized by PacL proteins to mediate resistance to multiple metals. The small membrane-associated proteins PacL1, PacL2, and PacL3 coordinate the clustering of P-type ATPase pumps, namely CtpC, CtpG, and CtpV, into functional complexes that we term *effluxosomes*. Using single-particle tracking, we reveal distinct dynamic populations, with highly mobile PacL proteins integrating into more slowly mobile effluxosomes. PacL proteins stabilize CtpC and CtpG within these assemblies, promoting cross-resistance to zinc and cadmium, with PacL1 acting as a multi-substrate metallochaperone that binds zinc, cadmium, and copper *via* a conserved C-terminal motif. Single-molecule-based super-resolution microscopy shows that conserved residues within the PacL transmembrane domain are essential for effluxosome assembly. Strikingly, proximity labeling reveals a broad PacL1 interaction network, suggesting that effluxosomes contribute to a wider stress adaptation program. These findings establish effluxosomes as dynamic membrane machineries that orchestrate coordinated multi-metal resistance in *M. tuberculosis*, opening new avenues for antimicrobial targeting.

**Keywords** Cadmium Resistance; Effluxosomes; Metal Homeostasis; *Mycobacterium tuberculosis*; PacL Proteins
**Subject Categories** Membranes & Trafficking; Microbiology, Virology & Host Pathogen Interaction

## Introduction

While transition metals are essential for numerous biochemical processes, acting as cofactors for enzymes and as structural components of proteins, an imbalance in these metals can be toxic. Indeed, bacteria must tightly regulate metal ion homeostasis to maintain cellular function and adapt to fluctuating environmental conditions (Capdevila et al, 2024). In particular, local concentrations and the availability of metal ions can shift during infection. Our research, along with that of others, has shown that immune cells exploit transition metals to intoxicate bacterial pathogens, and that metal efflux systems contribute to bacterial virulence (White et al, 2009; Botella et al, 2011; Neyrolles et al, 2013, 2015). How pathogens mount a coordinated response to such stress, remains unclear.

Specifically, we demonstrated that macrophage phagosomes containing *Mycobacterium tuberculosis*, the causative agent of tuberculosis, accumulate toxic levels of zinc (Botella et al, 2011). Our analysis also identified an unprecedented mycobacterial zinc detoxification system composed of a membrane P-ATPase metal efflux pump, CtpC (Botella et al, 2011), and a previously uncharacterized protein, PacL1 (Boudehen et al, 2022). CtpC plays a critical role in *M. tuberculosis* zinc tolerance in culture and in bacterial replication within host cells and tissues (Botella et al, 2011; Padilla-Benavides et al, 2013; Hanna et al, 2021; Sassetti and Rubin, 2003). PacL1, a small membrane-associated protein containing a domain of unknown function DUF1490, is indispensable for CtpC-dependent zinc tolerance (Boudehen et al, 2022). PacL1 directly interacts with CtpC, and both proteins colocalize in dynamic membrane patches. Acting as a chaperone, PacL1 is critical for stabilizing CtpC and has metal-binding activity. This activity is mediated by a C-terminal motif, which is essential for zinc tolerance under high-zinc conditions.

Intriguingly, *M. tuberculosis* also possesses two homologs of PacL1, namely PacL2 and PacL3 (Boudehen et al, 2022), whose biological functions remain unknown. All PacL proteins share common structural features, including an N-terminal transmembrane (TM) α-

[1]Institut de Pharmacologie et de Biologie Structurale, IPBS, Université de Toulouse, CNRS, UPS, Toulouse, France. [2]Immunology Program, Sloan Kettering Institute, New-York, NY 10065, USA. [3]Centre de Biologie Intégrative (CBI), Université de Toulouse, CNRS, UPS, 31000 Toulouse, France. [4]Infrastructure Nationale de Protéomique, ProFI, UAR 2048, Toulouse, France. [5]Interdisciplinary Institute for Neuroscience, University of Bordeaux, CNRS, Bordeaux, France. [6]Univ. Bordeaux, CNRS, INSERM, Bordeaux Imaging Center, BIC, UAR3420, US 4, F-33000 Bordeaux, France. [7]Present address: Institut Pasteur, CNRS UMR 3528, Université Paris Cité, Bacterial Cell Cycle Mechanisms Unit, F-75015 Paris, France. [8]Present address: G.CLIPS BIOTECH, Labège, France. ✉E-mail: Pierre.Dupuy@ipbs.fr; Olivier.Neyrolles@ipbs.fr

helix and a cytoplasmic domain containing Ala/Glu (AE) repeats. PacL2 and PacL3 are encoded within an operon alongside two other P-ATPases, CtpG and CtpV, as well as two transcriptional regulators, CmtR and CsoR. These regulators induce operon expression in response to cadmium (Chauhan et al, 2009; Wang et al, 2005) and copper (Liu et al, 2007), respectively. However, conflicting studies suggest that CtpG contributes to tolerance against zinc (Chen et al, 2022), cadmium (López et al, 2018), and copper (López et al, 2018), while CtpV appears to be primarily associated with copper tolerance (Ward et al, 2010).

Notably, both CtpG and CtpV, like CtpC, are induced during macrophage infection (Tailleux et al, 2008) and contribute to bacterial growth within the host (Ward et al, 2010; Chen et al, 2022). Furthermore, our previous research demonstrated that when PacL1, PacL2, and PacL3 are co-expressed as an artificial operon under a constitutive promoter in the *M. tuberculosis* non-pathogenic relative *Mycobacterium smegmatis*, they colocalize within membrane clusters (Boudehen et al, 2022). Overall, previous work hints at close functional collaboration between PacL orthologs and a potential relationship to various P-ATPase pumps in *M. tuberculosis*.

Here, we reveal a previously unrecognized interplay between PacL proteins and diverse P-ATPase pumps in *M. tuberculosis*, uncovering the existence of atypical multi-metal efflux machineries. By dissecting the functional roles of PacL2 and CtpG, we establish their specific involvement in cadmium tolerance and demonstrate that PacL2 is essential for both the stability and membrane localization of CtpG. Furthermore, our findings highlight a broader interaction network in which PacL1, PacL2, and PacL3 physically associate with each other, as well as with CtpC, CtpG, and CtpV, collectively contributing to multi-metal resistance. Using super-resolution photoactivated localization microscopy (PALM), we characterize these multiprotein membrane clusters and identify conserved residues critical for their assembly. Through single-protein tracking, we further reveal distinct populations with diverse dynamic behavior. Finally, using a proximity labeling approach, we map the PacL1 interaction network, uncovering the potential involvement of dozens of additional proteins within these clusters. Given the importance of these membrane clusters in coordinating the homeostasis of multiple metal ions, we name them effluxosomes. These findings not only deepen our understanding of metal homeostasis in *M. tuberculosis* but also suggest that similar dynamic membrane multiprotein machineries, involving PacL-Ctp pairs found across a variety of bacterial species and genera, may represent a conserved strategy for adapting to multi-metal stress, offering new perspectives on microbial resilience and potential targets for antimicrobial intervention.

## Results

### PacL/Ctp systems mediate dual tolerance to zinc and cadmium

To investigate the role of the three PacL/Ctp systems in *M. tuberculosis* metal tolerance, we expressed the *pacL1-ctpC*, *cmtR-pacL2-ctpG*, and *csoR-pacL3-ctpV* operons (Fig. 1A) driven by their native promoters in the non-pathogenic *M. tuberculosis* relative species, *M. smegmatis*. Using a disc diffusion assay, we observed that the expression of *pacL1-ctpC* and *pacL2-ctpG* conferred

increased tolerance to zinc (Fig. EV1A) and cadmium (Fig. EV1B), respectively, but not to other tested metals, including copper (Fig. EV1C), nickel (Fig. EV1D), manganese (Fig. EV1E), or iron (Fig. EV1F). Mutation of the conserved APC motif in the P-ATPase domain of CtpG (APC > AAA), which is predicted to be essential for metal binding and transport (Neyrolles et al, 2013), abolished the enhanced cadmium tolerance conferred by the expression of the *ctpG* operon (Fig. EV1B). Therefore, CtpG promotes cadmium resistance in an APC motif-dependent manner.

In contrast, the expression of the *ctpV* operon did not affect tolerance to any of the tested metals (Fig. EV1A–F). As this result contradicts a previous study suggesting that CtpV contributes to copper tolerance (Ward et al, 2010), we additionally expressed the *csoR-pacL3-ctpV* operon together with the downstream *Rv0970* gene (Fig. 1A) to assess whether this gene plays a role in CtpV function. Even in this strain, which expresses the complete operon, we saw no effect on copper tolerance in *M. smegmatis* when tested with either $Cu^{2+}$ (Fig. EV1G) or $Cu^{+}$ (Fig. EV1H).

Previous studies have shown that zinc (Botella et al, 2011), cadmium (Chauhan et al, 2009), and copper (Ward et al, 2010) induce the expression of *ctpC*, *ctpG*, and *ctpV*, respectively. To confirm that the observed metal specificity of the PacL1/CtpC and PacL2/CtpG systems arises from their intrinsic enzymatic activities rather than differential expression triggered by specific stress conditions, we expressed each PacL/Ctp pair under an anhydrotetracycline (ATC)-inducible (*tet*) promoter (Ehrt et al, 2005). In the absence of the inducer, no differences in metal tolerance were detected between strains (Fig. EV1I). ATC-induced expression of *pacL3-ctpV* did not confer increased tolerance to any tested metal, whereas *pacL2-ctpG* expression specifically enhanced cadmium tolerance (Figs. 1B–D and EV1J). Interestingly, ATC-induced expression of *pacL1-ctpC* conferred increased tolerance to both zinc and cadmium (Figs. 1B, C and EV1J), suggesting that, when sufficiently expressed, this system can facilitate the efflux of both metals.

To validate the role of PacL1/CtpC and PacL2/CtpG systems in cadmium tolerance in *M. tuberculosis*, we generated deletion mutants for each system individually and in combination (ΔpacL1ctpC, ΔpacL2ctpG, and ΔpacL1ctpCΔpacL2ctpG). Deletion of *pacL2* and *ctpG* resulted in increased sensitivity to cadmium (Fig. 1E), whereas ectopic expression of these genes in the mutant background fully restored cadmium tolerance (Fig. 1F). Notably, the ΔpacL1ctpC-pacL2ctpG mutant exhibited even greater sensitivity to cadmium than the ΔpacL2ctpG mutant (Fig. 1G), suggesting that the PacL1/CtpC system serves as a backup mechanism for PacL2/CtpG in mediating cadmium tolerance. Given that *ctpC* expression is induced by zinc in *M. tuberculosis*, we further investigated whether zinc supplementation could influence cadmium tolerance in the ΔpacL2ctpG mutant background. Interestingly, zinc enhanced cadmium tolerance in the mutant (Fig. 1H), revealing a cross-resistance mechanism that connects zinc to cadmium tolerance in this pathogen.

### PacL2 is essential for stabilizing CtpG within clusters

To elucidate the role of PacL2 in CtpG-dependent cadmium tolerance in *M. tuberculosis*, we expressed *ctpG* in *M. smegmatis*, either alone or in combination with *pacL2*. While co-expression of *pacL2* and *ctpG* conferred significant cadmium tolerance, expression of *pacL2* alone had no effect and *CtpG* expression alone

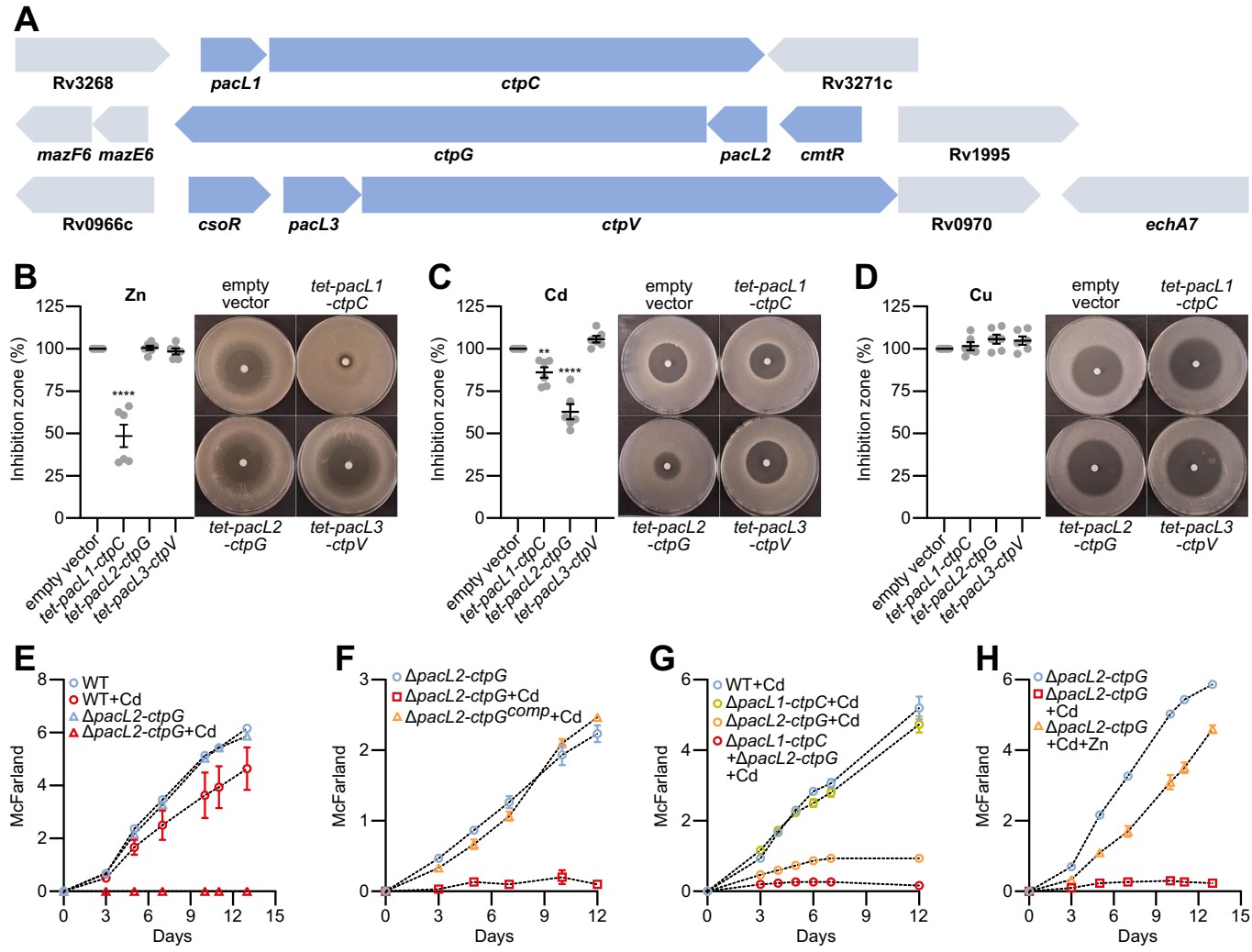

**Figure 1. CtpC and CtpG collaborate to confer cadmium tolerance in *M. tuberculosis*.**

(A) Genomic organization of *pacL*, *ctp*, and adjacent genes on the *M. tuberculosis* chromosome. *pacL1-ctpC*, *cmtR-pacL2-ctpG*, and *csoR-pacL3-ctpV* form operons in which *cmtR* and *csoR* encode the transcriptional regulators controlling their downstream genes. (B–D) Metal sensitivity of *M. smegmatis* strains harboring a genome-integrative vector expressing the indicated *M. tuberculosis pacL-ctp* genes under the control of an anhydrotetracycline (ATC)-inducible promoter (*tet*), assessed by disk diffusion assay upon exposure to zinc (ZnSO₄), cadmium (CdSO₄), or copper (CuSO₄) in the presence of the inducer. Inhibition zone diameters normalized to the empty vector control are shown as mean ± SEM from biological replicates, with individual values indicated by gray dots. Asterisks above the means indicate statistically significant differences compared to the reference strain (**$P < 0.01$; ****$P < 0.0001$), calculated by one-way analysis of variance (ANOVA) followed by a Dunnett post-test. Exact $p$ values and biological replicate numbers are reported in Table EV6. (E–H) Bacterial growth of the indicated *M. tuberculosis* strains in the absence or presence of the specified metals (20 μM CdSO₄ and 50 μM ZnSO₄), evaluated by turbidity measurements (McFarland units). Data represent mean ± SEM of three independent biological replicates. Δ*pacL2-ctpG*comp: ectopic expression of the *cmtR-pacL2-ctpG* operon under the control of its native promoter in the Δ*pacL2-ctpG* mutant background. Source data are available online for this figure.

provided only weak tolerance (Fig. 2A). To further validate the role of PacL2 in *M. tuberculosis*, we generated an in-frame deletion mutant of *pacL2*. Deletion of *pacL2* led to increased sensitivity to cadmium (Fig. 2B), which was fully rescued by ectopic expression of *pacL2* in the mutant background (Fig. 2C), confirming its essential role in cadmium tolerance.

To determine the subcellular localization of PacL2 and CtpG, we engineered C-terminal translational fusions with mTurquoise (PacL2mT) and mVenus (CtpGmV). Fluorescent fusions were introduced within the native operon (*cmtR-pacL2*mT*-ctpG*, *cmtR-pacL2-ctpG*mV, or *cmtR-pacL2*mT*-ctpG*mV) and cloned into a single-

copy integrative vector, thereby preserving the operon structure and ensuring expression of the tagged proteins under the control of the endogenous promoter. Using epifluorescence microscopy, we observed that PacL2 localized in distinct clusters, which became more intense and abundant upon cadmium exposure (Fig. 2D,E). Under the same conditions, CtpG also formed clusters that colocalized with PacL2 (Fig. 2F). While C-terminal tagging of PacL2 markedly reduced its functionality, though a weak cadmium tolerance was still detectable (Fig. EV1K), the CtpG-mVenus fusion remained fully functional (Fig. EV1L), indicating that the observed clusters represent active efflux machineries. Interestingly, in the

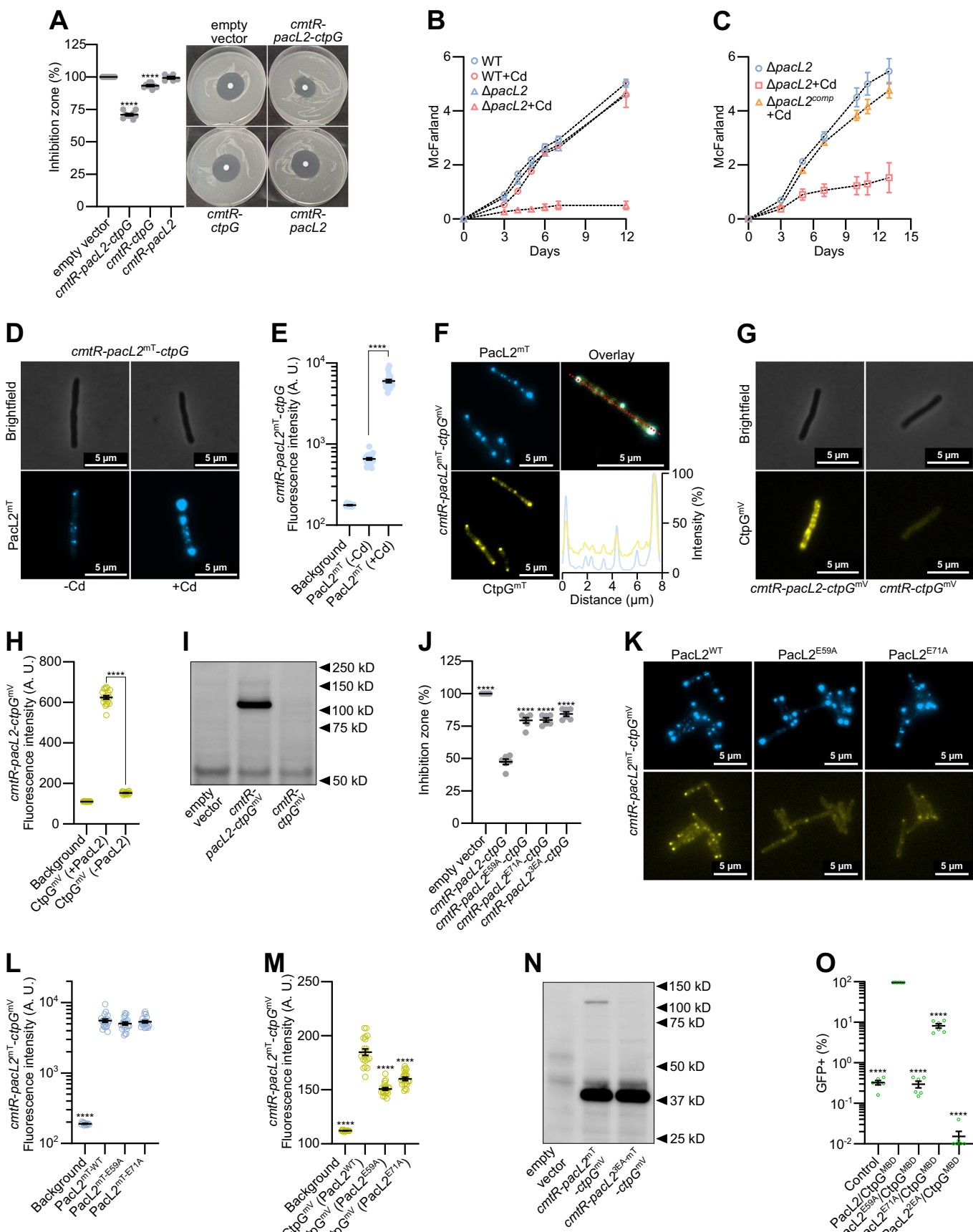

◀ **Figure 2. PacL2 is essential for stabilizing CtpG within clusters.**

(A) Cadmium (CdSO$_4$) sensitivity of *M. smegmatis* strains carrying a genome-integrative vector expressing the indicated *M. tuberculosis* genes under the control of their native promoter, assessed by disk diffusion assay. Inhibition zone diameters normalized to the empty vector control are shown as mean ± SEM from biological replicates, with individual values indicated by gray dots. (B, C) Bacterial growth of the indicated *M. tuberculosis* strains in the absence or presence of 20 μM CdSO$_4$, evaluated by turbidity measurements (McFarland units). Data represent mean ± SEM from three independent biological replicates. (D) Epifluorescence microscopy images of live *M. smegmatis* strains carrying a genome-integrative vector expressing the indicated *M. tuberculosis* genes under the control of their native promoter, cultivated without or with 100 nM CdSO$_4$. (E) Mean fluorescence intensity (gray values) quantified in ImageJ from 8-bit grayscale images of individual cells from the same genetic background and growth condition as in (D). Data were shown as mean ± SEM from individual cells represented by blue dots. (F) Epifluorescence microscopy images of fixed *M. smegmatis* strains carrying a genome-integrative vector expressing the indicated *M. tuberculosis* genes under the control of their native promoter, cultivated with 100 nM CdSO$_4$. The bottom-right panel shows colocalization of mTurquoise and mVenus fluorescence signals along the red line in the overlay image. (G) Epifluorescence microscopy images of live *M. smegmatis* strains carrying a genome-integrative vector expressing the indicated *M. tuberculosis* genes under the control of their native promoter, cultivated with 100 nM CdSO$_4$. (H) Mean fluorescence intensity (gray values) quantified in ImageJ from 8-bit grayscale images of individual cells from the same genetic background and growth condition as in (G). Data were shown as mean ± SEM from individual cells represented by circles. (I) Western blot analysis of membrane extracts from *M. smegmatis* strains carrying a genome-integrative vector expressing the indicated *M. tuberculosis* genes under the control of their native promoter, cultivated with 100 nM CdSO$_4$. Detection was performed using an anti-GFP antibody (CtpG$^{mV}$: 106.5 kDa). (J) Cadmium (CdSO$_4$) sensitivity of *M. smegmatis* strains carrying a genome-integrative vector expressing the indicated *M. tuberculosis* genes under the control of their native promoter, assessed by disk diffusion assay. Inhibition zone diameters normalized to the empty vector control are shown as mean ± SEM from biological replicates, with individual values indicated by gray dots. (K) Epifluorescence microscopy images of live *M. smegmatis* strains carrying a genome-integrative vector expressing the indicated *M. tuberculosis* genes under the control of their native promoter, cultivated with 100 nM CdSO$_4$. (L, M) Mean fluorescence intensity (gray values) quantified in ImageJ from 8-bit grayscale images of individual cells from the same genetic background and growth condition as in (K). Data were shown as mean ± SEM from individual cells represented by circles. (N) Western blot analysis of membrane extracts from *M. smegmatis* strains carrying a genome-integrative vector expressing the indicated *M. tuberculosis* genes under the control of their native promoter, cultivated with 100 nM CdSO$_4$. Detection was performed using an anti-GFP antibody (PacL2$^{mT}$: 36.5 kDa; CtpG$^{mV}$: 106.5 kDa). (O) Bipartite split-GFP assay in *M. smegmatis* expressing a PacL1-GFP11 fusion protein together with GFP1-10 (control) or the indicated proteins fused to GFP11 (first protein) or GFP1-10 (second protein). Fluorescence was recorded by flow cytometry and expressed as the percentage of GFP-positive cells. Data are shown as mean ± SEM from biological replicates, each represented by a green dot. For statistical analysis, (A, J, L, M, O) one-way ANOVA with a Dunnett post-test or (E, H) an unpaired *t*-test were performed. Statistical analyses were conducted on ln-transformed data for (E, L, O). Asterisks indicate statistically significant differences compared to the reference condition (empty vector (A, J), PacL2$^{mT-WT}$ (L), CtpG$^{mV}$ (PacL2$^{WT}$) (M), or PacL2/CtpG$^{MBD}$ (O)) or between conditions linked by a black line (E, H) (****$P < 0.0001$). Exact *p* values and biological replicate numbers presented in this figure are reported in Table EV6. Δ*pacL2*$^{comp}$: ectopic expression of the *cmtR-pacL2* operon under the control of its native promoter in the Δ*pacL2* mutant background.-Cd: absence of cadmium (CdSO$_4$) in the bacterial culture medium; +Cd: presence of cadmium (CdSO$_4$) in the bacterial culture medium; PacL2$^{mT}$: PacL2-mTurquoise fusion; CtpG$^{mV}$: CtpG-mVenus fusion; CtpG$^{MBD}$: N-terminal domain of CtpG encoding the predicted metal-binding domain of the P-ATPase; 2EA: E59A and E71A substitutions in PacL2; 3EA: E55A, E59A, and E71A substitutions in PacL2. Source data are available online for this figure.

absence of PacL2, CtpG clusters were undetectable (Fig. 2G,H), indicating that PacL2 is essential for stabilizing CtpG and facilitating its proper localization. Consistent with this observation, western blot analyses detected CtpG protein in bacterial lysates co-expressing PacL2, but not in those expressing CtpG alone (Fig. 2I).

Our previous study demonstrated that AE repeats located in the cytoplasmic domain of PacL1 are essential for both the PacL1/CtpC interaction and the stabilization of CtpC. Given that similar AE repeats are also present in the PacL2 sequence (Fig. EV2), we investigated their role in CtpG stability. Substitutions E59A or E71A, or the triple substitution E55A, E59A, and E71A abolished the resistance to cadmium when the mutated operon was expressed in *M. smegmatis* (Fig. 2J). In addition, although the E59A or E71A mutants did not prevent the formation of PacL2-mTurquoise clusters, they resulted in the destabilization of CtpG-mVenus (Fig. 2K–M) and the triple substitution E55A, E59A, and E71A completely abolished the presence of CtpG in bacterial lysates, as determined by western blot analysis (Fig. 2N). Finally, bipartite split-GFP experiments demonstrated that PacL2 interacts with the predicted metal-binding domain (MBD) of CtpG (Fig. 2O). Furthermore, the above E to A substitutions in PacL2 reduced this interaction (Fig. 2O). Altogether, these data demonstrate that the acidic residues present in conserved AE motifs on the PacL2 cytoplasmic domain are essential for its protein chaperone activity, facilitating the recruitment of CtpG into clusters and protecting it from degradation.

## PacL2 and CtpG form dynamic membrane-associated clusters

Analysis of the dynamics of the PacL2-CtpG clusters using epifluorescence video-microscopy revealed two distinct types of structures: large, immobile clusters frequently localized at the cell poles, and smaller, mobile clusters that moved dynamically throughout the cell (Movie EV1). The smaller clusters appeared to preferentially localize to specific regions where they interact with one another. To further characterize the nanoscale organization and dynamics of PacL2 and CtpG, we generated PacL2-mEos and CtpG-mEos fusion proteins for super-resolution microscopy. These constructs were based on the same integrative vectors used for the mTurquoise/mVenus fusions, except that the fluorescent proteins were replaced with mEos. Using 3D photoactivated localization microscopy (PALM) (Betzig et al, 2006), we constructed the tridimensional nanoscale organization of PacL2 and CtpG within individual *M. smegmatis* cells and found that both proteins form membrane-associated clusters (Fig. 3A,B and Movies EV2–5). Using the point cloud analyst software PoCA (Levet and Sibarita, 2023), we quantified the number of clusters (Fig. 3C), their volumes (Fig. 3D), and the proportion of PacL2 molecules localized within clusters (Fig. 3E). Our analysis revealed that ~60% of cellular PacL2 was organized into an average of 12.5 membrane clusters per cell (Fig. 3C,E), with cluster sizes ranging from 10$^5$ nm$^3$ to 10$^8$ nm$^3$ (Fig. 3D). CtpG formed nearly as many clusters per cell as PacL2 (Fig. 3C). However, the CtpG clusters were smaller (Fig. 3D) and less dense, with fewer than 20% of CtpG molecules localized within clusters (Fig. 3E).

We then monitored the mobility of PacL protein using single-protein tracking combined with PALM (sptPALM) (Manley et al, 2008). Analyzing the localization and dynamics of single PacL molecules over 2.5 min at 33 frames per second, we reconstructed the nanoscale organization of PacL in nanoclusters of different sizes distributed along the bacteria, as well as its dynamic behavior

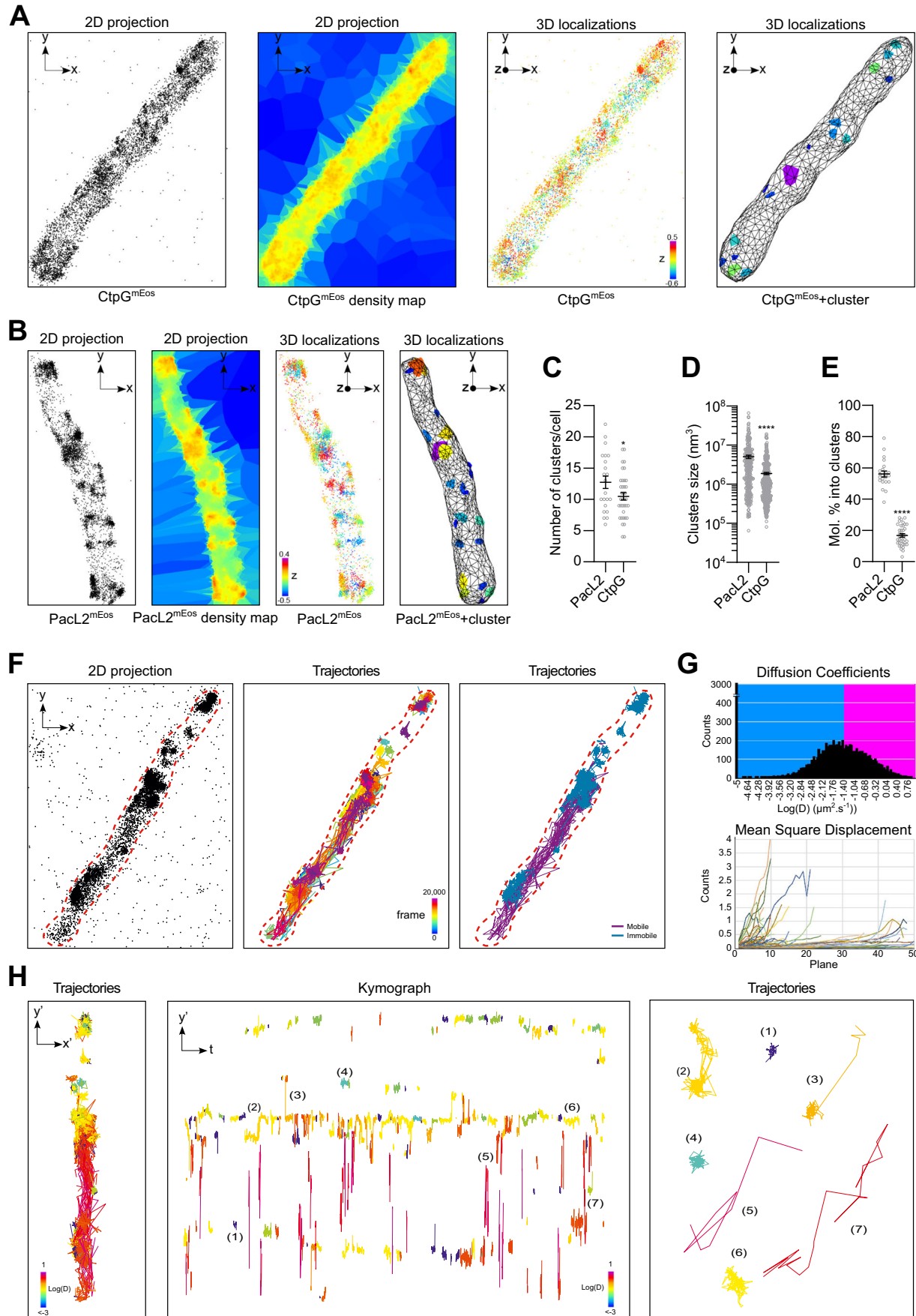

**Figure 3.   PacL2 clustering and dynamics.**

(A, B) Representative 3D super-resolution reconstructions of PacL2-mEos or CtpG-mEos in *M. smegmatis* cells using 3D PALM. From left to right: localizations, density map, 3D localizations, and cluster segmentation using the PoCA software. Quantification of (C) the number of PacL2 or CtpG clusters per cell, (D) cluster sizes, and (E) proportion of clustered molecules per cell. Results are presented as means ( ± SEM) from individual cells (C, E) or clusters (D), represented by gray dots. (F) Representative super-resolution reconstructions of PacL2-mEos protein acquired by sptPALM. From left to right: localizations, trajectories with lengths > 6 time points, overlay between slow (cyan) and fast (magenta) molecules, threshold = 0.03 $\mu m^2.s^{-1}$. (G) Top: Histogram of diffusion coefficients computed from 19 cells. Bottom: mean square displacement of the molecules computed from the trajectories of the cell in (F). (H) Trajectories and kymograph analysis along the long axis of the cell, color-coded with their diffusion coefficients, examples of trajectories showing immobilized (1 and 4), freely diffusive (2, 3, and 6), and highly mobile molecules (5 and 7). Asterisks above the means indicate statistically significant differences compared to the reference condition (PacL2) (*$P < 0.05$; ****$P < 0.0001$), calculated using an unpaired *t*-test. Exact *p* values and replicate numbers presented in this figure are reported in Table EV6. PacL2$^{mEos}$: PacL2-mEos fusion; CtpG$^{mEos}$: CtpG-mEos fusion. Source data are available online for this figure.

(Fig. 3F–H and Movie EV6). Diffusion analysis computed based on the mean square displacements (MSD) of individual single-molecule trajectories (9510 trajectories longer than six time points extracted from 19 isolated cells) revealed a broad heterogeneity of mobility with diffusion coefficient spanning more than three orders of magnitude and a large variety of MSD curves, without a clear cut-off between populations (Fig. 3G). To discriminate mobile from immobile molecules, we applied an empirical threshold of 0.03 $\mu m^2 \cdot s^{-1}$, consistent with diffusion regimes previously reported for membrane proteins (Kusumi et al, 1993; Saxton and Jacobson, 1997; Rossier et al, 2012), where diffusion coefficients above ~0.02–0.05 $\mu m^2 \cdot s^{-1}$ typically correspond to freely diffusive species, and lower values to confined or immobilized states. Using this criterion, our analysis revealed that 70% of the molecules were immobile, while 30% were classified as mobile. These two populations were spatially mapped within the bacteria, clearly distinguishing confined molecules within clusters from mobile molecules moving between them (Fig. 3G). Kymograph analysis further revealed the spatio-temporal distribution of individual PacL molecules along the bacterial long axis (Fig. 3H), identifying three distinct behaviors: (i) confined molecules forming large, stable clusters visible throughout the acquisition; (ii) transiently confined molecules, likely representing small, slowly mobile clusters, characterized by brief immobilizations; and (iii) highly mobile molecules traveling along the bacteria, depicted as long red traces. Only rare events of state transitions from immobile to mobile were observed. Together, these results highlight a heterogeneous and dynamic nanoscale organization of PacL, comprising both stable clusters and highly mobile molecules within the bacterial membrane.

## PacL proteins function as hubs that organize multiple P-ATPase pumps into shared membrane-associated clusters

Our previous study demonstrated that PacL1 colocalizes with PacL2 and PacL3 within shared membrane clusters when these genes are artificially co-expressed as an operon under the control of a constitutive promoter in *M. smegmatis* (Boudehen et al, 2022). This observation raises the question of whether these proteins physically interact. To investigate this, we performed bipartite split-GFP experiments and found that PacL1 interacts not only with itself but also with both PacL2 and PacL3, with PacL2 exhibiting a similar interaction pattern (Fig. 4A). As a control for potential nonspecific membrane interactions, we tested the association between PacL1 and the flotillin-like protein Rv1488, which forms

membrane clusters independent of PacL-dependent platforms (Boudehen et al, 2022). As expected, this membrane-associated control has a higher likelihood of random encounters with PacL proteins, resulting in a stronger fluorescence signal than that observed with the cytoplasmic GFP1-10 negative control. However, we observed a four-fold lower GFP signal in cells expressing the Rv1488/PacL1 pair compared to those expressing two PacL proteins, indicating that interactions between PacL proteins are specific rather than the result of random membrane proximity.

To determine whether PacL/PacL colocalization occurs in *M. tuberculosis* under more physiological expression conditions, we independently expressed PacL1-mVenus and PacL2-mTurquoise fusion proteins under the control of their respective native promoters, either in the absence of metals or in the presence of zinc, cadmium, or both. PacL1 clusters were consistently detected in *M. tuberculosis* membranes under all four conditions (Fig. 4B,C), with an increase in fluorescence intensity upon metal exposure, particularly in response to zinc, which induced the highest signal (Fig. 4C). In contrast, in the absence of metals, PacL2-mTurquoise fluorescence was very weak and displayed a homogeneous distribution within *M. tuberculosis* cells (Fig. 4B,D). Notably, exposure to both zinc and cadmium triggered the formation of membrane-associated PacL2 clusters, with cadmium eliciting the strongest increase in fluorescence signal (Fig. 4B,D). Under metal stress conditions, we found that PacL1 and PacL2 colocalized to the *M. tuberculosis* membrane (Fig. 4B). These results suggest that environmental conditions modulate the composition of PacL-dependent platforms, with zinc preferentially promoting PacL1-enriched clusters and cadmium favoring the formation of PacL2-enriched clusters.

Next, we explored whether the clustering of PacL1 and PacL2 also facilitates the co-clustering of CtpC and CtpG. To test this hypothesis, we expressed PacL1 and CtpC fused to mTurquoise, alongside PacL2 and CtpG fused to mVenus in *M. smegmatis* under the control of their respective native promoters. Remarkably, we observed that both P-ATPase pumps colocalized within shared clusters (Fig. 4E), revealing the presence of previously unrecognized multi-metal efflux machineries in mycobacteria.

## Redundant chaperone functions of PacL proteins in stabilizing P-ATPase pumps

Our previous study revealed that PacL1 physically interacts with the MBD of CtpC (Boudehen et al, 2022). Here, we used the bipartite split-GFP assay to investigate potential cross-interactions between PacL proteins and the MBDs of CtpC, CtpG, and CtpV.

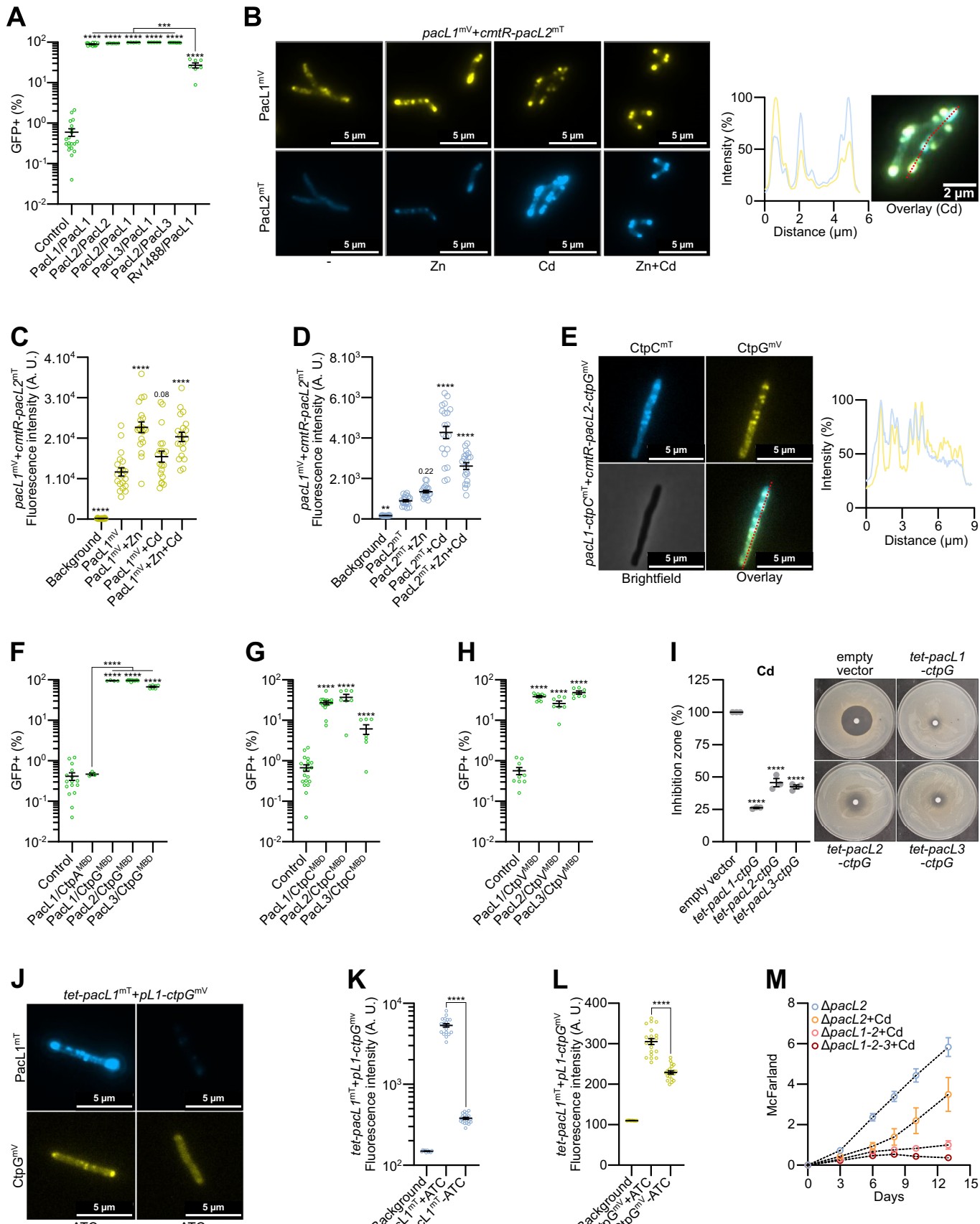

**Figure 4. PacL proteins collaborate to assemble multi-metal efflux machineries.**

(A) Bipartite split-GFP assay in *M. smegmatis* expressing a PacL1-GFP11 fusion protein along with GFP1-10 (control) or the indicated proteins fused to GFP11 (first protein) or GFP1-10 (second protein). Fluorescence was recorded by flow cytometry and expressed as the percentage of GFP-positive cells. Data were presented as mean ± SEM from biological replicates, represented by green dots. (B) Epifluorescence microscopy images of fixed *M. tuberculosis* strain carrying a genome-integrative vector expressing the indicated *M. tuberculosis* genes under the control of their native promoter, cultivated in the absence or presence of the indicated metals (100 μM $ZnSO_4$ and/or 10 μM $CdSO_4$). The right panel shows colocalization of mTurquoise and mVenus fluorescent signals along the red line in the overlay image. (C, D) Mean fluorescence intensity (gray values) quantified in ImageJ from 8-bit grayscale images of individual cells from the same genetic background and growth conditions as in (B). Results are presented as mean ± SEM from individual cells, represented by yellow or blue dots. (E) Epifluorescence microscopy images of fixed *M. smegmatis* strains carrying a genome-integrative vector expressing the indicated *M. tuberculosis* genes under the control of their native promoter, cultivated with 100 nM $CdSO_4$. The right panel shows colocalization of mTurquoise and mVenus fluorescent signals along the red line in the overlay image. (F–H) Bipartite split-GFP assay in *M. smegmatis* expressing a PacL1-GFP11 fusion protein along with GFP1-10 (control) or the indicated proteins fused to GFP11 (first protein) or GFP1-10 (second protein). Fluorescence was recorded by flow cytometry and expressed as the percentage of GFP-positive cells. Data were presented as mean ± SEM from biological replicates, represented by green dots. (I) Cadmium ($CdSO_4$) sensitivity of *M. smegmatis* strains harboring a genome-integrative vector expressing the indicated *M. tuberculosis* pacL-ctp genes under the control of an ATC-inducible promoter, assessed by disk diffusion assay in the presence of the inducer. Inhibition zone diameters normalized to the empty vector control (tet-empty) are shown as mean ± SEM from biological replicates, with individual values indicated by gray dots. (J) Epifluorescence microscopy images of live *M. smegmatis* strains carrying a genome-integrative vector expressing the indicated *M. tuberculosis* genes under the control of the indicated promoters, cultivated without or with 50 nM anhydrotetracycline (ATC). (K, L) Mean fluorescence intensity (gray values) quantified in ImageJ from 8-bit grayscale images of individual cells from the same genetic background and growth conditions as in (J). Results are presented as mean ± SEM from individual cells, represented by yellow or blue dots. (M) Bacterial growth of the indicated *M. tuberculosis* strains in the absence or presence of 10 μM $CdSO_4$, evaluated by turbidity measurements (McFarland units). Data represent mean ± SEM from biological triplicates. For statistical analysis, (A, C, D, F–I) one-way ANOVA with a Dunnett post-test or (K, L) an unpaired *t*-test were performed. Statistical analyses were conducted on ln-transformed data for (A, C, D, F–H. Asterisks indicate statistically significant differences compared to the reference condition (control (A, F, G, H), PacL1$^{mV}$ (C), PacL2$^{mT}$ (D), empty vector (I)), or between conditions linked by a black line (F, K, L) (**$P < 0.01$; ****$P < 0.0001$). Exact *p* values and biological replicate numbers presented in this figure are reported in Table EV6. PacL2$^{mT}$: PacL2-mTurquoise fusion; CtpG$^{mV}$: CtpG-mVenus fusion; tet-: ATC-inducible promoter; pL1: *pacL1* promoter; CtpA$^{MBD}$, CtpC$^{MBD}$, CtpG$^{MBD}$, and CtpV$^{MBD}$: N-terminal domain of CtpA, CtpC, CtpG, and CtpV encoding the predicted P-ATPase metal-binding domain. Source data are available online for this figure.

Interestingly, we found that the MBDs of CtpG and CtpV interact with PacL1, PacL2, and PacL3, whereas the MBD of CtpC interacts with PacL1 and PacL2, but less efficiently with PacL3 (Fig. 4F–H). In contrast, PacL1 did not interact with the MBD of CtpA (Fig. 4F), demonstrating that not all *M. tuberculosis* P-ATPases bind to PacL1. To assess whether PacL1 or PacL3 can substitute for PacL2 in stabilizing CtpG, we generated *M. smegmatis* strains expressing CtpG in combination with PacL1, PacL2, or PacL3 under the control of a *tet* promoter. Disc diffusion assays revealed that all three PacL proteins conferred CtpG-dependent cadmium tolerance, with PacL1 providing the highest level of resistance (Fig. 4I). To confirm the role of PacL1 in stabilizing CtpG within membrane clusters, we co-expressed a CtpG-mVenus fusion protein under a constitutive promoter along with a PacL1-mTurquoise fusion protein under the control of a *tet* promoter. In the presence of the inducer, PacL1 and CtpG formed fluorescent foci that colocalized in the membrane of *M. smegmatis* (Fig. 4J). In the absence of induction, the fluorescence intensity of both PacL1 and CtpG decreased significantly, indicating that PacL1 contributes to CtpG stability (Fig. 4J–L). Finally, to investigate the contribution of PacL proteins to cadmium tolerance in *M. tuberculosis*, we constructed multiple *pacL* knockout mutants. The Δ*pacL1*Δ*pacL2* double mutant exhibited greater sensitivity to cadmium than the Δ*pacL1* single mutant (Fig. 4M), demonstrating that at least two PacL proteins function cooperatively to confer cadmium tolerance in this pathogen.

## PacL orthologs have distinct metallochaperone activities

PacL1 contains a C-terminal metal-binding motif, $D^{87}LHDHDH^{93}$ (Fig. EV2) that binds zinc at a 1:1 molar ratio and is crucial for resistance to high-zinc concentrations in *M. tuberculosis*, supporting its role as a metallochaperone (Boudehen et al, 2022). The clustering of multiple P-ATPase pumps within shared membrane domains raised the question of whether the metallochaperone function of PacL1 extends beyond zinc binding. To test this hypothesis, we employed native mass spectrometry to assess the relative binding affinity of the purified cytoplasmic domain of PacL1 for various metals. Interestingly, PacL1 was capable of binding cadmium (Fig. 5A,C), zinc (Fig. 5D), and copper (Fig. 5E) but not manganese (Fig. 5F). Notably, while PacL1 predominantly bound a single zinc or cadmium atom (Fig. 5C,D), species containing one, two, or three copper atoms were detected (Fig. 5E). In order to assess the metal-binding affinity of PacL1 and account for potential nonspecific binding or gas-phase dissociation, competition assays were performed in ammonium acetate supplemented with either Cd/Zn/Cu/Mn or Cd/Zn/Mn. We observed that PacL1 bound exclusively to copper when incubated with Cd/Zn/Cu/Mn (Fig. EV3A) and exclusively to zinc when incubated with Cd/Zn/Mn (Fig. EV3B), indicating that PacL1 preferentially binds Cu, followed by Zn, and then Cd.

Next, we performed NMR spectroscopy experiments to identify the specific residues in PacL1 responsible for zinc and cadmium binding. Chemical shift perturbation (e.g., CSP) of amide bonds showed that the cadmium-binding domain clearly overlaps with shifts linked to zinc coordination, from $H_{89}$ to $H_{93}$ (Figs. 5G,H and EV3C–F). Detailed analysis of the 2D $^1H$-$^{15}N$ HSQC spectra for increasing concentrations of zinc and cadmium revealed a slightly different binding mode of cadmium compared to zinc, as the amide signals in the presence of cadmium are less prone to signal broadening due to intermediate chemical exchange (Fig. EV3C–F). Other than $H_{91}D_{92}$, which were still broadened, the amide peaks involved in the cadmium interaction shifted to the fast exchange regime. This is consistent with a lower binding affinity for cadmium and indicating the cadmium-bound domain may adopt a distinct conformation.

Deletion of the C-terminal metal-binding motif (PacL1$^{Δ5}$) significantly reduced its metal-binding capacity (Figs. 5A–E and EV3A,B),

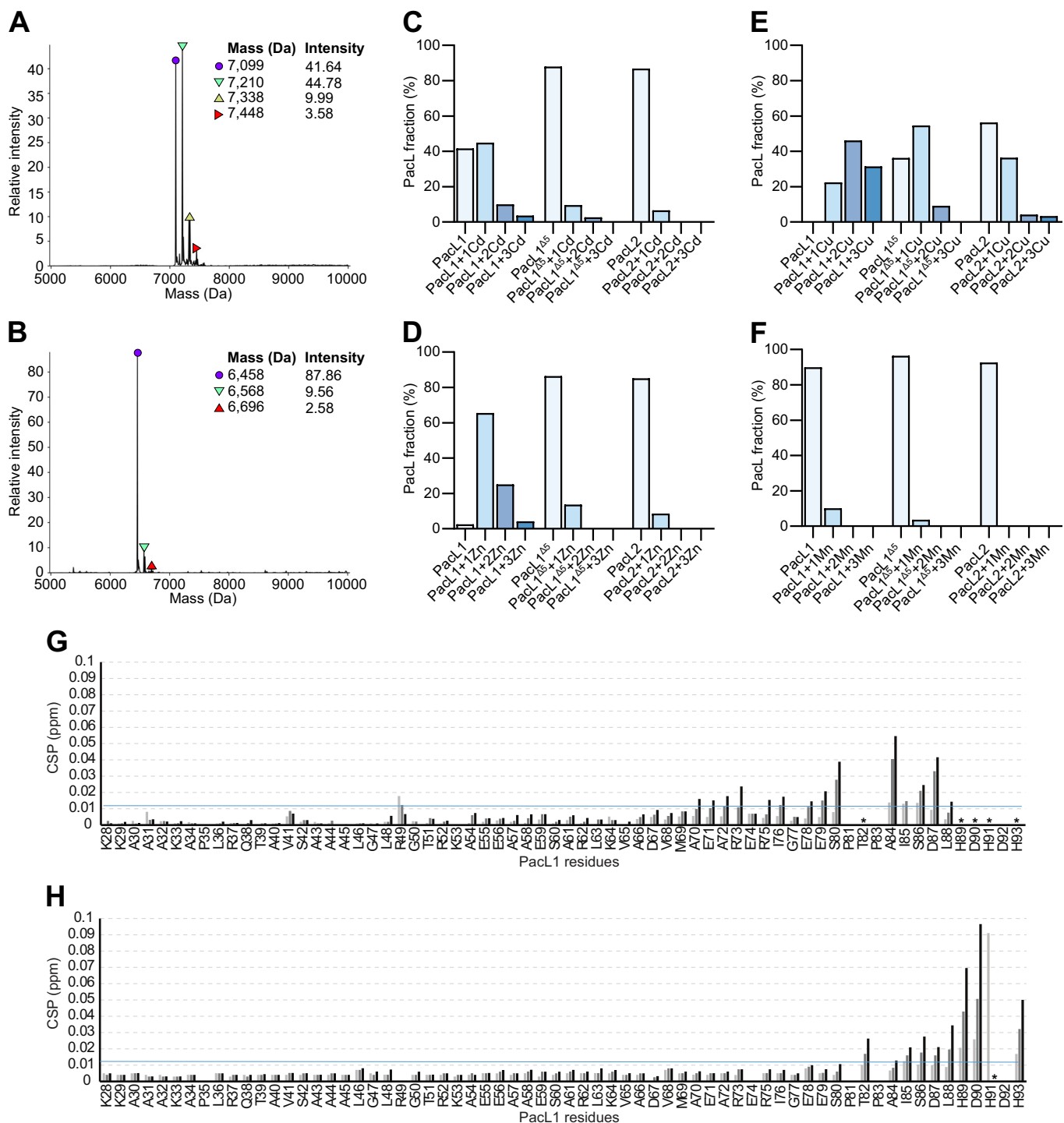

Figure 5.   Distinct metallochaperone activities of PacL family members.

(A, B) Deconvoluted mass spectra obtained by native MS showing cadmium binding to (A) PacL1 but not to (B) PacL1Δ5. (C–F) Proportion of the indicated purified proteins bound to specific numbers of metal atoms, assessed by native mass spectrometry. (G, H) Residue-specific perturbations induced by (G) zinc and (H) cadmium on PacL1. Summary of chemical shift perturbations of PacL1 upon zinc and cadmium interaction at 0.4 (light gray), 1.0 (gray), and 2 (black) equivalents of metal. Black stars indicate peak broadening below detection due to chemical exchange. PacL1: soluble domain of PacL1; PacL1Δ5: soluble domain of PacL1 deleted of its C-terminal metal-binding motif; PacL2: soluble domain of PacL2. Source data are available online for this figure.

confirming the critical role of this motif in metal coordination. Notably, PacL1$^{\Delta 5}$ was still able to bind one copper atom, suggesting that two copper atoms are coordinated through its canonical motif, while an additional atom may be bound at an alternative, unidentified site. To evaluate the metal-binding ability of PacL2, we conducted a similar analysis using its purified cytoplasmic domain. PacL2 bound copper, albeit with lower affinity than PacL1 (Fig. 5E), but displayed no detectable binding to zinc (Fig. 5D), cadmium (Fig. 5C), or manganese (Fig. 5F). Similarly, NMR spectroscopy confirmed that PacL2 does not bind zinc and cadmium (Appendix Fig. S1).

## The cytoplasmic domain of PacL proteins is not involved in cluster formation

The mechanism by which PacL proteins assemble into membrane clusters remains unclear. To assess whether the cytoplasmic domain of PacL proteins contributes to PacL-PacL interactions, we constructed *pacL2* deletion mutants lacking portions of this domain (PacL2$^{\Delta 55-84}$ and PacL2$^{\Delta 31-84}$) (Fig. 6A). The PacL2$^{\Delta 31-84}$ mutant appeared unstable (Appendix Fig. S2), but the PacL2$^{\Delta 55-84}$ variant was still able to form fluorescent foci in *M. smegmatis* (Fig. 6B), although a slight decrease in mean fluorescent signal per cell was observed (Fig. 6C). As expected, this deletion, which removed the AE repeats from PacL2 (Fig. 6A), decreased CtpG stability as determined by fluorescence microscopy (Fig. 6B,D), impaired the PacL2-CtpG interaction (Fig. 6E), and reduced cadmium tolerance (Fig. 6F).

We also generated *pacL1* deletion mutants lacking segments of its cytoplasmic domain (PacL1$^{\Delta 54-86}$ and PacL1$^{\Delta 37-86}$) (Fig. 6A). As reported in our previous study (Boudehen et al, 2022), both deletions reduced CtpC stability, as observed by epifluorescence microscopy and western blot (Fig. 6G–J). However, despite a diminished ability to form large clusters, both PacL1 deletion variants remained stable and continued to produce small fluorescent foci in the mycobacterial membrane (Fig. 6G). Overall, these results indicate that the cytoplasmic domain of PacL proteins, including the AE repeats, is not the primary determinant of PacL cluster formation, although it plays a crucial role in stabilizing interactions with P-ATPase pumps and ensuring proper protein function.

## Conserved TM-domain residues in PacL are essential for assembling functional multi-metal efflux machineries

To identify critical amino acids in the TM domain of PacL proteins essential for PacL/PacL interactions, we aligned 120 sequences of PacL-like proteins (DUF1490-containing proteins) identified in our previous study (Boudehen et al, 2022) across multiple bacterial species. The most highly conserved residues included a lysine (K9) and two glycines (G17 and G20), separated by two variable amino acids (Figs. 7A and EV4A). Notably, the two glycines form a groove in the TM domain (Fig. EV4B) resembling the GXXXG motif, a well-characterized sequence that facilitates helix-helix interactions in membrane proteins (Kleiger et al, 2002).

To investigate the role of these conserved residues in PacL/PacL interactions, we constructed *M. smegmatis* strains expressing PacL2 variants with substitutions of these residues (PacL2$^{G17L+G20L}$ and PacL2$^{K9A}$) (Fig. EV4B). To assess the functional impact of these mutations, we evaluated PacL2/CtpG-dependent cadmium tolerance using a disc diffusion assay. Both mutations significantly

reduced cadmium tolerance (Fig. 7B), indicating the importance of these residues for proper function. Interestingly, epifluorescence microscopy showed no reduction in fluorescence intensity in cells expressing CtpG-mVenus co-expressed with either PacL2, PacL2$^{G17L+G20L}$, or PacL2$^{K9A}$ (Fig. 7C,D). This suggests that while these mutations disrupt CtpG functionality, they do not impair the ability of PacL2 to stabilize the P-ATPase pump directly.

Interestingly, although the fluorescence intensity of PacL2-mTurquoise was weakly affected by the G17L + G20L and K9A mutations (Fig. 7E), we observed significant differences in cluster formation. In the absence of cadmium, while wild-type PacL2 formed multiple distinct fluorescent foci, the majority of cells expressing the PacL2$^{G17L+G20L}$ and PacL2$^{K9A}$ mutants exhibited no detectable foci (Fig. 7C). In the presence of cadmium, both mutants were still able to form large, immobile foci that colocalized with CtpG clusters. However, the number of small, mobile clusters, characteristic of wild-type PacL2, was strongly reduced in the PacL2$^{G17L+G20L}$ and PacL2$^{K9A}$ mutants (Fig. 7C; Movies EV1, 7, and 8). Additionally, the fluorescence signal of both mutants appeared more diffuse throughout the cells, suggesting that these mutations disrupt the ability of PacL2 to organize into distinct membrane clusters, resulting in a more homogeneous distribution of PacL proteins within the membrane.

To gain deeper insights into cluster formation in the PacL2$^{G17L+G20L}$ and PacL2$^{K9A}$ mutants, we constructed fusion proteins with mEos for super-resolution microscopy (Fig. 7F). Cluster segmentation of 3D localizations revealed that PacL2$^{G17L+G20L}$ and PacL2$^{K9A}$ formed an average of 6 and 8 clusters per cell, respectively, compared to 12 clusters in wild-type PacL2 (Fig. 7F,G). Additionally, both mutant proteins were more homogeneously distributed across the membrane, with only 10 and 19% of PacL2$^{G17L+G20L}$ and PacL2$^{K9A}$ molecules localizing within clusters, compared to 57% for the wild-type protein (Fig. 7F,H). Furthermore, the clusters formed by both mutants were significantly smaller than those formed by wild-type PacL2, with average volumes of $5.10^5$ and $2.10^6$ nm$^3$ for the mutants, compared to $4.10^6$ nm$^3$ for the wild-type (Fig. 7F,I). Altogether, these findings indicate that the GXXG motif and the K9 residue of the PacL2 transmembrane segment are essential for cluster formation. Moreover, they suggest that the function of CtpG depends on its localization within small, mobile clusters, rather than being associated with large, immobile aggregates or a homogeneously distributed membrane state.

## Interaction network of PacL1 with *M. tuberculosis* proteins

The results presented above reveal that PacL proteins act as hub molecules, organizing multiple P-ATPases into membrane-associated platforms. To investigate whether PacL-dependent clusters are limited to CtpC, CtpV, and CtpG or also include additional interacting proteins, we utilized a recently developed split ALFA-tag nanobody system for proximity labeling (Fay et al, 2025) to map the PacL1 interaction network in *M. tuberculosis*. For this analysis, we expressed a PacL1 protein fused to a C-terminal or an internal ALFA tag (Götzke et al, 2019) (PacL1$^{Cter-ALFA}$ and PacL1$^{Int-ALFA}$) in an *M. tuberculosis* $\Delta pacL1$ mutant, alongside a TurboID-nanobody fusion protein designed to bind the ALFA tag and biotinylate proteins in close proximity (Branon et al, 2018; Fay et al, 2025). The internal ALFA tag was incorporated between A$^{34}$ and P$^{35}$ of PacL1, downstream of its TM domain.

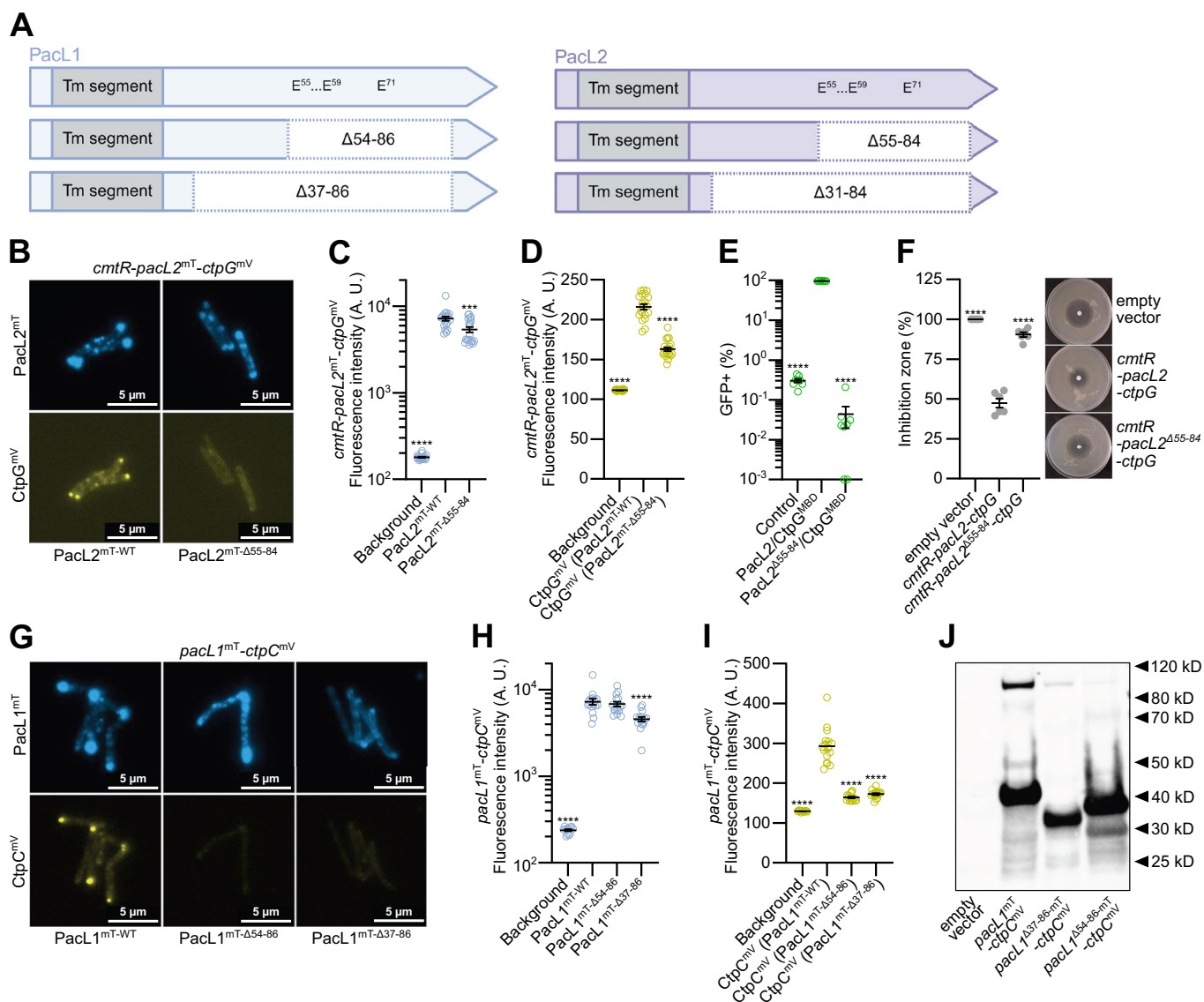

**Figure 6. The cytoplasmic domain of PacL proteins is not essential for PacL clustering.**

(A) Schematic representation of PacL1 and PacL2 proteins with deletions in the indicated segments of their cytoplasmic domains. Glutamic acid residues involved in PacL/Ctp interactions are indicated. (B) Epifluorescence microscopy images of live *M. smegmatis* strains carrying a genome-integrative vector expressing the indicated *M. tuberculosis* genes under the control of their native promoter, cultivated with 100 nM $CdSO_4$. (C, D) Mean fluorescence intensity (gray values) quantified in ImageJ from 8-bit grayscale images of cells of the indicated strains. Data were shown as mean ± SEM from individual cells, represented by yellow or blue dots. (E) Bipartite split-GFP assay in *M. smegmatis* expressing a PacL1-GFP11 fusion protein together with GFP1-10 (control) or the indicated proteins fused to GFP11 (first protein) or GFP1-10 (second protein). Fluorescence was recorded by flow cytometry and expressed as the percentage of GFP-positive cells. Data were shown as mean ± SEM from biological replicates, represented by green dots. (F) Cadmium ($CdSO_4$) sensitivity of *M. smegmatis* strains carrying a genome-integrative vector expressing the indicated *M. tuberculosis* genes under the control of their native promoter, assessed by disk diffusion assay. Inhibition zone diameters normalized to the empty vector control are shown as mean ± SEM from biological replicates, with individual values indicated by gray dots. (G) Epifluorescence microscopy images of live *M. smegmatis* strains carrying a genome-integrative vector expressing the indicated *M. tuberculosis* genes under the control of their native promoter. (H, I) Mean fluorescence intensity (gray values) quantified in ImageJ from 8-bit grayscale images of cells of the indicated strains. Data were shown as mean ± SEM from individual cells, represented by yellow or blue dots. (J) Western blot analysis of membrane extracts from *M. smegmatis* strains carrying a genome-integrative vector expressing the indicated *M. tuberculosis* genes under the control of their native promoter. Detection was performed using an anti-GFP antibody. CtpCmV: 103.7 kDa; PacL1mT: 36.9 kDa; PacL1mT Δ37-86: 31.7 kDa; PacL1mTΔ54-86: 33.4 kDa. For statistical analysis, one-way ANOVA with a Dunnett post-test were performed. Statistical analyses were conducted on ln-transformed data for (C, E, H). Asterisks indicate statistically significant differences compared to the reference condition: (C) PacL2mT-WT, (D) CtpGmV (PacL2WT), (E) PacL2/CtpGMBD, (F) empty vector, (H) PacL1mT-WT, (D) CtpCmV (PacL1WT) (***$P < 0.001$; ****$P < 0.0001$). Exact $p$ values and biological replicate numbers presented in this figure are reported in Table EV6. PacL2mT: PacL2-mTurquoise fusion; CtpGmV: CtpG-mVenus fusion; PacL1mT: PacL1-mTurquoise fusion; CtpCmV: CtpC-mVenus fusion; Δ55-84: PacL2 deleted for amino acids 55 to 84; Δ54-86: PacL1 deleted for amino acids 54 to 86; Δ37-86: PacL1 deleted for amino acids 37 to 86. Source data are available online for this figure.

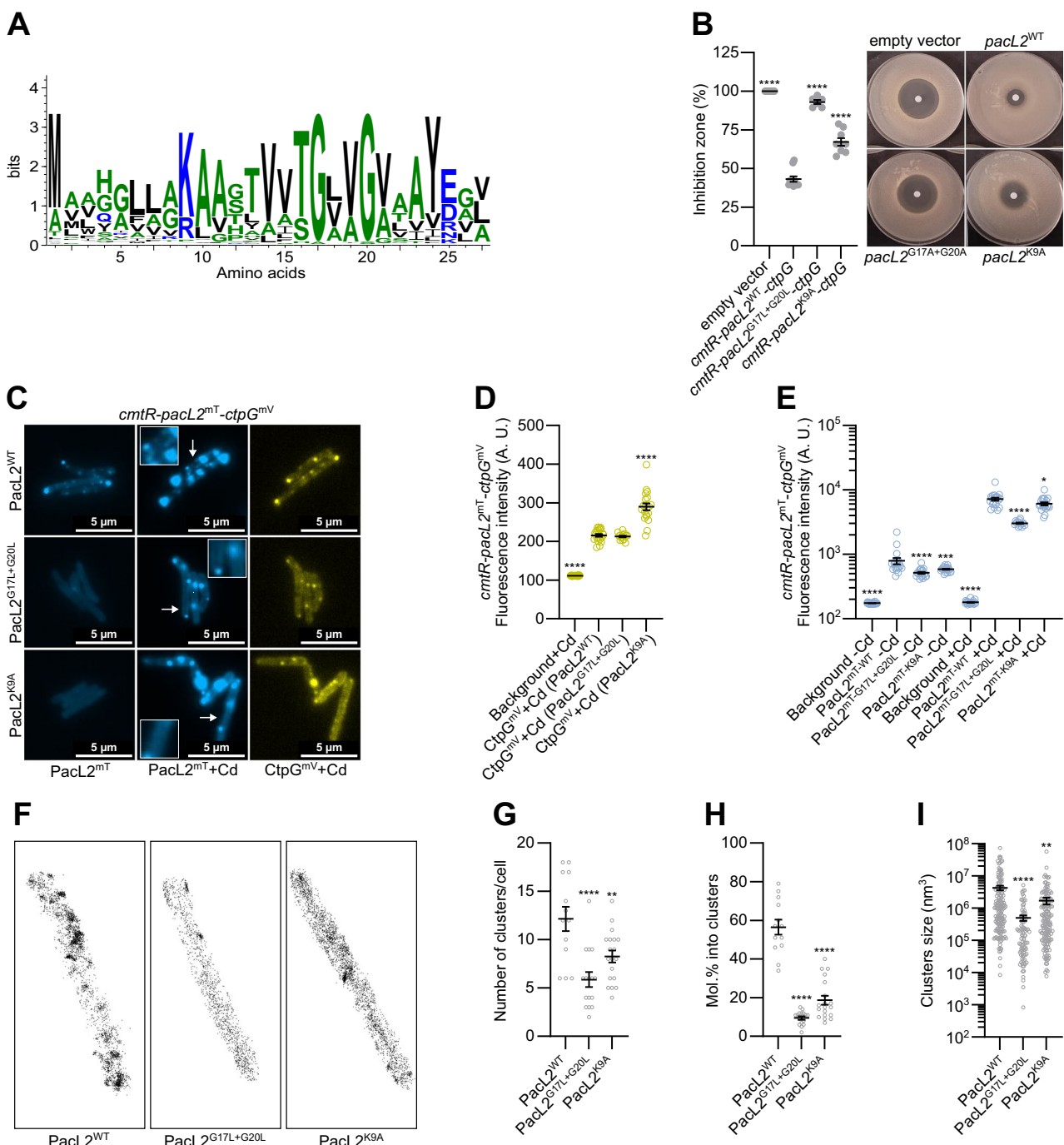

AlphaFold-predicted structures of PacL1^Cter-ALFA and PacL1^Int-ALFA indicated that the overall protein conformation may not be significantly affected by the addition of the ALFA epitope (Fig. EV5A). Immunoblot analyses confirmed that the engineered *M. tuberculosis* strains properly expressed the tagged PacL1 proteins and the TurboID-nanobody fusion (Fig. EV5B). Importantly, co-expression of PacL1^Cter-ALFA and PacL1^Int-ALFA with CtpC and the TurboID-nanobody fusion conferred zinc tolerance in both *M. smegmatis* (Fig. EV5C,D) and *M. tuberculosis* (Fig. EV5E), confirming that the binding of the fusion protein to PacL1 does not

abolish the formation or functionality of PacL-dependent membrane clusters and that the ALFA tag does not disrupt PacL1 function.

Biotinylated proteins were identified by proteomics and compared to those detected under control conditions, in which the ALFA epitope was fused to the first two transmembrane domains of the *E. coli* MalF protein (MalF_(1-2)-ALFA) and co-expressed with the TurboID-nanobody fusion. This provides a baseline for nonspecific interactions within the membrane. When co-expressed with a nanobody-GFP fusion, MalF_(1-2)-ALFA displays uniform fluorescence along the mycobacterial membrane (Fay

**Figure 7. Conserved amino acids of the TM domain of PacL proteins are essential to assemble functional PacL clusters.**

(A) Sequence conservation logo of the predicted transmembrane domain of 120 bacterial PacL-like proteins (DUF1490-containing proteins). (B) Cadmium (CdSO$_4$) sensitivity of *M. smegmatis* strains carrying a genome-integrative vector expressing the indicated *M. tuberculosis* genes under the control of their native promoter, assessed by disk diffusion assay. Inhibition zone diameters normalized to the empty vector control are shown as mean ± SEM from biological replicates, with individual values indicated by gray dots. (C) Epifluorescence microscopy images of live *M. smegmatis* strains carrying a genome-integrative vector expressing the indicated *M. tuberculosis* genes under the control of their native promoter, cultivated with or without 100 nM CdSO$_4$. Insets represent enlarged regions indicated by arrows. (D, E) Mean fluorescence intensity (gray values) quantified in ImageJ from 8-bit grayscale images of cells of the indicated strains. Results are shown as means ± SEM from individual cells, represented by yellow or blue dots. (F) Super-resolution 3D imaging of individual *M. smegmatis* cells carrying a genome-integrative vector expressing PacL2$^{mEos}$ (PacL2$^{WT}$, PacL2$^{G17L+G20L}$, or PacL2$^{K9A}$) under the control of its native promoter, cultivated with 100 nM CdSO$_4$, using PALM microscopy. Representative images are shown from dozens of analyzed cells. (G–I) Quantification of (G) the number of PacL2 clusters per cell, (H) cluster sizes, and (I) the proportion of molecules localized within a cluster per cell, determined via cluster segmentation of single-molecule localizations using the PoCA software platform. Results are presented as means ± SEM from individual cells (G, I) or individual clusters (H), represented by gray dots, pooled from $n = 3$ independent biological replicates. For statistical analysis, one-way ANOVA with a Dunnett post-test were performed. Statistical analyses were conducted on ln-transformed data for (E). Asterisks indicate statistically significant differences compared to the reference condition (empty vector (B) or WT (D, E, G–I)) (*$P < 0.05$; **$P < 0.01$; ****$P < 0.0001$). Exact *p* values and biological replicate numbers presented in this figure are reported in Table EV6. PacL2$^{mT}$: PacL2-mTurquoise fusion; CtpG$^{mV}$: CtpG-mVenus fusion; PacL2$^{mEos}$: PacL2-mEos fusion. Source data are available online for this figure.

et al, 2025), consistent with the ALFA epitope being cytosolic but membrane-anchored. We respectively identified 46 and 22 proteins that were at least twice as biotinylated in the strain expressing PacL1$^{Cter-ALFA}$ and PacL1$^{Int-ALFA}$ compared to the strain expressing MalF$_{(1,2)}$-ALFA (Tables 1 and EV1). Among them, seven were enriched in both PacL1$^{Cter-ALFA}$ and PacL1$^{Int-ALFA}$ strains: PacL1, Rv1265, CtpC, PPE20, MoeX, Rv2083, and AhpC. According to Gene Ontology (GO) annotations from QuickGO, among the 61 identified proteins, 25 are predicted to localize to the plasma membrane or cell wall, seven to the cytoplasm, four to extracellular compartments, while the subcellular localization of the remaining proteins remains unknown (Tables 1 and EV1). Notably, PacL1, CtpC, CtpV, and PacL3 were enriched, confirming that they are part of the effluxosome. The absence of PacL2 and CtpG from the identified proteins is likely due to their low expression levels, as the experiment was conducted without cadmium supplementation, which typically induces their expression.

The analysis revealed a diverse array of proteins with various functions. Among them, we identified several PE/PPE family proteins, so named for their conserved N-terminal Pro-Glu (PE) or Pro-Pro-Glu (PPE) motifs, including PPE20, PPE41, and Rv2083, as well as components of the ESX-1 secretion system, such as EsxB (CFP-10), Rv3877, Rv3878, and Rv3881c, which are known to play roles in *M. tuberculosis* virulence and immune modulation (D'Souza et al, 2023; Gröschel et al, 2016). Additionally, several proteins, including AhpC, CspA, and GreA, have previously been implicated in *M. tuberculosis* adaptation to environmental stress (Wong et al, 2017; Köster et al, 2017; Feng et al, 2020). Finally, the identification of multiple proteins with unknown functions, such as Rv1265, which was strongly enriched in both PacL1$^{Cter-ALFA}$ and PacL1$^{Int-ALFA}$ strains, suggests the presence of potential novel players in metal detoxification and/or stress adaptation. These findings indicate that the effluxosome could extend beyond CtpC, CtpG, and CtpV, encompassing a broader network of proteins that may enhance the ability of *M. tuberculosis* to withstand environmental stresses and contribute to its pathogenic potential.

## Discussion

In this study, we reveal that multiple P-ATPase pumps assemble within dynamic membrane-associated platforms, which we term

effluxosomes (Fig. 8). Our insights open up new questions about the biology and mechanisms underlying stress responses in Mycobacteria, and potentially other bacteria.

Specifically, we found that CtpC and CtpG confer cross-resistance to zinc and cadmium, while the function of CtpV remains unclear. We showed that PacL1, PacL2, and PacL3 cluster in the mycobacterial membrane *via* transmembrane GXXG motifs and recruit CtpC, CtpG, and CtpV through EA repeats, forming dynamic membrane complexes essential for P-ATPase function. We also demonstrated that PacL1 binds multiple metals, enhancing P-ATPase-mediated tolerance (Boudehen et al, 2022).

Building on previous research, we recently discussed various examples of bacterial membrane proteins that form clusters, highlighting the potential advantages of such assemblies within membranes (Dupuy et al, 2023). These benefits include promoting oligomerization, protecting proteins from degradation, and facilitating preferential subcellular organization. Here, our findings indicate that the clustering of P-ATPase pumps by PacL proteins not only prevents their degradation but is also essential for their functionality. Future studies will elucidate the functional implications of these clusters in the context of pathogenesis. Another intriguing insight into the potential role of clustering different P-ATPases arises from the ability of PacL1 to bind multiple metals. This activity suggests the hypothesis that PacL1 can locally concentrate various metal ions, potentially enhancing the efflux efficiency of P-ATPases specialized in exporting specific ions. Our previous study demonstrated that zinc binding by PacL1 enhances the ability of CtpC to confer metal tolerance (Boudehen et al, 2022). However, whether direct metal transfer occurs between PacL1 and individual P-ATPases remains to be determined.

Fluorescence microscopy combined with single-molecule localization and tracking revealed three distinct populations of PacL proteins with diverse dynamic behaviors. First, we identified a population of confined molecules forming large, stable clusters predominantly localized at the cell poles. This immobile fraction may represent non-functional assemblies, as similar clusters persist even when essential residues for effluxosome-mediated metal tolerance (G17, G20, and K9) are mutated. Second, we observed transiently confined molecules, likely representing small, slowly mobile clusters. We hypothesize that these structures correspond to functional effluxosomes, where multiple P-ATPases and PacL proteins are clustered to optimize metal efflux efficiency. Third, we

**Table 1.** Network of PacL1 interactions with *M. tuberculosis* proteins.

| Genes | Protein descriptions | Fold change (log2) | p value (−log10) | Subcellular location* |
|---|---|---|---|---|
| pacL1 | DUF1490 family protein | 6,34 | 5,91 | Plasma membrane and cell wall |
| Rv1265 | Uncharacterized protein | 3,74 | 4,40 | Plasma membrane and cell wall |
| ctpV | Metal cation transporter P-type ATPase | 2,29 | 3,33 | Plasma membrane and cell wall |
| ctpC | Metal cation transporter P-type ATPase | 2,08 | 5,12 | Plasma membrane |
| PPE20 | PPE family protein | 1,93 | 4,29 | Secreted |
| efp | Elongation factor P | 1,65 | 2,34 | Cytoplasm |
| Rv3898c | Uncharacterized protein | 1,58 | 2,45 | Unknown |
| Rv3559c | Short-chain dehydrogenase | 1,54 | 3,28 | Unknown |
| citE | Citrate (Pro-3S)-lyase beta subunit | 1,53 | 3,69 | Unknown |
| Rv2302 | DUF1918 domain-containing protein | 1,52 | 1,42 | Cell wall |
| Rv2114 | Uncharacterized protein | 1,47 | 2,26 | Cytosol |
| Rv0123 | DNA-binding protein | 1,44 | 1,50 | Plasma membrane |
| pacL3 | DUF1490 family protein | 1,44 | 4,58 | Plasma membrane and cell wall |
| Rv3357 | Antitoxin | 1,44 | 3,91 | Unknown |
| Rv3878 | ESX-1 secretion-associated protein EspJ | 1,43 | 2,65 | Secreted |
| Rv1637c | Metallo-beta-lactamase superfamily protein | 1,37 | 2,15 | Unknown |
| moeX | Molybdopterin biosynthesis protein | 1,36 | 2,45 | Unknown |
| sahH | Adenosylhomocysteinase | 1,34 | 3,47 | Cytoplasm |
| Rv2857c | Short-chain dehydrogenase | 1,33 | 3,99 | Unknown |
| hns | Histone-like protein | 1,32 | 2,66 | Plasma membrane |
| TB9.4 | ATP-binding protein | 1,30 | 1,83 | Unknown |
| Rv3788 | Nucleoside diphosphate kinase regulator | 1,26 | 1,85 | Unknown |
| rpmG2 | Large ribosomal subunit protein bL33 | 1,24 | 2,12 | Cytoplasm and ribosome |
| Rv2083 | PPE family protein | 1,22 | 1,53 | Plasma membrane |
| greA | Transcription elongation factor GreA | 1,21 | 2,21 | Cytosol, plasma membrane, and cell wall |
| Rv0121c | Pyridoxamine 5′-phosphate oxidase | 1,21 | 2,46 | Unknown |
| PPE41 | Uncharacterized protein | 1,20 | 2,90 | Secreted |
| Rv0250c | Uncharacterized protein | 1,19 | 4,26 | Unknown |
| mog | Molybdopterin biosynthesis Mog protein | 1,19 | 3,63 | Unknown |
| Rv2830c | Uncharacterized protein | 1,16 | 2,02 | Unknown |
| Rv1566c | Inv protein | 1,16 | 2,39 | Unknown |
| Rv3474 | Probable transposase | 1,15 | 3,93 | Plasma membrane |
| ahpC | Alkyl hydroperoxide reductase subunit C | 1,15 | 4,84 | Cytosol, plasma membrane, and cell wall |
| cspA | Cold shock protein A | 1,12 | 3,25 | Cytoplasm and cell wall |
| Rv3697A | CopG family DNA-binding protein | 1,10 | 2,14 | Unknown |
| argR | Arginine repressor | 1,10 | 2,45 | Cytoplasm |
| esxB | ESAT-6-like protein | 1,08 | 2,27 | Secreted, plasma membrane, and cell wall |
| Rv3881c | Uncharacterized protein | 1,07 | 3,14 | Secreted |
| galU | UTP--glucose-1-phosphate uridylyltransferase | 1,07 | 3,11 | Plasma membrane |
| fadD30 | Acyl-CoA synthetase | 1,04 | 2,85 | Unknown |
| hbhA | Iron-regulated heparin-binding hemagglutinin | 1,03 | 3,56 | Cytosol, plasma membrane, and cell wall |
| Rv3421c | Gcp-like domain-containing protein | 1,01 | 2,07 | Cytosol and cell wall |
| Rv3046c | DUF3349 domain-containing protein | 1,01 | 2,59 | Unknown |
| frdA | Fumarate reductase flavoprotein subunit | 1,01 | 1,83 | Plasma membrane |

**Table 1.** (continued)

| Genes | Protein descriptions | Fold change (log2) | p value (−log10) | Subcellular location* |
|---|---|---|---|---|
| *Rv0887c* | VOC domain-containing protein | 1,01 | 2,44 | Unknown |
| *mihF* | Integration host factor | 1,01 | 1,30 | Cytosol, plasma membrane, and cell wall |

*M. tuberculosis* proteins were significantly more biotinylated (Log2 fold change >1; −log10 *p* value >1.3) in a strain expressing a TurboID-nanobody fusion protein along with the PacL1 protein containing a C-terminal ALFA tag (PacL1[Cter-ALFA]) or an internal α-tag (PacL1[Int-ALFA]), compared to a strain expressing the TurboID-nanobody fusion protein with the TM domain of the *E. coli* MalF protein carrying an ALFA-tag (MalF[(1,2)]-ALFA).

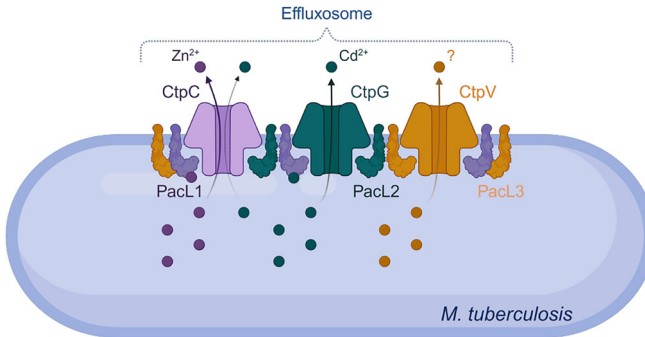

**Figure 8. Conceptual model of mycobacterial effluxosome for multi-metal cross-resistance.**

The P-ATPases CtpC and CtpG facilitate the efflux of zinc and cadmium from the cell, while the role of CtpV remains unclear. The PacL1, PacL2, and PacL3 proteins interact with each other through the GXXG motif in their transmembrane domains and with the metal-binding N-terminal domains of CtpC, CtpG, and CtpV via EA repeat sequences in their cytoplasmic domains. These interactions are essential for the metal resistance mediated by these P-ATPases and for clustering them into dynamic mobile membrane complexes. PacL1, but not PacL2, binds a variety of metal ions, thereby enhancing the metal tolerance conferred by the P-ATPases.

detected a population of highly mobile molecules diffusing throughout the bacterial cell. Notably, some of these molecules transitioned from a mobile state to a confined one, suggesting that freely diffusing PacL proteins can dynamically integrate into effluxosomes. Given the metal-binding capacity of some PacL proteins, we propose that this mobile pool may function as "patrollers", roaming the cell to capture metals and deliver them to effluxosomes. Alternatively, highly mobile PacL proteins may "scan" the membrane for metal-rich hotspots (Victor et al, 2018), thereby promoting the localized assembly of effluxosomes in response to metal accumulation.

While these hypotheses require experimental validation, the observed functional partitioning based on mobility suggests that effluxosomes are not merely static structures but dynamic hubs for stress sensing and response. This behavior is reminiscent of membrane nanodomains in eukaryotic cells (Li et al, 2024), and may reflect a conserved bacterial strategy to counteract host-imposed metal stress, shedding light on mechanisms of microbial resilience and virulence. The lipid composition of effluxosomes and the potential influence of specific membrane lipids on their dynamics and metal efflux activity remain unknown. Future studies will address these questions, e.g., using the recently developed styrene-maleic acid anhydride lipid particle (SMALP) technology (Bada Juarez et al, 2019).

To explore the broader relevance of these findings, we examined the distribution of PacL proteins across bacterial species. In a previous study (Boudehen et al, 2022), we identified 98 bacterial species encoding at least one PacL protein (containing a DUF1490 domain), all of which belong to the Actinobacteria phylum. The majority of these species (84%) harbor a single *pacL* gene, which is constantly in operon with a gene encoding a P-ATPase pump. Interestingly, 10% of these species encode two PacL/Ctp pairs, 4% encode three pairs, and 2% encode five pairs. This distribution suggests that effluxosomes may not be unique to *M. tuberculosis* but could also be present in other pathogenic or opportunistic bacteria, such as *Mycobacterium marinum*, which notably encodes five PacL/Ctp pairs. Moreover, all identified organisms encode at least one PacL protein with a predicted C-terminal metal-binding motif (Boudehen et al, 2022), supporting the notion that the metallochaperone function of PacL1 is conserved across effluxosome systems. PacL proteins share close similarity with DUF6110-containing proteins, which we refer to as PacL-like proteins. These homologs are primarily found in Firmicutes but also occur in other bacterial phyla, including Actinobacteria and Proteobacteria, as well as in Archaea. With few exceptions, these organisms encode only a single PacL-like protein. However, since our study demonstrates that each PacL protein can interact with multiple P-ATPase pumps, we cannot exclude the possibility that a single PacL protein might associate with P-ATPases encoded elsewhere in the genome. This raises the intriguing possibility that effluxosome-like assemblies could form even in bacteria encoding only one PacL or PacL-like protein.

Given the role of P-ATPases in metal tolerance, we also investigated the specific contribution of CtpG. Previous studies have reported that CtpG contributes to tolerance against zinc, cadmium, and copper (Chen et al, 2022; López et al, 2018). In our study, we assessed the ability of CtpG to confer tolerance to multiple metals, including cadmium, zinc, copper, nickel, manganese, and iron. Our findings reveal that CtpG specifically plays a role in cadmium tolerance, but does not contribute to resistance against the other metals. This discrepancy with previous reports could, in part, be explained by the requirement of PacL2 for CtpG function, as enzymatic activity measured in isolation from the P-ATPase may not reflect optimal physiological conditions.

The relevance of cadmium tolerance in *M. tuberculosis* infection remains an open question. Although cadmium has no known biological function in bacteria, it interacts with nucleic acids, competes with essential metal cofactors in proteins, and induces oxidative stress by promoting reactive oxygen species production (Wang and Crowley, 2005). However, its potential role in pathogen clearance by the immune system remains unexplored. Notably, *ctpG* expression is induced during macrophage infection (Tailleux et al, 2008) and has been shown to contribute to bacterial growth within the host (Chen et al, 2022). The cadmium specificity of CtpG raises the intriguing question of whether *M. tuberculosis* encounters toxic cadmium concentrations during infection. Cadmium, largely derived from environmental pollution and cigarette smoke, accumulates in alveolar macrophages (Grasseschi et al, 2003), the

natural niche of *M. tuberculosis*, suggesting that the pathogen is likely exposed to this metal in the host. Future studies should investigate whether macrophages locally elevate cadmium concentrations in the phagosome to intoxicate *M. tuberculosis*, as has been previously reported for zinc (Botella et al, 2011).

Beyond cadmium, we also examined the role of CtpV, a P-ATPase previously implicated in copper tolerance (Ward et al, 2010). A previous study demonstrated that an *M. tuberculosis ctpV* mutant exhibits increased sensitivity to copper, suggesting that CtpV plays a role in copper efflux (Ward et al, 2010). This would be consistent with the fact that CtpV clusters with copper efflux pumps in phylogenetic analyses (Neyrolles et al, 2013). However, in this study, we did not observe an impact of *ctpV* expression in *M. smegmatis* on tolerance to copper or other metals. This finding is particularly unexpected given the strong induction of the *ctpV* operon in response to copper (Ward et al, 2010; Liu et al, 2007). Although no increased copper sensitivity was detected in an *M. tuberculosis* triple *pacL* mutant (data not shown), it remains possible that CtpV depends on an unidentified partner encoded in the *M. tuberculosis* genome, but absent from *M. smegmatis*, for its function, or that functional redundancy with one or more copper detoxification pathways masks its contribution. Alternatively, CtpV may not primarily function in copper detoxification but instead facilitates copper transport to deliver it as a cofactor to periplasmic and membrane-associated proteins, as has been proposed for CtpA (López-Ruíz et al, 2025) and CtpB (León-Torres et al, 2020). Future studies will be needed to clarify the precise role of CtpV in *M. tuberculosis* and its potential involvement in effluxosome function.

Interestingly, our findings suggest that the role of the effluxosome extends beyond metal detoxification. Several proteins in proximity to PacL1 appear to have broader functions in stress adaptation, immune response modulation, and virulence. Among them, Rv1265 is particularly intriguing, as it exhibited a 13- and a 22-fold increase in biotinylation in strains expressing PacL1[Cter-ALFA] and PacL1[Int-ALFA], respectively, compared to the control. Rv1265 is known to be induced early during macrophage infection and responds to starvation and elevated cAMP levels under hypoxic conditions (Luo et al, 2016), suggesting a role in mycobacterial stress adaptation. Notably, Rv1265 protein has been reported to associate with the bacterial cell envelope, where it modulates cell envelope composition and enhances mycobacterial survival within macrophages (Luo et al, 2016). However, a more recent study suggests that Rv1265 functions as an ATP-binding transcription factor regulating the expression of the small non-coding RNA Mcr11 (Girardin et al, 2018). How these various functions of Rv1265 are coordinated and relate to effluxosomes remains to be determined. Another protein of interest is PPE20, which was enriched 4- and 5-fold in PacL1[Cter-ALFA] and PacL1[Int-ALFA] strains, respectively. Interestingly, PPE20 was recently identified as a transporter involved in calcium translocation across the *M. tuberculosis* outer membrane (Boradia et al, 2022). Future studies will be crucial to determine whether Rv1265, PPE20, and other identified proteins form stable, membrane-associated clusters in mycobacteria and to elucidate their specific functional roles within these newly identified membrane machineries.

Overall, our study provides new insights into metal ion stress, indicating that bacterial pathogens organize specialized membrane platforms that can resist toxic conditions. The effluxosome concept also captures a unique cross-resistance strategy that enables *M.*

*tuberculosis* to withstand stress induced by multiple metals, and possibly other compounds, encountered within the phagosome (Neyrolles et al, 2015). Given that CtpC, CtpG, and CtpV contribute to mycobacterial growth within the host (Botella et al, 2011; Padilla-Benavides et al, 2013; Hanna et al, 2021; Sassetti and Rubin, 2003; Chen et al, 2022; Ward et al, 2010), and that other proteins, including stress-response factors, may also be part of the effluxosome, our findings have important implications for the development of novel antimicrobial strategies. Future research will reveal whether the effluxosome plays a broader role in stress adaptation beyond metal detoxification, and whether it contributes to *M. tuberculosis* virulence.

# Methods

## Reagents and tools table

| Reagent/resource | Reference or source | Identifier or catalog number |
|---|---|---|
| **Experimental models** | | |
| Strains used in this study | See Table EV2 | See Table EV2 |
| **Recombinant DNA** | | |
| Plasmids used in this study | See Table EV4 | See Table EV4 |
| **Antibodies** | | |
| Anti-eGFP | Abcam | F56-6A1.2.3 RRID: AB_889471 |
| Anti-ALFA HRP | Genscript | A01861 RRID: AB_3083750 |
| Anti-BirA | Invitrogen | PA5-80251 RRID: AB_2787583 |
| Anti-RpoB | Biolegend | 663905 RRID: AB_2566583 |
| Goat anti-mouse IgG (H + L), HRP conjugate | Advansta | R-05071 RRID: AB_10718209 |
| **Oligonucleotides and other sequence-based reagents** | | |
| Primers used in this study | See Tables EV3 and 5 | See Tables EV3 and 5 |
| **Chemicals, enzymes and other reagents** | | |
| LB broth (Lennox) | Merck | L3022 |
| LB broth with agar (Lennox) | Merck | L2897 |
| Difco Middlebrook 7H9 broth | Becton Dickinson (BD) | 271310 |
| Difco Middlebrook 7H10 broth | Becton Dickinson (BD) | 262710 |
| Difco Middlebrook 7H11 broth | Becton Dickinson (BD) | 212203 |
| Ampicillin | Merck | A9518-25G |
| Hygromycin B | Invivogen | Ant-hg1 |
| Kanamycin | Fisher Scientific | 11-815-032 |

| Reagent/resource | Reference or source | Identifier or catalog number |
|---|---|---|
| Streptomycin | Fisher Scientific | 11-860-038 |
| Zeocin | Invivogen | ant-zn-1 |
| Isopropyl β-D-1-thiogalactopyranoside | Merck | I6758 |
| Cytiva Whatman™ Antibiotic Assay Disks | Cytiva | 2017013 |
| Zinc sulfate | Merck | 221376 |
| Cadmium sulfate | Merck | 383082 |
| Copper(II) sulfate | Merck | 451657 |
| Nickel(II) sulfate hexahydrate | Merck | 227676 |
| Manganese(II) chloride | Merck | 244589 |
| Iron(II) sulfate heptahydrate | Merck | 215422 |
| Copper chloride | Merck | 751944 |
| In-Fusion® Snap Assembly Master Mix | Takara | 638948 |
| PrimeSTAR GXL DNA Polymerase | Takara | R050A |
| Agarose D5 | EUROMEDEX | D5-E |
| Certified Low-Melt Agarose | Bio-Rad | 1613111 |
| GeneRuler 1 kb DNA ladder | Thermo Scientific | SM0311 |
| GenJet Gel Extraction Kit | Thermo Scientific | K0691 |
| GenJet Plasmid Miniprep Kit | Thermo Scientific | K0503 |
| QIAprep Spin Miniprep kit | Qiagen | 27104 |
| High precision microscope cover glasses | Marienfeld Superior | 0107052 |
| Frame-Seal™ in situ PCR and Hybridization Slide | Bio-Rad | SLF0601 |
| GSTrap FF column | Cytiva | 17513101 |
| Superdex 75 Increase 10/300 GL column | Cytiva | 29148721 |
| 4–12% Bis-Tris SDS-polyacrylamide gels | Thermo Scientific | NP0321 |
| 0.2 µm PVDF membranes | Bio-Rad | 1704272 |
| Precision Plus Protein Kaleidoscope Prestained Protein Standards | Bio-Rad | 1610375 |
| WesternBright Quantum HRP substrate | Advansta | K-12042 |
| **Software** | | |
| Clone manager 9.51 | https://scied.com/index.htm | |
| Prism | GraphPad | |
| ImageJ | https://imagej.net/ij/ | |
| ThunderSTORM (ImageJ plugin) | (Ovesný et al, 2014) | |
| PoCA (Point Cloud Analyst) | (Levet and Sibarita, 2023) | |
| PALM_Tracer module | (Butler et al, 2022) | |
| NIS-Elements AR | Nikon | |
| MassLynx 4.1 | Waters | |

| Reagent/resource | Reference or source | Identifier or catalog number |
|---|---|---|
| UniDec 4.4.0 | (Marty et al, 2015) | |
| TopSpin 4.1.4 | Bruker | |
| CARA (Computer Aided Resonance Assignment) | (Keller, 2004) | |
| Sparky 3.190 | University of California | |
| FlowJo v10 | BD Biosciences | |
| Spectronaut 19.1 | Biognosys | |
| **Other** | | |
| Eclipse Ti-E/B wide-field epifluorescence microscope | Nikon | |
| Ti-E/B inverted microscope with Perfect Focus System and N-STORM TIRF filter set | Nikon | |
| Neo SCC-02124 sCMOS camera | Andor Technology | |
| iXon Ultra DU897 EM-CCD camera | Andor Technology | |
| SpectraX LED light source | Lumencor | |
| Cube 405-100 C, 405 nm laser | Coherent | |
| Sapphire 561 nm laser | Coherent | |
| MicAO 3DSR deformable mirror | Imagine Optic | |
| LSR Fortessa flow cytometer | BD Biosciences | |
| Densimat turbidity meter | BioMérieux | |
| NanoElute2 nanoLC system | Bruker | |
| TimsTOF HT mass spectrometer with Captive Spray source | Bruker | |
| Synapt G2-Si mass spectrometer | Waters | |
| TriVersa Nanomate | Advion Biosciences | |
| Emulsiflex-C5 homogenizer | Avestin | |
| Odyssey XF Imaging System | LI-COR Biosciences | |

## Methods and protocols

### Bacterial strains

The bacterial strains used in this study are listed in Table EV2. *Escherichia coli* strains StellarTM (Takara bio), used as a recipient for plasmid constructions and BL21 Star (DE3), used for protein production, were grown at 37 °C in LB broth (Difco) or on L-agar plates (Difco) supplemented with streptomycin (25 µg.ml$^{-1}$), ampicillin (100 µg.ml$^{-1}$), or kanamycin (40 µg.ml$^{-1}$) when required. The *Mycobacterium smegmatis* mc$^2$155 strains (ATCC 700084) were grown in Middlebrook 7H9 medium (Difco) at 37 °C, supplemented with 0.5% glycerol, 0.5% dextrose, and 0.05% Tween-80. The *M. smegmatis* strains used in this study had deletions in the genes encoding PacL1 (MSMEG_6059), CtpC (MSMEG_6058), and the ZitA cation diffusion facilitator (MSMEG_0755) (Boudehen et al, 2022). No additional PacL

proteins are encoded in *M. smegmatis*. *M. tuberculosis* H37Rv (ATCC 27294) strains were grown at 37 °C in 7H9 medium (Difco) supplemented with 10% albumin-dextrose-catalase (ADC, Difco) and 0.05% Tween-80 (Sigma-Aldrich), or on complete 7H11 solid medium (Difco) supplemented with 10% oleic acid-albumin-dextrose-catalase (OADC, Difco). When required, streptomycin (5 µg.ml$^{-1}$), hygromycin (50 µg.ml$^{-1}$), or zeocin (25 µg.ml$^{-1}$) were added to the culture media. Work with *M. tuberculosis* was conducted in a certified biosafety level 3 laboratory at the *Institut de Pharmacologie et de Biologie Structurale*, in accordance with national and institutional regulations.

### Generation of M. tuberculosis mutants

Mutant strains of *M. tuberculosis* H37Rv were generated via allelic exchange using recombineering, following established protocols (Boudehen et al, 2022; van Kessel and Hatfull, 2007; Boudehen et al, 2020). Approximately 0.5 kb DNA fragments flanking the target genes were amplified using PrimeSTAR GXL DNA polymerase (Takara Bio), genomic DNA from *M. tuberculosis* H37Rv, and specific primers (Table EV3). These fragments were fused with a zeocin-resistance cassette via a three-fragment PCR strategy, incorporating *M. tuberculosis dif* site variants for unmarked deletions. For recombineering, recipient strains of *M. tuberculosis* H37Rv, harboring pJV53H encoding recombineering enzymes (van Kessel and Hatfull, 2007), were cultured to mid-log phase in 7H9 medium. Recombineering enzymes were induced with 0.2% acetamide, followed by electrotransformation with 100 ng of linear AES for allelic exchange. After 48 h at 37 °C, zeocin-resistant clones were selected and expanded. PCR analysis confirmed successful allele replacement. The *dif* site-flanked zeocin-resistance cassette was excised via spontaneous XerCD-mediated recombination, and the pJV53H plasmid was lost by serial passaging in antibiotic-free medium, followed by phenotypic screening for sensitivity to antibiotics, as described previously (Boudehen et al, 2020).

### Plasmids

Plasmids and oligonucleotides used in this study are listed in Tables EV3–5. Plasmids were constructed in *Escherichia coli* Stellar™ (Takara Bio), and cloning was performed using the In-Fusion recombination cloning kit (Takara), as recommended by the manufacturer. DNA fragments were amplified via PCR using PrimeSTAR GXL DNA polymerase (Takara Bio), plasmid or genomic DNA from *Mycobacterium tuberculosis* H37Rv as template, and specific primer pairs as described in Table EV4 and listed in Table EV5. Plasmids for the expression of the *pacL* and *ctp* genes under their native promoters were generated by amplifying the open reading frames (ORFs) along with their 5′ flanking regions (500 bp) via PCR and inserting them into digested pDB60. For the ATC-inducible plasmids, ORFs were amplified without 5′ flanking regions and cloned into ClaI-digested pMSG419. Plasmids for bipartite split GFP were constructed by cloning the ORFs of the proteins of interest (PacL, MBD of Ctp, or Rv1488) along with the GFP N-terminal fragment (GFP1-10) or the GFP C-terminal fragment (GFP11), separated by a linker sequence, into the pGMC vector. Derivative plasmids containing deletions, substitutions, or adding tags to the relevant gene were constructed by In-Fusion cloning using the appropriate primer pairs as described in Table EV4 and listed in Table EV5. The presence of these changes and the absence of additional mutations in the constructs was verified by DNA sequencing.

### Disc diffusion assay

Bacteria were grown to the exponential phase, diluted to OD$_{600}$ = 0.01 in 3 ml of prewarmed (50 °C) top agar (7H9 with 6 mg/ml agar), and plated on 7H10 agar. A filter paper disc was placed on the solidified top agar and spotted with 2.5 µl of 1 M ZnSO$_4$, CdSO$_4$, CuSO$_4$, NiSO$_4$, MnCl$_2$, FeSo$_4$, or CuCl$_2$. After incubation at 30 °C for 72 h, the diameter of the growth inhibition zone was measured.

### M. tuberculosis cadmium sensitivity assay

*M. tuberculosis* cultures were initiated at an OD$_{600}$ of 0.01 in glass tubes containing 7H9 medium supplemented with varying concentrations of CdSO$_4$. The cultures were incubated statically at 37 °C, and bacterial growth was assessed over time by measuring turbidity in McFarland units using a Densimat device (BioMérieux).

### Bipartite split GFP

Log phase cultures of *M. smegmatis* strains transformed with relevant plasmids were grown in 7H9 medium containing 5 µg.ml$^{-1}$ streptomycin to an OD$_{600}$ of 0.05. After overnight incubation at 37 °C, the bacterial cells were analyzed using fluorescence-activated cell sorting (FACS). FACS was performed using an LSR Fortessa (BD Biosciences) flow cytometer. The GFP fluorescence signal was processed with FlowJo (v10) software.

### Western blotting

About 15 mL of *M. smegmatis* cultures at an OD$_{600}$ of 0.8 were harvested by centrifugation (10 min, 4000 rpm, 4 °C), washed twice with cold PBS containing 0.05% Tween-80, and resuspended in 1 mL of the same buffer. Cells were disrupted by sonication (four cycles of 10 pulses, 2 s per pulse, 60% amplitude). Unbroken cells and large debris were removed by centrifugation (13,000 rpm, 2 min, 4 °C). The supernatant was centrifuged for 1 h at 13,000 rpm, 4 °C to separate the cell wall (pellet) from the membrane and cytosolic fractions (supernatant). The latter was ultracentrifuged (45,000, 1 h, 4 °C) to obtain membrane (pellet) and cytosolic (supernatant) fractions. Membrane pellets were resuspended in 150 µL of 5% SDS and stored at -20 °C. Protein lysates (15 µL of each sample) were separated on 4–12% Bis-Tris SDS-polyacrylamide gels (Thermo Fisher Scientific, #NP0321) for 1 h at 150 V, followed by electroblotting onto 0.2-µm PVDF membranes (Bio-Rad, #1704272) using the Trans-Blot Turbo Transfer System (Bio-Rad, #1704150). Precision Plus Protein Kaleidoscope Prestained Protein Standards (Bio-Rad, #1610375) were used as molecular weight markers. Membranes were blocked for 1 h in 5% BSA in TBS containing 0.05% Tween-20 (TTBS) and incubated overnight at 4 °C with a 1:1000 dilution of an anti-eGFP monoclonal antibody (F56-6A1.2.3, Abcam). After washing, membranes were incubated for 1 h at room temperature with a 1:1000 dilution of an HRP-conjugated secondary antibody (Advansta, R-05071). Chemiluminescent signals were detected using the WesternBright Quantum HRP substrate (Advansta, K-12042) and imaged with the Odyssey XF Imaging System (LI-COR Biosciences).

### Construction of fluorescent strains

Fluorophores (mTurquoise, mVenus, or mEos3.2), linked via a five-amino acid flexible linker (LEGSG), were fused in frame to the C-terminus of PacL and Ctp proteins. The fluorescent fusions were encoded on an integrative vector carrying the attachment site (*attP*) from mycobacteriophage L5, which enables site-specific integration into the mycobacterial chromosome through the L5 integrase system. The plasmid integrates as a single copy either in an *M. smegmatis* strain lacking the *pacL/ctp* system or in wild-type *M. tuberculosis*. To preserve the native genetic context, the entire *M. tuberculosis* operons (*cmtR-pacL2-ctpG* or *pacL1-ctpC*), together with 500 bp of their upstream promoter regions, were cloned. This strategy ensures expression levels of the fluorescent fusion proteins that closely approximate physiological conditions.

### Fluorescence microscopy

Fluorescence microscopy was performed on mycobacterial cultures harboring plasmids expressing fluorescent reporters fused to the proteins of interest, cultivated in the absence or presence of various metals. Acquisitions were performed on either live (Figs. 2D,G,K, 4J, 6B,G, and 7C) or fixed (Figs. 2F and 4B,E) cells. Fixation was carried out with 4% paraformaldehyde (PFA) for 2 h in *M. tuberculosis* and 20 min in *M. smegmatis*, followed by three washes with PBS. Cultures were deposited as 1 μL drops onto a 1% agarose layer in growth medium, as previously described (Diaz et al, 2015). Imaging was conducted using an Eclipse TI-E/B wide-field epifluorescence microscope equipped with a phase-contrast objective (CFI Plan APO LBDA 100X oil NA 1.45) and Semrock filters for YFP (Ex: 500BP24; DM: 520; Em: 542BP27), CFP (Ex: 438BP24; DM: 458; Em: 483BP32), or FITC (Ex: 482BP35; DM: 506; Em: 536BP40). Images were captured with an Andor Neo SCC-02124 camera, using 30-50% illumination from a SpectraX LED light source (Lumencor) and 0.3–0.6 s exposure time depending on the fluorochrome used. The focal plane was determined in phase-contrast mode by adjusting the focus to the region of maximal contrast, which corresponds to the cell center due to its higher refractive index relative to surrounding regions. Time-lapse imaging was performed over a 45-second period, with an image captured every second. Image acquisition and analysis were performed using Nis-Elements AR software (Nikon) and ImageJ (v 1.53f51). Fluorescence intensity measurements were obtained from individual cells using the grayscale mean intensity of the ROI, as measured with the ImageJ analysis tool.

### Photoactivated localization microscopy (PALM) and cluster analysis

PALM was performed using an inverted Nikon Ti-E/B microscope equipped with the "Perfect Focus System" (PFS, Nikon), a piezo stage (Nano Z100-N - Mad City Labs), a CFI Apochromat TIRF 100X Oil Objective (NA 1.49), and a quad-band dichroic mirror (97335 Nikon N-STORM TIRF Filter Set). A TIRF illumination arm was used, specifically in a highly inclined and laminated optical sheet (HILO) illumination mode. The system included an EM-CCD ANDOR iXon Ultra DU897 camera and a thermostatic chamber set to 24 °C to minimize instrumental drift. Excitation was controlled with an AOTF. For mEos-EM photoconversion, we used a 405 nm laser (Coherent™ - Cube 405-100 C), while imaging was performed with a 561 nm laser (Coherent™, Sapphire) at a power of ~0.1 kW/cm². The 405-nm laser was applied at low power to enhance the photoconversion rate. The imaging laser (561 nm) was used

continuously, with an acquisition time of 30 ms per frame (between 10,000 and 20,000 images). Images were acquired using Nis-Elements AR software (Nikon). Point-spread function (PSF) astigmatism was introduced using adaptive optics with a deformable mirror (MicAO 3DSR - Imagine Optic™) placed in the detection path, just before the EM-CCD camera. This setup enabled axial localization of single fluorescent molecules. PSF calibrations were performed using commercial fluorescent beads (FluoSpheres™ Carboxylate-Modified Microspheres, 0.17 μm, orange fluorescent (540/560), 2% solids, Invitrogen™).

3D localization was performed using ThunderSTORM, an ImageJ plugin (Ovesný et al, 2014). For each frame in the image sequence, raw images were filtered using a wavelet filter (B-Spline, order 3, scale 2.0). Approximate molecular detections were identified using the local maximum method, followed by sub-pixel localization using the PSF elliptical Gaussian method (Fitting Radius: 6 pixels, Initial Sigma: 1.6). The plugin allowed data filtering based on specific criteria. In this study, we eliminated localizations from the first 1000 frames and retained only molecules detected with a localization precision of less than 50 nm. Strict environmental controls were applied to minimize sample drift, including air conditioning, a thermostatic chamber, the Nikon Perfect Focus System, and an x/y-controlled stage. Residual drift was estimated using an autocorrelation algorithm in Thunder-STORM software (Ovesný et al, 2014). SMLM dataset visualization was also performed using ThunderSTORM. The voxel size was determined to ensure three pixels per resolution unit, and localizations were blurred using a theoretical PSF corresponding to the expected resolution.

To analyze the cluster organization of our protein of interest, we used the 3D DBSCAN (Ester et al, 1996) module of PoCA (Point Cloud Analyst) software (Levet and Sibarita, 2023). PoCA utilizes 3D localization coordinates computed from 3D PALM data to compute clusters based on the localization density within a given sphere (minimum localizations: 2; minimum number of localizations per cluster: 0.5% of total localizations; maximum distance between two localizations: 40 nm). From the 3D localizations, we generated both 3D and 2D Voronoi diagrams, the former computed directly from the 3D coordinates, and the latter from their projection onto the 2D XY plane. Density maps were generated by computing the local localization density using the 2D Voronoi diagram, as described in (Levet et al, 2015; Levet, 2023). To obtain 3D envelopes, localizations corresponding to bacteria were separated from background localizations by applying a 250 nm cutoff distance on the 3D Voronoi diagram. A Poisson surface (Kazhdan et al, 2006) was then fitted to the localizations located at the bacterial boundary.

sptPALM acquisitions were performed on the same microscopy setup as the PALM acquisitions described above, using the 561 nm laser at a reduced power of ~0.1 kW/cm² to visualize the trajectories. The imaging laser (561 nm) was used continuously, with an acquisition time of 30 ms per frame for 20,000 images. Single-molecule localization and analysis was performed using the PALM_Tracer module (Butler et al, 2022), which combines wavelet filtering and two-dimensional Gaussian fitting for the localization (Kechkar et al, 2013), and a simulated annealing algorithm for the tracking. Localization detections were selected using a threshold of 18 gray levels in the second wavelet plane, and a maximum reconnection distance of 4 pixels (corresponding to a maximum

instantaneous velocity of $20\,\mu m.s^{-1}$), in order to optimize the tracking of fast-moving molecules. Reconnection artefacts were minimized by maintaining a low activation density of mEos molecules. Diffusion coefficients ($D$) were calculated by linear fitting the first four points of the mean square displacement (MSD) function (Sibarita, 2014), using the equation $MSD(t) = 4Dt + b$, where $b = MSD(t = 0)$ accounts for the localization error. The MSD was computed for each trajectory as:

$$MSD(n\Delta t) = \frac{1}{N-n}\sum_{i=1}^{N-n}\left[(x_{i+n} - x_i)^2 + (y_{i+n} - y_i)^2\right]$$

where $\Delta t$ is the time interval between frames, $N$ the total number of positions in the trajectory, and $n$ the time lag. Diffusion coefficients below $10^{-5}\,\mu m^2.s^{-1}$ were set to this minimum value.

### Expression and purification of SolPacL1 and SolPacL2

The soluble domain of PacL1 was expressed and purified as reported previously (Boudehen et al, 2022). The gene of the soluble domain of PacL2 (SolPacL2) was codon optimized and cloned into pExp-GST. After amplification and purification with the QIAprep Spin Miniprep Kit, the plasmid was subsequently transformed into BL21 Star (DE3). The production of unlabeled SolPacL2 was performed in Luria Broth medium supplemented with $100\,\mu g/mL$ Amp in a 1 L flask at $37\,°C$ under vigorous shaking at 170 rpm. SolPacL2 expression was induced during 4 h by the addition of 1 mM isopropyl-β-D-thiogalactopyranoside (IPTG) at $37\,°C$. Then, cells were harvested by centrifugation ($5000\times g$, 10 min, $4\,°C$) and stored at $-70\,°C$. The cells were suspended in 5 to 10 mL of lysis buffer per gram of wet cell (50 mM Tris-HCl, pH 7.5, 200 mM NaCl, 0.5% Tween 20, 1 mM phenylmethanesulfonyl fluoride (PMSF), 1 mM Dithiothreitol (DTT), DNAse ($0.01–1\,mg/mL$), and cOmplet™ EDTA-free protease inhibitor cocktail. Cells were lysed by passing the suspension through the Avestin® Emulsiflex-C5 (three times at 15,000 psi) at $4\,°C$. Lysate was centrifuged ($12,000\times g$, 30 min, $4\,°C$). The supernatant was filtered on a $0.22\,\mu m$ membrane filter and loaded at 1 mL/min onto $2\times5\,mL$ GSTrap™ FF column (Cytiva) conditioned in the equilibration buffer (50 mM Tris-HCl, pH 7.5, 200 mM NaCl, and 1 mM DTT). The column was washed with 10 CV of equilibration buffer supplemented with 0.5 mM PMSF and cOmplet™ EDTA-free protease inhibitor cocktail. The protein was eluted with 4 to 5 CV of elution buffer (50 mM Tris, pH 8.0, and 10 mM reduced glutathione). The 8-His-GST tag was cleaved by addition of TEV protease at a mass ratio 1:50 in the elution buffer supplemented with $500\,\mu M$ EDTA, 1 mM DTT, and $100\,\mu M$ PMSF. The cleavage reaction was performed overnight at $4\,°C$ in a Spectra/Por®3 dialysis membrane (MWCO: 3.5 kDa) against 2 L of dialysis buffer (50 mM Tris-HCl, pH 8.0, 125 mM NaCl, $100\,\mu M$ PMSF, and $500\,\mu M$ DTT). The cleaved SolPacL2 was concentrated using a 3000 MWCO PES Vivaspin (Cytiva) at $4\,°C$, then loaded on Superdex 75 Increase 10/300 GL column (Cytiva) and eluted ($V_E$ ~12.5 mL) with a buffer containing 25 mM MES, pH 7.0, and 200 mM NaCl. Each step of the purification process was characterized by SDS-PAGE (5–10%). The concentration of the purified SolPacL2 was determined by integration (e.g., 1.0 to 0.82 ppm) of the $^1H$ methyl groups of I, L, V residues by $^1H$ NMR with respect to the $116.6\,\mu M$ of sodium salt of 3-(Trimethylsilyl)-1-propanesulfonic acid sodium salt (DSS) used as internal reference. For NMR purposes, both $^{15}N$-labeled

SolPacL1 and SolPacL2 expression and purification were performed under similar conditions except that cells were grown in minimal M9 medium supplemented with $100\,\mu g/mL$ Amp, 5 g/L glucose, 2 g/L $NH_4Cl$ $^{15}N$, 2 mM $MgSO_4$, 0.1 mM $CaCl_2$, trace elements, and vitamins (Hoopes et al, 2015).

### Titration of zinc and cadmium by NMR spectroscopy

NMR experiments were performed on a 600 Avance III HD spectrometer (Bruker) equipped with a 5 mm triple resonance cryoprobe at 280 K in 25 mM MES buffer, pH 7.0, 200 mM NaCl, and $126.6\,\mu M$ DSS as chemical shift reference. The solution of $^{15}N$-labeled PacL1 at $90\,\mu M$ was titrated with 0, 0.2, 0.4, 0.65, 1, 1.5, and 2 equivalents of $ZnCl_2$ and $CdCl_2$. $^1H,^{15}N$-HSQC NMR data were acquired with 1024 and 256 data points, respectively, for $^1H$ and $^{15}N$ dimensions (e.g., 70.7 and 95.6 ms acquisition times, respectively) and with eight scans. All residues were assigned previously at 700 MHz (Avance III HD Bruker) on a $124\,\mu M$ $^{15}N$-$^{13}C$-labeled SolPAcL1 in 10 mM MES buffer, pH 6.5, and 100 mM NaCl. The sequential backbone resonance assignment was performed using the best version of HNCACB, HN(CO)CACB, and HNCO experiments (Lescop et al, 2007) with selective $^1H$ pulses centered at 8.5 ppm, covering a bandwidth of 4.0 ppm, and a standard HN(CA)CO experiment. NMR spectra were processed using Topspin 4.1.4, and assignment of resonances was performed with CARA (Keller, 2004). The chemical shift perturbations (CSP) on $^{15}N$-labeled PacL1 induced by the interaction of zinc and cadmium with $^{15}N$-labeled PacL1 were classically determined with a = 0.1 (Williamson, 2013) (or B = 9.9) using Sparky 3.190 (Sparky - NMR Assignment Program).

### Native mass spectrometry

Prior to native MS analysis, SolPacL samples at $100\,\mu M$ were desalted in 200 mM ammonium acetate, pH 7, supplemented with $500\,\mu M$ of zinc acetate dihydrate, manganese acetate tetrahydrate, copper (II) acetate hydrate or cadmium iodide (all from Euromedex, Souffelweyersheim, France) using Micro Bio-Spin devices (Bio-Rad, Marnes-la-Coquette, France). Samples were analyzed on a Synapt G2-Si mass spectrometer (Waters Scientific, Wilmslow, UK) running in sensitivity mode, positive ion mode and coupled to an automated chip-based nano-electrospray source (Triversa Nanomate, Advion Biosciences, Ithaca, NY). The voltage applied to the chip and the cone voltage were set to 1.6 kV and 20 V, respectively. The instrument was calibrated with a 2-mg/mL cesium iodide solution in 50% isopropanol. Raw data were acquired in the 1000–8000 m/z range with MassLynx 4.1 (Waters, Manchester, UK) and deconvoluted with UniDec 4.4.0 (Marty et al, 2015) using the following parameters: m/z range: 500–2000 Th; Gaussian smoothing: 10; charge range: 1–15; mass range: 5000–10,000 Da; sample mass every 1 Da; smooth charge states distributions; smooth nearby points: some; suppress artifacts: none; peak detection range: 22 Da, and peak detection threshold: 0.05.

### Proximity labeling

Strains for proximity labeling were prepared as follows. Cells were pre-grown in 7H9 OADC supplemented with $20\,\mu g/ml$ kanamycin and $20\,\mu g/ml$ streptomycin to an $OD_{600}$ of 1.5–2.0 in roller bottles (Corning, # 430195). Cells were then pelleted by centrifugation at $3700\times g$ for 10 min, washed once with 10 mL of biotin-free Sauton's media, subcultured into biotin-free Sauton's media maintaining

kanamycin and streptomycin selection, 3 × 25 ml cultures/strain, in 60 ml inkwell bottles (Nalgene, #342020-060) at a final OD$_{600}$ of 0.4. ATC (50 ng/ml) was added, and cultures were incubated at 37 °C for 18 h. Following ATC induction cells were again collected by centrifugation at 3700×g for 10 min at room temperature, resuspended in 2 ml antibiotic and ATC-free Sauton's media, transferred to 2 ml O-ring sealed tube, collected by centrifugation at 3700×g for 5 min at room temperature, and finally resuspended in 1 ml Sauton's media. Cultures were incubated for 3 h with 200 μM Biotin (Sigma, B4501) with shaking at 37 °C. Cells were collected by centrifugation at 20,000×g for 1 min at 4 °C and placed immediately on ice. Cell pellets were washed with 2 ml cold TBS (20 mM Tris, 150 mM NaCl, pH 7.5), then collected by centrifugation at 20,000×g for 1 min at 4 °C and cell pellets were frozen at −80 °C.

For lysate preparation, frozen cell pellets were thawed on ice and resuspended in 250 μL bead mix (1:1 mix, Biospec 11079101z and 1079107zx) and 1 ml cold TBS with protease inhibitors (PI) (Pierce, PIA32955). Lysis was performed by mechanical disruption 3 × 45 s (Biospec Minibeadbeater) with 5 min icing between runs. Unbroken cells/debris were collected by centrifugation at 20,000×g for 1 min at 4 °C, supernatants collected, and residual pellets lysed again with beads and 400 μL cold TBS with protease inhibitors. Pooled supernatants were mixed with 150 μL of 10x RIPA buffer detergents (TBS + 10% Triton X-100, 1% SDS, and 5% sodium deoxycholate), mixed by inversion, and centrifuged at 20,000×g. The supernatant was filtered twice through EMD Millipore™ Ultrafree™-CL Centrifugal Filter Devices with Durapore™ Membrane to remove any residual *M. tuberculosis*. For biotinylated protein capture, protein lysates were mixed with 80 μL of Streptavidin-Agarose (GoldBio, S-105-10) prewashed 3X in 1 ml TBS and incubated rotating at 4 °C for 18 h. Agarose beads were collected at 250×g for 1 min at 4 °C and washed 2xs with 1 ml RIPA/SDS (TBS + 0.1% Triton X-100, 0.1% SDS, and 0.5% sodium deoxycholate) followed by four washes with TBS.

For quantitation of biotinylated proteins, the beads were resuspended in 80 μL of 2 M Urea, 50 mM ammonium bicarbonate (ABC) and treated with DL-dithiothreitol (DTT) (final concentration 1 mM) for 30 min at 37 °C with shaking at 1100 rpm on a Thermomixer (Thermo Fisher). Free cysteine residues were alkylated with 2-iodoacetamide (IAA) (final concentration 3.67 mM) for 45 min at 25 °C with shaking at 1100 rpm in the dark. The reaction was quenched using DTT (final concentration 3.67 mM). LysC (750 ng) was added, and the samples were incubated for 1 h at 37 °C with shaking at 1150 rpm. Trypsin (750 ng) was then added, and the digestion mixture was incubated for 16 h at 37 °C with shaking at 1150 rpm. Following this, an additional 500 ng of trypsin was added, and the samples were incubated for another 2 h at 37 °C with shaking at 1150 rpm. The digest was then acidified to pH <3 with 50% trifluoroacetic acid (TFA). Peptides were desalted using C18 stage tips (Empore C18 extraction disks). The stage tips were conditioned with sequential additions of: (i) 100 mL methanol, (ii) 100 mL 70% acetonitrile (ACN)/0.1% TFA, (iii) 100 mL 0.1% TFA twice. After conditioning, the acidified peptide digest was loaded onto the stage tip, followed by two washes with 100 mL 0.1% formic acid (FA). Peptides were eluted with 50 mL 70% ACN/0.1% FA twice. Eluted peptides were dried under vacuum in a SpeedVac centrifuge, reconstituted in 12 μL of 0.1% FA, sonicated and transferred to an autosampler vial. Peptide yield was quantified using a NanoDrop (Thermo Fisher).

For mass spectrometry (MS) analyses, peptides were separated on a 25 cm column with a 75 mm diameter and 1.7 mm particle size, composed of C18 stationary phase (IonOpticks Aurora 3 1801220) using a gradient from 2 to 35% Buffer B over 90 min, followed by an increase to 95% Buffer B for 7 min (Buffer A: 0.1% FA in HPLC-grade water; Buffer B: 99.9% ACN, 0.1% FA) with a flow rate of 300 nL/min on a NanoElute2 system (Bruker). MS data were acquired on a TimsTOF HT (Bruker) with a Captive Spray source (Bruker) using a data-independent acquisition PASEF method (dia-PASEF). The mass range was set from 100 to 1700 m/z, and the ion mobility range from 0.60 V.s/cm$^2$ (collision energy 20 eV) to 1.6 V.s/cm$^2$ (collision energy 59 eV), a ramp time of 100 ms, and an accumulation time of 100 ms. The dia-PASEF settings included a mass range of 400.0 to 1201.0 Da, mobility range 0.60–1.60, and an estimated cycle time of 1.80 s. The dia-PASEF windows were set with a mass width of 26.00 Da, a mass overlap of 1.00 Da, and 32 mass steps per cycle.

For DIA data analysis, raw data files were processed using Spectronaut version 19.1 (Biognosys) and searched with the PULSAR search engine against the *Mycobacterium smegmatis* MC²155 database (12,661 entries downloaded on 2024/11/21). Cysteine carbamidomethylation was set as fixed modifications, while methionine oxidation, protein N-terminal acetylation, and asparagine/glutamine deamidation were specified as variable modifications. A maximum of two trypsin missed cleavages was allowed. A reversed sequence decoy strategy was used to control the peptide false discovery rate (FDR), with a 1% FDR threshold applied for identification. Differential analysis was performed using an unpaired *t*-test to calculate *p* values, and volcano plots were generated based on log2 fold change (log2FC) and *q* value (multiple testing corrected *p* value using Benjamini–Hochberg method). A *q* value of ≤0.05 was considered the statistically significant cut-off.

### Statistical analysis

One-way analysis of variance (ANOVA) and a Dunnett post-test or unpaired *T*-test were performed using Prism9 software (GraphPad) for all statistical analyses of this work. For bipartite split GFP experiments, the statistical analysis was conducted on ln-transformed data.

# Data availability

All numerical data, microscopy images, and Western blot gels used in the figures of this study are provided in the Source Data files. The mass spectrometry proteomics data for the metal-binding assays and proximity labeling have been deposited to the ProteomeXchange Consortium via the PRIDE (Perez-Riverol et al, 2025) partner repository with the dataset identifiers PXD070650 (https://www.ebi.ac.uk/pride/archive/projects/PXD070650) and PXD070766 (https://www.ebi.ac.uk/pride/archive/projects/PXD070766), respectively. All other supporting data are available from the corresponding author upon reasonable request.

The source data of this paper are collected in the following database record: biostudies:S-SCDT-10_1038-S44318-026-00715-1.

# Peer review information

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

## Acknowledgements

We warmly thank Florence Levillain (IPBS, Toulouse) for her training and assistance in the BSL3 facility. We are also grateful to Emmanuelle Näser and Penelope Viana (Génotoul TRI-IPBS platform, Toulouse) for their support with flow cytometry. Additionally, we thank Dr. Pascal Demange (IPBS, Toulouse), Louis Benastre (IPBS, Toulouse), and all members of the Neyrolles lab (IPBS, Toulouse) for their valuable scientific and technical advice. We are particularly grateful to Dr. Yoann Rombouts for proposing the term *effluxosome*. This work was supported by funding from Fondation pour la Recherche Médicale (grant EQU202103012733 to ON), Agence Nationale de la Recherche (ANR-21-CE11-0031 BAC-MMEP to ON, JM, OS, and J-BS and ANR-10-INBS-04 FranceBioImaging to J-BS), ANRS—Maladies infectieuses émergentes (ANRS0697b postdoctoral fellowship to PD) Fondation Bettencourt Schueller (Grant Explore-TB to ON), the European Union (MTB-DETOX: 101063199 Marie Skłodowska-Curie postdoctoral fellowship to PD), and National Institutes of Health (NIH grants P30 AI168433, P30 CA08748, and U19AI135990 to MSG).

## Author contributions

**Pierre Dupuy**: Conceptualization; Data curation; Formal analysis; Supervision; Funding acquisition; Validation; Investigation; Visualization; Methodology; Writing—original draft; Project administration; Writing—review and editing. **Yves-Marie Boudehen**: Formal analysis; Validation; Investigation; Visualization; Methodology. **Marion Faucher**: Formal analysis; Validation; Investigation; Visualization; Methodology. **John A Buglino**: Resources; Formal analysis; Validation; Investigation; Visualization; Methodology. **Allison Fay**: Resources; Formal analysis; Validation; Investigation; Visualization; Methodology. **Sylvain Cantaloube**: Conceptualization; Formal analysis; Validation; Investigation; Visualization; Methodology. **Yasmina Grimoire**: Investigation; Methodology. **Julien Marcoux**: Conceptualization; Formal analysis; Funding acquisition; Validation; Investigation; Visualization; Methodology; Writing—review and editing. **Florian Levet**: Resources; Software. **Laetitia Bettarel**: Investigation. **Bertille Voisin**: Investigation. **Jérôme Rech**: Resources. **Jean-Yves Bouet**: Resources; Writing—review and editing. **Olivier Saurel**: Conceptualization; Formal analysis; Funding acquisition; Validation; Investigation; Visualization; Methodology; Writing—review and editing. **Jean-Baptiste Sibarita**: Conceptualization; Data curation; Software; Formal analysis; Funding acquisition; Visualization; Methodology; Writing—original draft; Writing—review and editing. **Michael Glickman**: Resources; Funding acquisition; Methodology; Writing—review and editing. **Claude Gutierrez**: Conceptualization; Data curation; Formal analysis; Supervision; Funding acquisition; Validation; Investigation; Visualization; Methodology; Writing—original draft; Project administration; Writing—review and editing. **Olivier Neyrolles**: Conceptualization; Supervision; Funding acquisition; Writing—original draft; Project administration; Writing—review and editing.

Source data underlying figure panels in this paper may have individual authorship assigned. Where available, figure panel/source data authorship is listed in the following database record: biostudies:S-SCDT-10_1038-S44318-026-00715-1.

## Disclosure and competing interests statement

MSG has received consulting fees from Vedanta Biosciences, PRL NYC and has equity in Vedanta Biosciences. The remaining authors declare no competing interests.

# Expanded View Figures

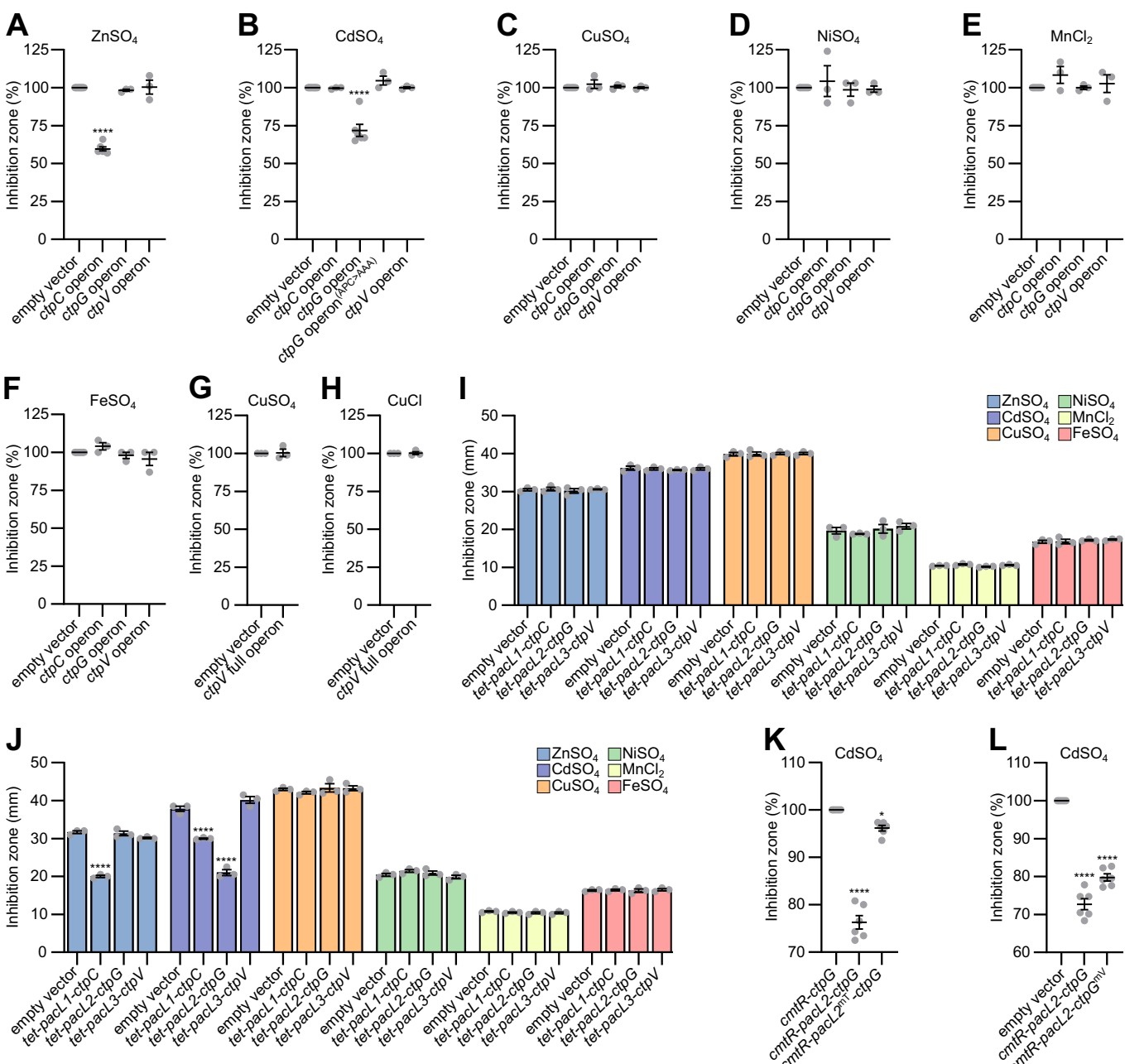

**Figure EV1. Metal sensitivity of *M. smegmatis* strains expressing PacL/Ctp systems.**

(**A–L**) Metal sensitivity of *M. smegmatis* strains carrying a genome-integrative vector expressing the indicated *M. tuberculosis* genes under the control of their native promoter (**A–H, K, L**) or an ATC-inducible promoter (**I**, -ATC; **J**, +ATC), assessed by disk diffusion assay. Inhibition zone diameters, normalized to the reference strain (empty vector (**A–J, L**) or *cmtR-ctpG* (**K**), are shown as mean ± SEM from biological replicates, with individual values indicated by gray dots. For statistical analysis, one-way ANOVA with a Dunnett post-test were performed. Asterisks indicate statistically significant differences compared to the reference strain ($*P < 0.05$; $****P < 0.0001$). Exact $p$ values and biological replicate numbers presented in this figure are reported in Table EV6. ATC anhydrotetracycline, *tet* ATC-inducible promoter; *ctpC* operon: *pacL1-ctpC*; *ctpG* operon: *cmtR-pacL2-ctpG*; *ctpV* operon: *csoR-pacL3-ctpV*; *ctpV* full operon: *csoR-pacL3-ctpV-rv0970*; *pacL2*$^{mT}$, C-terminal translational fusion of PacL2 with mTurquoise; *ctpG*$^{mV}$, C-terminal translational fusion of CtpG with mVenus; (APC > AAA), amino acid substitutions within the conserved APC motif in the P-ATPase domain of CtpG. Source data are available online for this figure.

**A**

```
>Rv3269(PacL1)
MAIQVFLAKATTTVITGLAGVTAYEILKKAAAKAPLRQTAVSAAALGLRGTRKAEEAAESARLKVADVMA
EARERIGEESPTPAISDLHDHDH

>Rv1993c(PacL2)
MVTHELLVKAAGAVLTGLVGVSAYETLRKALGTAPIRRASVTVMEWGLRGTRRAEAAAESARLTVADVVA
EARGRIGEEAPLPAGARVDE

>Rv0968(PacL3)
MVWHGFLAKAVPTVVTGAVGVAAYEALRKMVVKAPLRAATVSVAAWGIRLAREAERKAGESAEQARLMFA
DVLAEASERAGEEVPPLAVAGSDDGHDH
```

**B**

```
PacL1    MAIQVFLAKATTTVITGLAGVTAYEILKKAAAKAPLRQTAVSAAALGLRG
PacL2    MVTHELLVKAAGAVLTGLVGVSAYETLRKALGTAPIRRASVTVMEWGLRG
PacL3    MVWHGFLAKAVPTVVTGAVGVAAYEALRKMVVKAPLRAATVSVAAWGIRL
         :. : :*.**. :*:** .**:*** *:*   .**:* ::*:.   *:*

PacL1    TRKAE----EAAESARLKVADVMAERERIGEESPTPAISDLHD-HDH
PacL2    TRRAE----AAAESARLTVADVVAEARGRIGEEAPLPAGARVDE----
PacL3    AREAERKAGESAEQARLMFADVLAEASERAGEEVPPLAVAGSDDGHDH
         :*.**     :**.*** .***:***  * *** *  * :   .:
```

**Figure EV2.  High conservation of amino acid sequences among the three *M. tuberculosis* PacL proteins.**

(**A**) Amino acid sequences of *M. tuberculosis* PacL1, PacL2, and PacL3 proteins. (**B**) Sequence alignment of *M. tuberculosis* PacL1, PacL2, and PacL3. Predicted transmembrane domains are shown in blue, AE repeats in red, and putative metal-binding motifs in purple.

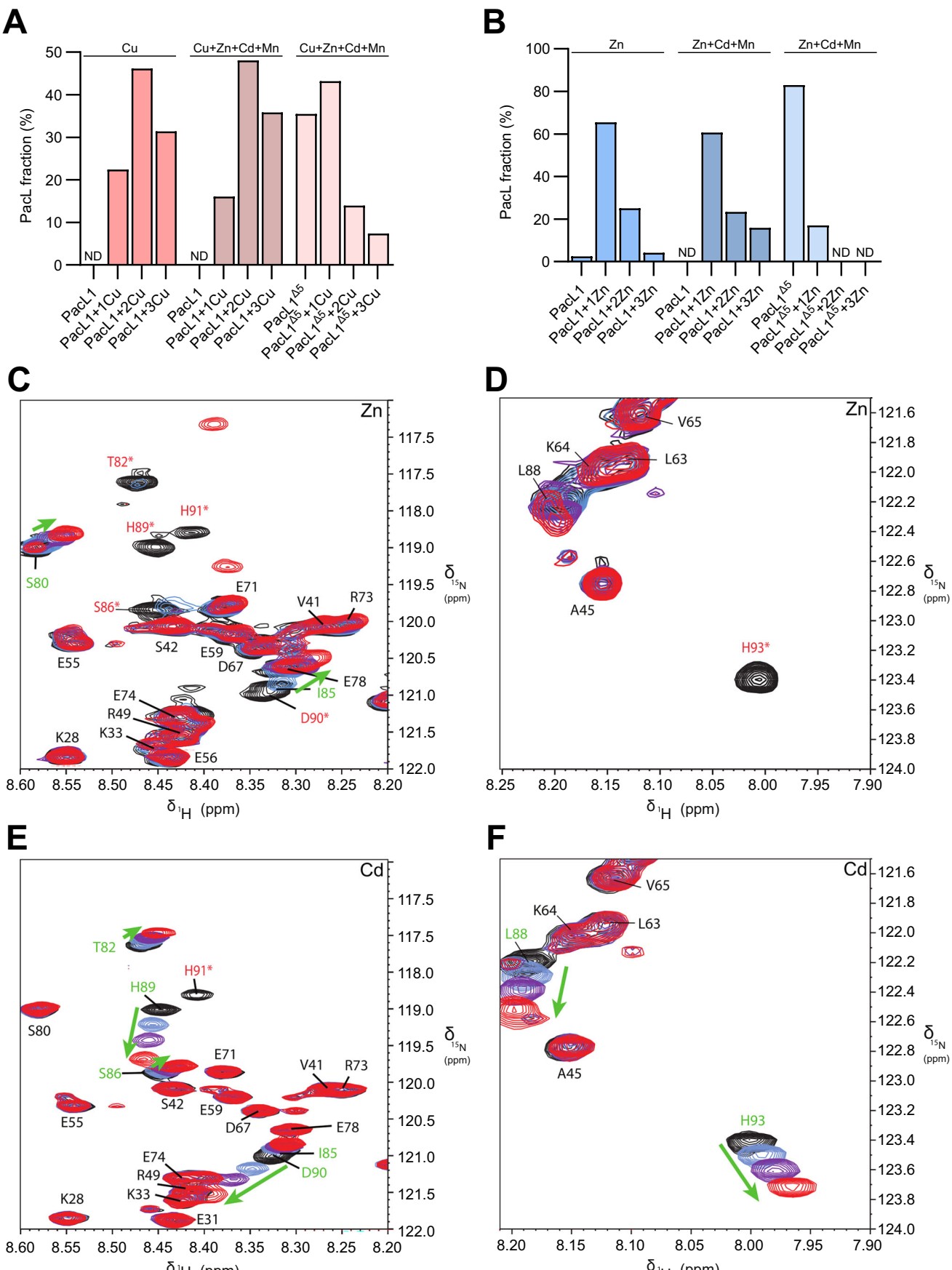

◄ **Figure EV3. PacL1 binds various metal ions mainly through its metal-binding motif, with distinct affinities and binding modes.**

(A, B) Proportion of the indicated purified proteins bound to specific numbers of metal atoms, assessed by native mass spectrometry after incubation with either (A) 500 μM Zn/Cu/Cd/Mn or (B) 500 μM Zn/Cd/Mn. PacL1: soluble domain of PacL1; PacL1Δ5: soluble domain of PacL1 deleted of its C-terminal metal-binding motif. (C–F) $^1$H-$^{15}$N HSQC overlay spectra of 90 μM $^{15}$N-labeled SolPacL1 with 0 (black), 0.4 (light blue), 1.0 (purple), and 2.0 (red) equivalents of zinc, spectra (C, D), and cadmium, spectra (E, F). Peak assignments are directly annotated on the spectra with the following color code: black, no perturbation; green, perturbed residues in fast exchange; red with a star, perturbed residues in intermediate exchange. We observed a distinct binding mode for zinc, which induces significant peak broadening (intermediate chemical exchange) beyond the detection of residues involved in the metal-binding site (H89-H93), compared to cadmium, which remains mainly in fast exchange except for H91. Source data are available online for this figure.

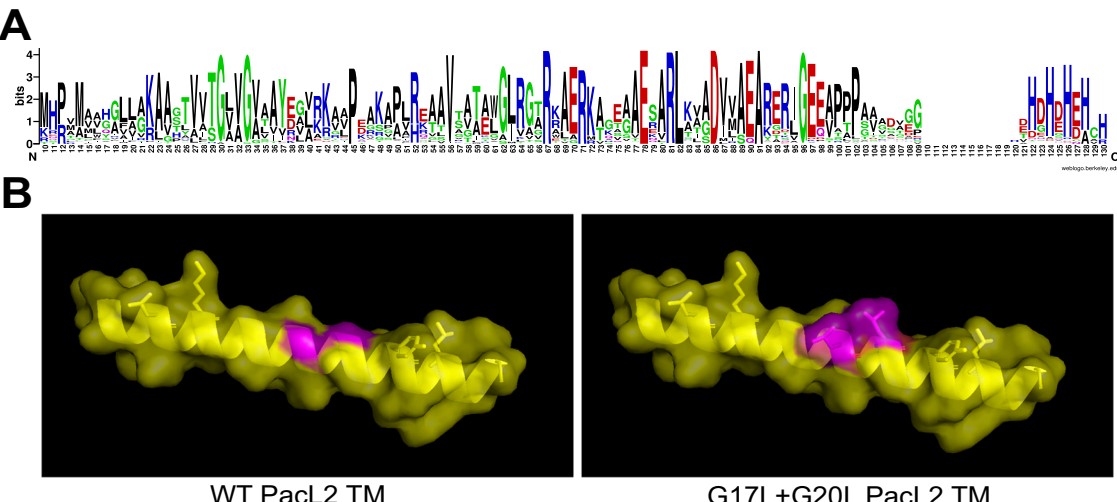

**Figure EV4. Modeling of the conserved GXXG motif in PacL2.**

(**A**) Sequence conservation logo of 120 bacterial PacL-like proteins (DUF1490-containing proteins). (**B**) Predicted structural models of the PacL2 transmembrane domain: (left) wild-type (WT) and (right) G17L + G20L mutant. Amino acids 17 and 20 are shown in magenta.

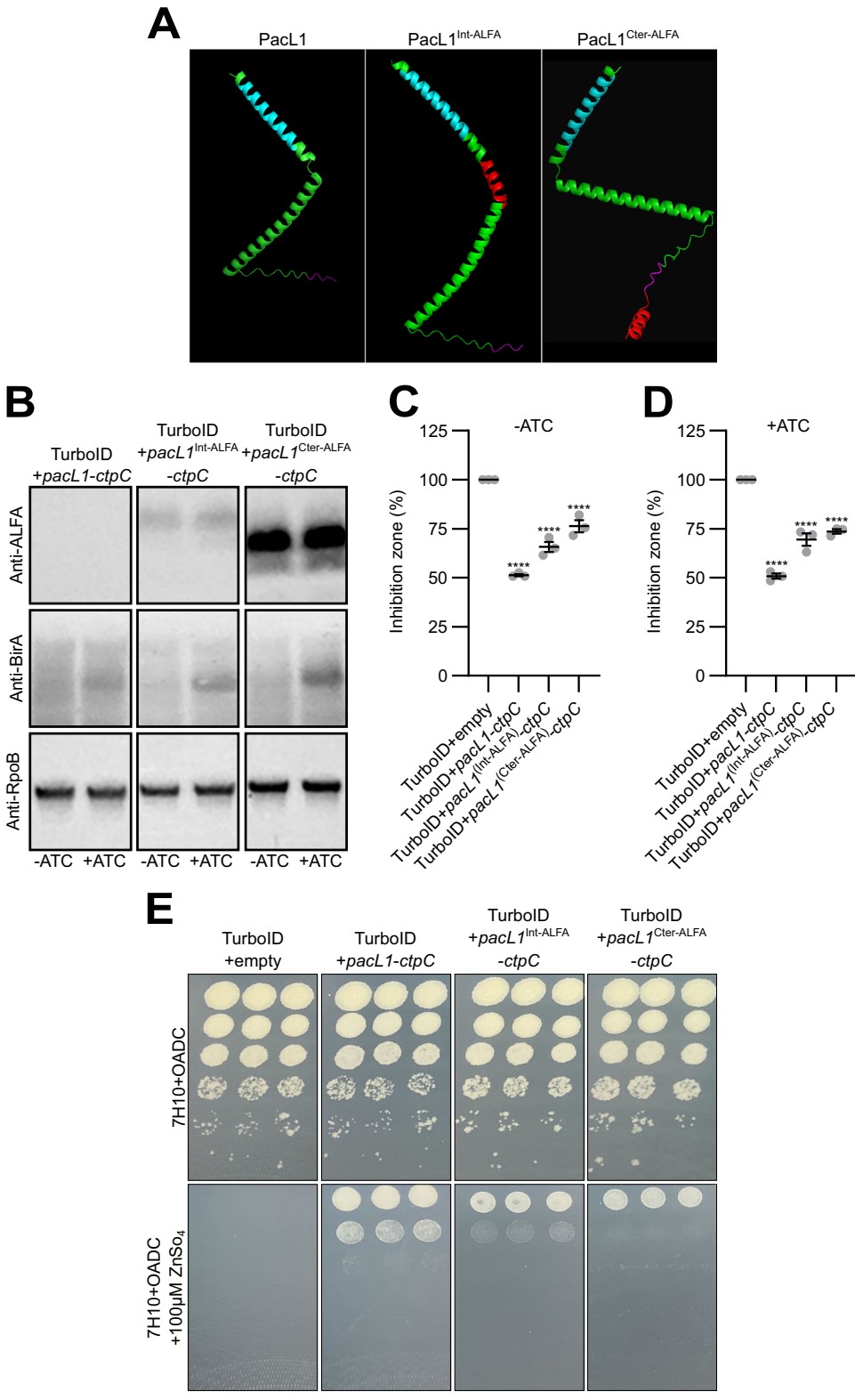

◀  **Figure EV5.  Binding of TurboID to the PacL1<sup>ALFA</sup> proteins does not abolish zinc tolerance.**

(A) AlphaFold structural models of PacL1, PacL1 fused to a C-terminal ALFA tag (PacL1$^{Cter-ALFA}$), and PacL1 fused to an internal ALFA tag (PacL1$^{Int-ALFA}$), with the predicted transmembrane domain shown in blue and the ALFA epitope in red. (B) Anti-ALFA, anti-BirA, and anti-RpoB immunoblots of the indicated *M. tuberculosis* strains harboring the indicated plasmids, cultivated in the absence or presence of the anhydrotetracycline (ATC) inducer. (C, D) Zinc (ZnSO$_4$) sensitivity of *M. smegmatis* strains harboring the indicated plasmids, assessed by disk diffusion assay. The experiment was performed in (C) the absence or (D) the presence of the ATC inducer in the agar medium. Inhibition zone diameters normalized to the empty vector control are shown as mean ± SEM from biological replicates, with individual values indicated by gray dots. For statistical analysis, one-way ANOVA with a Dunnett post-test were performed. Asterisks indicate statistically significant differences compared to the TurboID+empty strain (****$P < 0.0001$). Exact *p* values and biological replicate numbers presented in this figure are reported in Table EV6. (E) Serial dilutions (5 μL) of *M. tuberculosis* cultures harboring the indicated plasmids, spotted on agar plates supplemented or not with 100 μM ZnSO$_4$. Three independent biological replicates are shown per condition. TurboID, replicative plasmid carrying the TurboID-nanobody fusion protein expressed under the control of an ATC-inducible (*tet*) promoter and detected with the anti-BirA antibody; *pacL1-ctpC*, *pacL1*$^{int-ALFA}$-*ctpC*, and *pacL1*$^{Cter-ALFA}$-*ctpC*, genome-integrative plasmids expressing WT PacL1, PacL1 with an internal ALFA tag, or PacL1 with a C-terminal ALFA tag, in operon with *ctpC* under the control of their native promoter. Source data are available online for this figure.

