## [Peer Review File · The EMBO Journal]

Membrane-associated effluxosomes coordinate multi-metal resistance in *Mycobacterium tuberculosis*

Pierre Dupuy, Yves-Marie Boudehen, Marion Faucher, John Buglino, Allison Fay, Sylvain Cantaloube, Yasmina Grimoire, Julien Marcoux, Florian Levet, Laetitia Bettarel, Bertille Voisin, Jérôme Rech, Jean-Yves Bouet, Olivier Saurel, Jean-Baptiste Sibarita, Michael Glickman, Claude Gutierrez, and Olivier Neyrolles

Corresponding authors: Olivier Neyrolles (olivier.neyrolles@ipbs.fr) , Pierre Dupuy (Pierre.Dupuy@ipbs.fr)

Review Timeline:

Submission Date:	16th May 25
Editorial Decision:	18th Jun 25
Revision Received:	16th Nov 25
Editorial Decision:	11th Dec 25
Revision Received:	8th Jan 26
Accepted:	23rd Jan 26

Editor: Ieva Gailite

Transaction Report:

Dear Dr. Neyrolles,

Thank you for submitting your manuscript for consideration by the EMBO Journal. We have now received comments from a full set of reviewers, which are included below for your information.

As you will see, reviewers #1 and #2 (expertise in bacterial metal homeostasis and transport) are generally positive in their assessment and suggest mainly minor revisions. At the same time, reviewer #3 with expertise in bacterial super-resolution microscopy, raises significant concerns with the experimental approach and data analysis that would have to be addressed to support the proposed conclusions. In the cross-commenting session, reviewer #1 agreed with the technical concerns raised by reviewer #3, and the point regarding the functionality and the localisation of the tagged proteins is also raised in the report by reviewer #2.

Based on the interest in the study expressed by reviewers #1 and #2, I invite you to submit a revised manuscript in which you address the reviewers' concerns along the lines outlined above. I think that it would be useful to discuss the revision in more detail via email or phone/videoconferencing - please let me know which option you prefer.

We generally allow three months as standard revision time, which can be extended to six months in the case of major revisions. Should you foresee a problem in meeting this deadline, please let us know in advance to discuss an extension.

As a matter of policy, competing manuscripts published during this period will not negatively impact on our assessment of the conceptual advance presented by your study. However, please contact me as soon as possible upon publication of any related work to discuss the appropriate course of action.

When preparing your letter of response to the referees' comments, please bear in mind that this will form part of the Review Process File and will therefore be available online to the community. For more details on our Transparent Editorial Process, please visit our website: <https://www.embopress.org/page/journal/14602075/authorguide#transparentprocess>. Please also see the attached instructions for further guidelines on preparation of the revised manuscript.

Please feel free to contact me if you have any further questions regarding the revision. Thank you for the opportunity to consider your work for publication. I look forward to discussing your revision.

With best regards,

Ieva

Ieva Gailite, PhD
Senior Scientific Editor
The EMBO Journal
Meyerohofstrasse 1
D-69117 Heidelberg
Tel: +4962218891309
i.gailite@embojournal.org

We realize that it is difficult to revise to a specific deadline. In the interest of protecting the conceptual advance provided by the work, we recommend a revision within 3 months (16th Sep 2025). Please discuss the revision progress ahead of this time with the editor if you require more time to complete the revisions.

Referee #1:

Dupuy, Neyrolles and colleagues report on the local organization of three metal-exporting P-type ATPases in mycobacteria, which are orchestrated by so-called PacL protein (PacL1-3). The study builds on a previous study authored by the same group three years back, which showed that the zinc efflux pump CtpC requires co-expression of the operon partner PacL1 for functioning. In this paper, the authors employ a combination of fluorescence microscopy, PALM microscopy allowing for single particle tracking and functional assays using *Mycobacterium smegmatis* and *Mycobacterium tuberculosis* as model systems to shed further light on the molecular mechanisms of metal efflux in mycobacteria. Thereby, they could show that i) PacL1-CtpC as well PacL2-CtpG confer cadmium resistance, ii) CtpC, CtpG, and CtpV with the help of their chaperones PacL1, PacL2 and PacL3 co-localize within both in *Mycobacterium smegmatis* and *Mycobacterium tuberculosis* cells and thereby form patches on the membrane, iii) these co-localizations are required for metal efflux function, as demonstrated by mutations in the PacL proteins that disturb co-localization. The authors could not confirm previous data by other groups concerning copper efflux by CtpV (in their hands, they did not find a metal substrate for CtpV). Furthermore, they could not confirm efflux of Cu, Ni, Fe and Mn by CtpG, as has been reported by a previous paper. In a final set of experiments, the authors used TurboID to identify additional interaction partners that co-localize with PacL and/or P-type ATPase proteins. The authors conclude that local organization of CtpC, CtpG, and CtpV in effluxosomes is required for establishing for metal efflux function.

This is an extensive and interesting study, which makes the surprising discovery that *M. tuberculosis* expresses P-type ATPase that confer cadmium resistance. Further, the paper convincingly establishes that PacL proteins help to assemble an entire set of metal-effluxing P-type ATPases in small, slowly mobile clusters at the cytosolic membrane and that such localization in effluxosomes is required for metal export function. The paper also contains a thorough functional analysis of the PacL proteins and the respective P-type ATPases, involving both *Mycobacterium smegmatis* and *Mycobacterium tuberculosis*. Given the complexity (and time span) to work on *Mycobacterium tuberculosis*, this is an impressive achievement. I also liked the fact that the authors did not shy away from pointing out findings that are in conflict with previous works from other labs, as I consider such negative results as equally important than positive ones. The paper acknowledges the complexity of the matter, and provides the necessary experimental details. Overall, I consider the findings to be of broad significance and this works paves the way to better understand mycobacterial transporter biology in the context of the living cell.

Major points

- 1) The authors nicely show by mass spectrometry and NMR that PacL1 binds zinc and other metals and that it happens via a C-terminal metal binding motif. The also show (line 276) that deletion of the last 5 amino acids strongly reduces metal binding (PacL1 Δ 5 construct). But what about localization experiments and functional experiments (zinc and cadmium resistance)? Did the authors test this in this or a previous paper? Providing a good overview on the full picture of the PacL1 Δ 5 construct would be very helpful, even if it does not play a functional role.
- 2) The authors performed a TurboID experiment to identify further potential proteins that co-localize with the effluxosome. To this end, they introduced a ALFA-tag into PacL1 at two positions (internally and at the C-terminus). While they show that ALFA-tagged PacL1 still functions, they do not formally show (at least if I understand the paper correctly) whether the ALFA-Nb would

recognize the ALFA tags at the respective positions. Does the ALFA-Nb fold properly in the cytosol, since it contains a disulfide bond? Further, it would be helpful to see the internal ALFA-tag position by predicting the tagged (versus untagged) versions by AlphaFold.

3) Still regarding the TurboID experiment, it is unclear where the ALFA-tag would be located in the control protein MalF. It is assumed to be in the cytosol; pls clarify. It is surprising to see that the proteins identified by TurboID to interact with PacL1 include PPE20, which in fact is expected to end up in the outer mycobacterial cell wall. The authors should go through the entire list of proteins they identified (Table 1) and report their expected cellular localization. In principle, only cytosolic and inner membrane proteins should be identified in this approach.

Minor points

- 1) Line 138 (and also elsewhere): did the authors perform some Western blotting (or similar) to show whether CtpG is less produced when expressed alone versus together with its operon partner PacL2? I appreciate that there are fluorescence microscopy images suggesting this, but an orthogonal test could be helpful. The operon context may have a strong effect on expression/translation of the transporter gene.
- 2) Line 406: the authors suggest that "clustering of P-ATPase pumps by PacL proteins shields them from degradation". Is there any experimental proof this this, maybe in another paper on protein clusters?
- 3) Line 410: PacL1 is the only mycobacterial PacL protein capable of metal binding. But it is curious that it binds all kind of metals, whereas the co-localized P-type ATPases only seem to transport zinc and cadmium. Do the authors have any explanation why metal binding by PacL1 does not appear to match the specificities of the transporters? Does PacL1 have preferences for some metals (that could be tested with mass spectrometry when incubating PacL1 with a mix of metals)?
- 4) Line 429: the authors mention the existence of "metal hot spots". Is there experimental evidence for such hotspots to exist in bacteria in general or mycobacteria in particular?
- 5) Figure 8: consider drawing the PacL proteins and P-type ATPases to scale (e.g. by using AlphaFold models) and/or as a minimum indicate where on the P-type ATPases the metal binding domains would be located (because they seem to interact with the PacL proteins via their EA motifs, as the authors explain...).
- 6) Figure 8, last sentence of figure legend; Author statement: "PacL1, but not PacL2, binds a variety of metal ions, thereby enhancing the metal tolerance conferred by the P-ATPases."
It is unclear whether the authors have provided experimental evidence for their notion that metal binding by PacL1 enhances metal tolerance (see also main comment above).

Referee #2:

In the manuscript, Neyrolles and collaborators present a significant follow-up study to what I consider a groundbreaking paper published in 2022 (ref 5). That paper established that small, single-pass transmembrane proteins, termed PacL1, PacL2 and PacL3 play important roles in maintaining the functionality of one of more metal-effluxing P-type ATPases. This follow-up work presented here is stunning in every way. The authors provide strong experimental support for the existence of what they term effluxosomes, that may be broadly distributed among a wide range of bacteria.

This manuscript, while somewhat dense, is meticulously presented, beautifully organized and a pleasure to read. The supplemental figures are clear. All the controls are here, particularly those that speak to the functionality of variously tagged proteins, and all of the major findings are well supported by the data. The results from a variety of fluorescence imaging experiments anchor the work and include very high-quality single molecule tracking of labeled PacL proteins relative labeled P-type ATPases. New insights are provided as to how PacL1 interacts with metals, while bipartite-split GFP approaches are used to establish if two proteins or domains interact in cells, and if so, how. PacL proteins clearly interact in the membrane, both with themselves as well as other PacL orthologs. Finally, the results of two complementary PacL1 proximity labeling experiments are presented and are conservatively discussed; they suggest other currently unknown players that may enhance or diversify effluxosome function in cells.

I only have a number of minor comments:

- 1) Although perhaps not central to this manuscript, the lack of a clearly defined function of CtpV largely on the basis of disc diffusion assays (implicated in protection against Cu toxicity for many years) is rather puzzling. The introduction of Cu(I) or Cu(II) to these assays may have resulted in low bioavailability to cells; alternatively, there is another Cu resistance determinant that is not being considered or has escaped detection. Is the *ricR* regulon induced under these conditions? Is this system constitutively on for some reason? Both would complicate interpretation of CtpV function from these assays
- 2) The authors detect three diffusive populations of PacL2 (Fig. 3) and claim that the "immobile" fraction is inactive (line 419). What is the evidence for this? Perhaps tone this conjecture down a bit.
- 3) Fig. 5(a-f): I'm not a big fan of using native mass spectrometry to define metal binding characteristics of a protein or peptide, given the very real problem of dissociation of weakly stable (as evidenced by NMR) metal complexes upon ejection into the gas phase. A negative result (as with Mn) doesn't really mean anything. I suggest addition of a sentence that makes the

shortcomings of this experiment clear to the general reader.

4) Fig. 4a- the "negative control" does not return to the control level of fluorescence. Please explain. I worry about the impact of non-physiological expression on this assay.

5) There are no references on transition metal homeostasis (lines 37-40). I suggest this one, which specifically discusses PacL proteins as potential metallochaperones: PMC11672702.

6) What is known about the lipid composition of these microdomains? Do the authors expect those lipids to differ from bulk lipids in the membrane? Perhaps this can be discussed in the Discussion section.

7) Minor typos, comments:

a) line 154, change A59A to E59A

b) line 156, change Fig. Fig. to Fig.

c) line 379, please define PE/PPE for the general reader

d) line 440, change constantly to consistently

e) line 492, replace "interacting with" to "in proximity to"

Referee #3:

Dupuy et al propose a model in which small membrane-associated proteins of the PacL family mediate the clustering of P-type ATPase pumps that lead to the efflux of different metals, thereby providing resistance to high concentrations of metals such as cadmium and zinc.

I have several reservations about this study concerning the cell biological aspects.

A) Line 144 „To determine the subcellular localization of PacL2 and CtpG, we engineered PacL2-mTurquoise and CtpG-mVenus fusion proteins". I was unable to find a description of how the strains were constructed. Were fluorescent protein (FP) gene fusions integrated at the original gene locus, or expressed as a merodiploid? Which promoter drove the expression of fusion genes? If original gene locus, are there downstream genes/operon structures?

B) Fig. 2d and most of the following epifluorescence images show strong fluorescent foci/patches at the cell poles. This may be the true localization of the proteins, or the result of protein overexpression or other aspects like the formation of protein aggregates. Therefore, it is important to control that

a) proteins are expressed as full length proteins, and not as truncations, and

b) that they are expressed at physiological levels. Western blotting is necessary to rule out any artifacts causing mislocalization of protein fusions.

C) I am also missing experiments showing that the FP fusions functionally replace the wild type proteins. Strains must be assayed for sensitivity to metals used in this study to prove their functionality.

D) It is claimed in Fig. 2 that fusions localize to the cell membrane. This is not evident to my eye. Please provide membrane staining and higher magnification images that clearly show foci are in the membrane and not in the cytosol.

E) Fig. 3 How were the cells imaged? To my eye, it looks as if the cells were imaged with the focal plane being at the cell surface rather than at the central plane, otherwise the foci would all be at the periphery of the cells. Not that molecules moving along the short axis of the cell are underestimated for their displacement because of the curvature of the cell. In any event, please state how imaging was performed in terms of the focal plane for Fig. 2 and 3.

F) Line 592 "When necessary, mycobacterial cells were fixed with 4% paraformaldehyde (PFA) treatment (2 hours for *M. tuberculosis* and 20 minutes for *M. smegmatis*), followed by three PBS washing steps." Please state which experiments were done using fixed cells.

G) Line 656 "We used a threshold of 18.." 18 what? Please state the units or parameters that the number refers to.

H) Line 188 "To further characterize and map mobile and immobile molecules, we applied an experimental threshold of $0.03 \mu\text{m}^2\cdot\text{s}^{-1}$ (Fig. 3g). Our analysis revealed that 70% of the molecules were immobile, while 30% were classified as mobile." It is not appropriate to distinguish mobile from immobile molecules based on MSD analyses. First of all, it is unclear why the authors come up with the threshold, this seems arbitrary. At the least, they must provide a convincing reason for this number. Secondly, Fig. 3g does not show an MSD plot, but a log plot of numbers of diffusion constants, i.e. trajectories with a certain MSD. Please show a proper MSD plot to evaluate the quality of the data. If the authors want to investigate if different subpopulations exist,

e.g. static and mobile molecules, they should use squared displacement analyses, or another appropriate evaluation. E.g. TARDIS from the Endesfelder lab <https://doi.org/10.1038/s41592-023-02149-7>

l) Fig. 6D where is fluorescence gone in the truncation mutants? Please provide a Western showing expression levels of the truncation constructs.

Finally, I have a bit of a conceptual problem: why would a single-celled organism cluster efflux pumps? In order to efficiently move toxic compounds across the cell membrane, isn't it more sensible to distribute all exporters relatively evenly across the membrane surface in order to decrease the time transporters interact with their substrates? I find it counterintuitive to generate large clusters that compete for substrate binding at few sites in the cell. Even if PacL proteins bind to metals, would this really increase the local concentration of a metal? Hard to believe. I am also puzzled by the statement in line 405: Here, our findings indicate that the clustering of P-ATPase pumps by PacL proteins not only shields them from degradation but is also essential for their functionality." I was unable to find any Western blot in this study, so how can the authors make claims about stability?

Response to referees (Dupuy *et al.*, EMBO J)

Referee #1

We thank the referee for their thorough and positive evaluation of our work and for highlighting the significance of our findings. We are grateful for their appreciation of both the scope and rigor of our study, including our efforts to address discrepancies with previous reports. The encouraging comments are deeply appreciated by all co-authors. Below are our point-by-point responses to the reviewer's comments.

Major points

1) The authors nicely show by mass spectrometry and NMR that PacL1 binds zinc and other metals and that it happens via a C-terminal metal binding motif. They also show (line 276) that deletion of the last 5 amino acids strongly reduces metal binding (PacL1 Δ 5 construct). But what about localization experiments and functional experiments (zinc and cadmium resistance)? Did the authors test this in this or a previous paper? Providing a good overview on the full picture of the PacL1 Δ 5 construct would be very helpful, even if it does not play a functional role.

In our previous study (Boudehen *et al.*, 2022), we demonstrated that deletion of the C-terminal metal-binding motif of PacL1 impairs CtpC-dependent zinc tolerance for high concentrations of zinc (see graph). This finding is reiterated in the current manuscript:

- “PacL1 contains a C-terminal metal-binding motif, D⁸⁷LHDHDH⁹³ (Fig. EV2) that binds zinc at a 1:1 molar ratio and is crucial for resistance to high zinc concentrations in *M. tuberculosis*, supporting its role as a metallochaperone (Boudehen *et al.*, 2022)”

To better highlight this result in our manuscript, we added the following sentence in the Discussion:

- “Our previous study demonstrated that zinc binding by PacL1 enhances the ability of CtpC to confer metal tolerance (Boudehen *et al.*, 2022).”

However, we did not investigate whether deletion of this motif affects PacL1 localization. Given that PacL2 lacks this motif but still forms clusters similar to PacL1, we hypothesize that the motif is not required for PacL1 localization.

2) The authors performed a TurboID experiment to identify further potential proteins that co-localize with the effluxosome. To this end, they introduced an ALFA-tag into PacL1 at two positions (internally and at the C-terminus). While they show that ALFA-tagged PacL1 still functions, they do not formally show (at least if I understand the paper correctly) whether the ALFA-Nb would recognize the ALFA tags at the respective positions. Does the ALFA-Nb fold properly in the cytosol, since it contains a disulfide bond? Further, it would be helpful to see

the internal ALFA-tag position by predicting the tagged (versus untagged) versions by AlphaFold.

We acknowledge Referee #1's concern regarding the proper folding of the nanobody in the cytosol. While a formal experimental validation would be challenging, we believe that the strong enrichment of PacL1 as the most biotinylated protein provides compelling evidence that the nanobody binds to ALFA-tagged PacL1. As suggested, we have included AlphaFold structure predictions of both the wild-type and ALFA-tagged PacL1 in Figure EV5A and added the following sentence in the main text:

- “AlphaFold-predicted structures of PacL1^{Cter-ALFA} and PacL1^{Int-ALFA} indicated that the overall protein conformation may not be significantly affected by the addition of the ALFA epitope (Fig. EV5A)”.

3) Still regarding the TurboID experiment, it is unclear where the ALFA-tag would be located in the control protein MalF. It is assumed to be in the cytosol; pls clarify. It is surprising to see that the proteins identified by TurboID to interact with PacL1 include PPE20, which in fact is expected to end up in the outer mycobacterial cell wall. The authors should go through the entire list of proteins they identified (Table 1) and report their expected cellular localization. In principle, only cytosolic and inner membrane proteins should be identified in this approach.

The ALFA tag is located at the C-terminus of the first two transmembrane domains of *E. coli* MalF. In the topology of *E. coli* MalF, the ALFA tag is cytosolic but remains physically associated with the membrane. We now clarify this in the main text:

- “Biotinylated proteins were identified by proteomics and compared to those detected under control conditions, in which the ALFA epitope was fused to the first two transmembrane domains of the *E. coli* MalF protein (MalF₍₁₋₂₎-ALFA) and co-expressed with the TurboID–nanobody fusion. This provides a baseline for non-specific interactions within the membrane. When co-expressed with a nanobody–GFP fusion, MalF₍₁₋₂₎-ALFA displays uniform fluorescence along the mycobacterial membrane (Fay *et al.*, 2025), consistent with the ALFA epitope being cytosolic but membrane-anchored.”

We agree with Referee #1 that the detection of outer cell wall proteins is unexpected. However, we hypothesize that a small fraction of these proteins may localize to the cytosol or

inner membrane. Alternatively, since proximity labeling can capture transient interactions, these outer membrane proteins might be labeled during their translocation through the inner membrane on their way to the outer cell wall. As asked, we now provide the predicted subcellular localization of all proteins listed in new proximity labeling tables (Table 1 and supplemental table S4).

Minor points

1) Line 138 (and also elsewhere): did the authors perform some Western blotting (or similar) to show whether CtpG is less produced when expressed alone versus together with its operon partner PacL2? I appreciate that there are fluorescence microscopy images suggesting this, but an orthogonal test could be helpful. The operon context may have a strong effect on expression/translation of the transporter gene.

We agree with the Referee's comment. We have now added anti-GFP western blot showing the CtpG^{mV} protein is destabilized in both the absence of PacL2 or when PacL2^{mT} carries the 3EA mutation impacting the PacL2/CtpG interaction. The following figures and sentences were added to the main text (PacL2^{mT}: 36.5 kDa; CtpG^{mV}: 106.5 kDa):

- “Consistent with this observation, western blot analyses detected CtpG protein in bacterial lysates co-expressing PacL2, but not in those expressing CtpG alone (**Fig. 2I**)”
- “In addition, although the E59A or E71A mutants did not prevent the formation of PacL2-mTurquoise clusters, they resulted in destabilization of CtpG-mVenus (**Fig. 2K-M**) and the triple substitution E55A, E59A, and E71A completely abolished the presence of CtpG in bacterial lysates, as determined by western blot analysis (**Fig. 2N**).”

2) Line 406: the authors suggest that "clustering of P-ATPase pumps by PacL proteins shields them from degradation". Is there any experimental proof this this, maybe in another paper on protein clusters?

This sentence refers to the observation that, in the absence of PacL proteins, CtpC and CtpG are degraded. This phenomenon was reported both in the present study and in our previous work (Boudehen *et al*, 2022), based on epifluorescence microscopy and new western blot analyses. We agree that the term “shield” may be misleading and have therefore revised the sentence as follows:

- “Here, our findings indicate that the clustering of P-ATPase pumps by PacL proteins not only prevents their degradation but is also essential for their functionality.”

3) Line 410: PacL1 is the only mycobacterial PacL protein capable of metal binding. But it is curious that it binds all kind of metals, whereas the co-localized P-type ATPases only seem to transport zinc and cadmium. Do the authors have any explanation why metal binding by PacL1 does not appear to match the specificities of the transporters? Does PacL1 have

preferences for some metals (that could be tested with mass spectrometry when incubating PacL1 with a mix of metals)?

Although we were unable to detect activity for CtpV, this protein has been reported to function as a copper transporter, which could explain why PacL1 also binds copper. In addition, our proximity-labeling experiments indicate that several other proteins may be part of the effluxosome, potentially including transporters for other metals. We thank the Referee for their insightful suggestion to perform competition assays using mixtures of Zn, Cd, Cu, and Mn. We have now conducted these experiments and observed that PacL1 displays distinct affinities for different metals, following the order Cu > Zn > Cd > Mn. These results have been incorporated into the revised manuscript (Figs. S3E–F) and the following sentences have been added:

- "In order to assess the metal-binding affinity of PacL1 and account for potential non-specific binding or gas-phase dissociation, competition assays were performed in ammonium acetate supplemented with either Cd/Zn/Cu/Mn or Cd/Zn/Mn. We observed that PacL1 bound exclusively to copper when incubated with Cd/Zn/Cu/Mn (**Fig. EV3A**) and exclusively to zinc when incubated with Cd/Zn/Mn (**Fig. EV3B**), indicating that PacL1 preferentially binds Cu, followed by Zn, and then Cd."

- “Notably, while PacL1 predominantly bound a single zinc or cadmium atom (**Fig. 5C, D**), species containing one, two, or three copper atoms were detected (**Fig. 5E**)”
- “Notably, PacL1^{Δ5} was still able to bind one copper atom, suggesting that two copper atoms are coordinated through its canonical motif, while an additional atom may be bound at an alternative, unidentified site.”

4) Line 429: the authors mention the existence of "metal hot spots". Is there experimental evidence for such hotspots to exist in bacteria in general or mycobacteria in particular?

Our wording was intended to be conceptual, to illustrate the possibility that local variations in metal ion concentration could promote the recruitment or assembly of metal-handling machineries. While direct imaging of subcellular metal gradients in mycobacteria has not yet been achieved to our knowledge, heterogenous distribution of zinc within *E. coli* cells was reported (DOI: [10.1038/s41598-018-31461-y](https://doi.org/10.1038/s41598-018-31461-y)). We added the reference in the main text.

5) Figure 8: consider drawing the PacL proteins and P-type ATPases to scale (e.g. by using AlphaFold models) and/or as a minimum indicate where on the P-type ATPases the metal binding domains would be located (because they seem to interact with the PacL proteins via their EA motifs, as the authors explain...).

We appreciate the Referee's suggestion. However, Figure 8 is intended as a simplified, schematic model summarizing our working hypothesis rather than a structural representation. Including AlphaFold-based representations or domain annotations would risk overinterpreting currently unavailable structural data and would make the figure less clear and readable. We therefore prefer to retain the schematic format as it stands.

6) Figure 8, last sentence of figure legend; Author statement: "PacL1, but not PacL2, binds a variety of metal ions, thereby enhancing the metal tolerance conferred by the P-ATPases." It is unclear whether the authors have provided experimental evidence for their notion that metal binding by PacL1 enhances metal tolerance (see also main comment above).

As mentioned above, our previous study (Boudehen *et al*, 2022) demonstrated that deletion of the C-terminal metal-binding motif of PacL1 impairs CtpC-dependent zinc tolerance. To better highlight this result in our manuscript, we added the following sentence in the Discussion:

- "Our previous study demonstrated that zinc binding by PacL1 enhances the ability of CtpC to confer metal tolerance."

Referee #2

We sincerely thank the referee for their exceptionally positive and encouraging evaluation of our work. We are delighted that they found the study well organized, clearly presented, and technically sound. Their appreciation of our imaging and interaction analyses, as well as the broader implications for effluxosome function, is greatly valued by all co-authors. Below are our point-by-point responses to the reviewer's comments.

1) Although perhaps not central to this manuscript, the lack of a clearly defined function of CtpV largely on the basis of disc diffusion assays (implicated in protection against Cu toxicity for many years) is rather puzzling. The introduction of Cu(I) or Cu(II) to these assays may have resulted in low bioavailability to cells; alternatively, there is another Cu resistance determinant that is not being considered or has escaped detection. Is the *ricR* regulon induced under these conditions? Is this system constitutively on for some reason? Both would complicate interpretation of CtpV function from these assays

We agree with the Referee that the absence of CtpV-related phenotypes is intriguing, and that the proposed hypotheses could explain the lack of phenotypes under our experimental conditions. We acknowledge that metal bioavailability may be limited in agar-based assays. However, the presence of a clear zone of growth inhibition around copper discs indicates that sufficient bioavailable copper reaches the bacteria to exert toxicity. We also agree that additional copper-resistance determinants may contribute to the observed phenotype. Measuring *ricR* expression in agar-based assays is technically difficult, but it is reasonable to expect that the *ricR* regulon is induced under these conditions. To address this point, we have slightly revised the Discussion to include the possibility that functional redundancy with other copper detoxification pathways could obscure the contribution of CtpV:

- “Although no increased copper sensitivity was detected in an *M. tuberculosis* triple *pacL* mutant (data not shown), it remains possible that CtpV depends on an unidentified partner encoded in the *M. tuberculosis* genome, but absent from *M. smegmatis*, for its function, or that functional redundancy with one or more copper detoxification pathways mask its contribution.”

2) The authors detect three diffusive populations of PacL2 (Fig. 3) and claim that the "immobile" fraction is inactive (line 419). What is the evidence for this? Perhaps tone this conjecture down a bit.

We thank the Referee for this comment. The rationale for this statement is detailed in the Discussion:

- “First, we identified a population of confined molecules forming large, stable clusters predominantly localized at the cell poles. These clusters persist even when mutations are introduced at positions G17, G20, and K9 in PacL2, amino acids essential for

effluxosome-mediated metal tolerance, suggesting that these assemblies are non-functional aggregates.”

To avoid overinterpretation, we have slightly rephrased the sentence in the main text to temper our conclusion and clarify that this interpretation is based on indirect evidence:

- “This immobile fraction may represent non-functional assemblies, as similar clusters persist even when essential residues for effluxosome-mediated metal tolerance (G17, G20, K9) are mutated.”

3) Fig. 5(a-f): I'm not a big fan of using native mass spectrometry to define metal binding characteristics of a protein or peptide, given the very real problem of dissociation of weakly stable (as evidenced by NMR) metal complexes upon ejection into the gas phase. A negative result (as with Mn) doesn't really mean anything. I suggest addition of a sentence that makes the shortcomings of this experiment clear to the general reader.

We agree that gas-phase dissociation of weakly bound metals is a potential limitation of native mass spectrometry. To address this concern, and also in response to Referee #1's comment, we performed competition assays using mixtures of Zn, Cd, Cu, and Mn. These assays partially account for possible gas-phase dissociation effects, since if Mn had a higher affinity than Zn or Cu but dissociated upon ionization, no binding would be detected, yet this was not the case.

We have now added figures **EV3A** and **B** and modified the text to explicitly mention this potential limitation and the results of our new competition assays:

- "In order to assess the metal-binding affinity of PacL1 and account for potential non-specific binding or gas-phase dissociation, competition assays were performed in ammonium acetate supplemented with either Cd/Zn/Cu/Mn or Cd/Zn/Mn. We observed that PacL1 bound exclusively to copper when incubated with Cd/Zn/Cu/Mn (**Fig. EV3A**) and exclusively to zinc when incubated with Cd/Zn/Mn (**Fig. EV3B**), indicating that PacL1 preferentially binds Cu, followed by Zn, and then Cd.”

4) Fig. 4a- the "negative control" does not return to the control level of fluorescence. Please explain. I worry about the impact of non-physiological expression on this assay.

We agree that split-GFP assays are technically challenging and that their results should be interpreted with caution. However, our main conclusions regarding protein interactions are also supported by colocalization analyses using epifluorescence microscopy. The baseline fluorescence control corresponds to *M. smegmatis* expressing a PacL1–GFP11 fusion together with a cytoplasmic GFP1–10 fragment. In line with the Referee’s concern about non-physiological expression levels in this assay, we included an additional control using Rv1488, a flotillin-like protein known to form membrane clusters distinct from PacL-dependent platforms (Boudehen *et al*, 2022). This membrane-associated control has a higher likelihood of random encounters with PacL proteins at the membrane, which explains why the fluorescence signal is higher than that observed with the cytoplasmic GFP1–10 negative control. We modified the paragraph as followed to improve clarity:

- “As a control for potential non-specific membrane interactions, we tested the association between PacL1 and the flotillin-like protein Rv1488, which forms membrane clusters independent of PacL-dependent platforms⁵. As expected, this membrane-associated control has a higher likelihood of random encounters with PacL proteins, resulting in a stronger fluorescence signal than that observed with the cytoplasmic GFP1–10 negative control. However, we observed a four-fold lower GFP signal in cells expressing the Rv1488/PacL1 pair compared to those expressing two PacL proteins, indicating that interactions between PacL proteins are specific rather than the result of random membrane proximity.”

5) There are no references on transition metal homeostasis (lines 37-40). I suggest this one, which specifically discusses PacL proteins as potential metallochaperones: PMC11672702.

The reference was added.

6) What is known about the lipid composition of these microdomains? Do the authors expect those lipids to differ from bulk lipids in the membrane? Perhaps this can be discussed in the Discussion section.

The lipid composition of effluxosomes remains unknown. We agree with the Referee that this is a fascinating and important question that will be addressed in future studies. To highlight this point, we have added the following sentences to the Discussion:

- “The lipid composition of effluxosomes and the potential influence of specific membrane lipids on their dynamics and metal efflux activity remain unknown. Future studies will address these questions, e.g., using the recently developed styrene–maleic acid anhydride lipid particle (SMALP) technology.”

Minor comments:

a) line 154, change A59A to E59A

Done

b) line 156, change Fig. Fig. to Fig.

Done

c) line 379, please define PE/PPE for the general reader

PE and PPE refer to two large families of mycobacterial proteins named after their conserved N-terminal motifs “Pro-Glu” (PE) and “Pro-Pro-Glu” (PPE). These proteins are mostly found in pathogenic mycobacteria and are often associated with virulence, antigenic variation, and interactions with the host immune system. We have now clarified this in the revised manuscript:

- “The analysis revealed a diverse array of proteins with various functions. Among them, we identified several PE/PPE family proteins, **so named for their conserved N-terminal Pro-Glu (PE) or Pro-Pro-Glu (PPE) motifs**, including PPE20, PPE41, and Rv2083, as well as components of the ESX-1 secretion system, such as EsxB (CFP-10), Rv3877, Rv3878, and Rv3881c, which are known to play roles in *M. tuberculosis* virulence and immune modulation(D’Souza *et al*, 2023; Gröschel *et al*, 2016).”

d) line 440, change constantly to consistently

Done

e) line 492, replace "interacting with" to "in proximity to"

Done

Referee #3

We thank the referee for their careful evaluation of our work and for recognizing the relevance of our study. We appreciate their constructive feedback and address their concerns regarding the cell biological aspects point by point below.

A) Line 144 „To determine the subcellular localization of PacL2 and CtpG, we engineered PacL2-mTurquoise and CtpG-mVenus fusion proteins". I was unable to find a description of how the strains were constructed. Were fluorescent protein (FP) gene fusions integrated at the original gene locus, or expressed as a merodiploid? Which promoter drove the expression of fusion genes? If original gene locus, are there downstream genes/operon structures?

Fluorescent fusion proteins were expressed as single copies in an *M. smegmatis* strain lacking the *pacL/ctp* system or in wild-type *M. tuberculosis*, under the control of the native *M. tuberculosis* promoter to ensure expression levels as close as possible to physiological conditions. We agree with the Referee that details regarding the fluorescent strains were missing, and we have added the following sentences to the manuscript to improve clarity:

- “Fluorophores (mTurquoise, mVenus, or mEos3.2), linked via a five–amino acid flexible linker (LEGSG), were fused in frame to the C-terminus of PacL and Ctp proteins. The fluorescent fusions were encoded on an integrative vector carrying the attachment site (*attP*) from mycobacteriophage L5, which enables site-specific integration into the mycobacterial chromosome through the L5 integrase system. The plasmid integrates as a single copy either in an *M. smegmatis* strain lacking the *pacL/ctp* system or in wild-type *M. tuberculosis*. To preserve the native genetic context, the entire *M. tuberculosis* operons (*cmtR–pacL2–ctpG* or *pacL1–ctpC*), together with 500 bp of their upstream promoter regions, were cloned. This strategy ensures expression levels of the fluorescent fusion proteins that closely approximate physiological conditions.” (New section in Methods)
- “To determine the subcellular localization of PacL2 and CtpG, we engineered C-terminal translational fusions with mTurquoise (PacL2^{mT}) and mVenus (CtpG^{mV}). Fluorescent fusions were introduced within the native operon (*cmtR-pacL2^{mT}-ctpG*, *cmtR-pacL2-ctpG^{mV}*, or *cmtR-pacL2^{mT}-ctpG^{mV}*) and cloned into a single-copy integrative vector, thereby preserving the operon structure and ensuring expression of the tagged proteins under the control of the endogenous promoter.”
- “To further characterize the nanoscale organization and dynamics of PacL2 and CtpG, we generated PacL2-mEos and CtpG-mEos fusion proteins for super-resolution microscopy. These constructs were based on the same integrative vectors used for the mTurquoise/mVenus fusions, except that the fluorescent proteins were replaced with mEos.”

B) Fig. 2d and most of the following epifluorescence images show strong fluorescent foci/patches at the cell poles. This may be the true localization of the proteins, or the result of protein overexpression or other aspects like the formation of protein aggregates. Therefore, it is important to control that

a) proteins are expressed as full length proteins, and not as truncations, and

b) that they are expressed at physiological levels. Western blotting is necessary to rule out any artifacts causing mislocalization of protein fusions.

a) To address this concern, we performed anti-GFP western blot analyses using protein lysates from the *M. smegmatis* strain expressing the *cmtR-PacL2^{mT}-ctpG^{mV}* operon and confirmed that both proteins were expressed as full-length species (PacL2^{mT}: 36.5 kDa; CtpG^{mV}: 106.5 kDa) (new Fig. 2N).

b) Because we do not have antibodies against *M. tuberculosis* PacL or Ctp proteins, and since these native proteins are not tagged, we could not perform western blot analyses to compare their expression levels with those of the fluorescent fusions. However, as stated in response to comment A, the fluorescent proteins are expressed as single-copy genes under the control of the native *M. tuberculosis* promoter. Therefore, we do not expect their expression levels to differ substantially from those of the wild-type proteins.

Although the fluorescent fusion proteins are expressed as full-length species and at physiological levels, we hypothesize that the large, immobile clusters often observed at the cell poles may represent protein aggregates. This is supported by the observation that such clusters persist in the G17L+G20L and K9A mutants, which are impacted in cadmium tolerance. We now clarify this point in the manuscript as follows:

- “This immobile fraction may represent non-functional assemblies, as similar clusters persist even when essential residues for effluxosome-mediated metal tolerance (G17, G20, K9) are mutated.”

C) I am also missing experiments showing that the FP fusions functionally replace the wild type proteins. Strains must be assayed for sensitivity to metals used in this study to prove their functionality.

To address this comment, we conducted metal sensitivity assays in strains expressing the fluorescent proteins (New Supplementary Figures S1K and L) and added the following figures and sentence to the main text:

- “Using epifluorescence microscopy, we observed that PacL2 localized in distinct membrane clusters, which became more intense and abundant upon cadmium exposure (Fig. 2D and E). Under the same conditions, CtpG also formed clusters that co-localized with PacL2 (Fig. 2F). While C-terminal tagging of PacL2 markedly reduced its functionality, though a weak cadmium tolerance was still detectable (Fig. EV1K), the CtpG-mVenus fusion remained fully functional (Fig. EV1L), indicating that the observed clusters represent active efflux machineries.”

D) It is claimed in Fig. 2 that fusions localize to the cell membrane. This is not evident to my eye. Please provide membrane staining and higher magnification images that clearly show foci are in the membrane and not in the cytosol.

CtpG is a P-type ATPase, a class of proteins widely recognized as membrane-associated, and our previous study demonstrated that PacL1 is localized to the membrane (Boudehen *et al*, 2022). Since PacL2 co-localizes with both CtpG and PacL1, it is likely that PacL2 is similarly associated with the cell membrane. Moreover, our PALM imaging data further support the membrane localization of PacL2 (see Movies 4 and 5). Accordingly, we removed the term “membrane-associated” from the epifluorescence paragraph but retained it in the PALM section. Particularly, the following sentence was added:

- “Using 3D photoactivated localization microscopy (PALM)(Betzig *et al*, 2006), we constructed the tridimensional nanoscale organization of PacL2 and CtpG within individual *M. smegmatis* cells and found that both proteins form membrane-associated clusters (Fig. 3A, B, and Movies 2-5).”

E) Fig. 3 How were the cells imaged? To my eye, it looks as if the cells were imaged with the focal plane being at the cell surface rather than at the central plane, otherwise the foci would all be at the periphery of the cells. Not that molecules moving along the short axis of the cell are underestimated for their displacement because of the curvature of the cell. In any event, please state how imaging was performed in terms of the focal plane for Fig. 2 and 3.

Epifluorescence acquisitions were performed in phase-contrast mode, which relies on phase shifts of the light wave as it passes through different cellular structures. Because the center of

the cell has a higher refractive index than the surrounding regions, the contrast is strongest at this position and thus serves as the focal plane. As a result, the microscope captures the entire cell thickness, producing a **projection** that combines signals from the upper, lower, and central regions of the cell. Consequently, confirming the membrane localization of fluorescent clusters using our epifluorescence microscopy setup is challenging, even when membrane staining is performed. The following sentence was added to the Methods section:

- “The focal plane was determined in phase-contrast mode by adjusting the focus to the region of maximal contrast, which corresponds to the cell center due to its higher refractive index relative to surrounding regions.”

F) Line 592 "When necessary, mycobacterial cells were fixed with 4% paraformaldehyde (PFA) treatment (2 hours for *M. tuberculosis* and 20 minutes for *M. smegmatis*), followed by three PBS washing steps." Please state which experiments were done using fixed cells.

The following sentence was added in the Methods section:

- “Acquisitions were performed on either live (Fig. 2D, 2G, 2K, 4J, 6B, 6G, and 7C) or fixed (Fig. 2F, 4B, and 4E) cells. Fixation was carried out with 4% paraformaldehyde (PFA) for 2 hours in *M. tuberculosis* and 20 minutes in *M. smegmatis*, followed by three washes with PBS.”

G) Line 656 "We used a threshold of 18.." 18 what? Please state the units or parameters that the number refers to.

We thank the reviewer for pointing this out. 18 is a hard threshold (in grey level unit) for the wavelet-based single molecule localization method. This is specified in the methods.

H) Line 188 "To further characterize and map mobile and immobile molecules, we applied an experimental threshold of $0.03 \mu\text{m}^2\cdot\text{s}^{-1}$ (Fig. 3g). Our analysis revealed that 70% of the molecules were immobile, while 30% were classified as mobile." It is not appropriate to distinguish mobile from immobile molecules based on MSD analyses. First of all, it is unclear why the authors come up with the threshold, this seems arbitrary. At the least, they must provide a convincing reason for this number. Secondly, Fig. 3g does not show an MSD plot, but a log plot of numbers of diffusion constants, i.e. trajectories with a certain MSD. Please show a proper MSD plot to evaluate the quality of the data. If the authors want to investigate if different subpopulations exist, e.g. static and mobile molecules, they should use squared displacement analyses, or another appropriate evaluation. E.g. TARDIS from the Endesfelder lab <https://doi.org/10.1038/s41592-023-02149-7>

We thank the reviewer for this constructive comment and for pointing out the importance of properly justifying the diffusion threshold used to discriminate mobile and immobile molecules. The empirical threshold of $0.03 \mu\text{m}^2\cdot\text{s}^{-1}$ was not chosen arbitrarily, but rather based on well-established diffusion regimes reported in the literature for membrane proteins. Previous single-particle tracking studies have shown that diffusion coefficients above $\sim 0.02\text{--}0.05 \mu\text{m}^2\cdot\text{s}^{-1}$ typically correspond to freely diffusive molecules, whereas values below this range indicate confined or immobilized states (Kusumi *et al.*, 1993 (PMID: 8298032); Saxton

& Jacobson, 1997 (PMID: 9241424); Rossier *et al.*, 2012 (PMID: 23023225)). Our choice of $0.03 \mu\text{m}^2\cdot\text{s}^{-1}$ thus represents a conservative, literature-supported cutoff commonly used to distinguish mobile and immobile populations in bacterial or eukaryotic membranes. We have clarified this justification in the revised text.

To address the reviewer's second point, we acknowledge that Fig. 3G originally displayed the distribution of diffusion coefficients rather than the MSD plots themselves. We have now added representative MSD curves directly in Fig. 3g, illustrating the large variability in PacL displacements and the heterogeneity of diffusion behaviors captured in our dataset.

We agree that more advanced population analyses (such as the TARDIS framework, Schreiber *et al.*, Nat. Methods, 2023) could indeed provide a more quantitative classification of molecular states. However, performing such exhaustive kinetic modeling is beyond the scope of this work, which aims primarily to describe the overall complexity of PacL dynamics and to raise hypotheses regarding its nanoscale organization that will be systematically addressed in future studies.

I) Fig. 6D where is fluorescence gone in the truncation mutants? Please provide a Western showing expression levels of the truncation constructs.

The fluorescence signal is lost because the truncated PacL2 variant (PacL2 Δ 31–84) is unstable. To avoid confusion, we removed this mutant from the manuscript, as its instability renders the Cd sensitivity and split-GFP experiments uninterpretable. We also added a western blot showing that the PacL1 truncated variants are expressed at similar levels to the wild type and at the expected molecular sizes (Fig. 6J). This blot also confirmed the loss of CtpC stability when co-expressed with the truncated PacL1 mutants, consistent with the fluorescence quantification reported in our previous manuscript. Accordingly, we have substantially revised this entire section as follows:

- “The mechanism by which PacL proteins assemble into membrane clusters remains unclear. To assess whether the cytoplasmic domain of PacL proteins contributes to PacL–PacL interactions, we constructed *pacL2* deletion mutants lacking portions of this domain (PacL2 Δ 55–84 and PacL2 Δ 31–84) (Fig. 6A). The PacL2 Δ 31–84 mutant appeared unstable (data not shown), but the PacL2 Δ 55–84 variant was still able to form fluorescent foci in *M. smegmatis* (Fig. 6B), although a slight decrease in mean fluorescent signal per cell was observed (Fig. 6C). As expected, this deletion, which removed the AE repeats from PacL2 (Fig. 6A), decreased CtpG stability as determined by fluorescence microscopy (Fig. 6B, D), impaired the PacL2–CtpG interaction (Fig. 6E), and reduced cadmium tolerance (Fig. 6F).

- We also generated *pacL1* deletion mutants lacking segments of its cytoplasmic domain (PacL1^{Δ54-86} and PacL1^{Δ37-86}) (**Fig. 6A**). As reported in our previous study (Boudehen *et al.*, 2022), both deletions reduced CtpC stability, as observed by epifluorescence microscopy and western blot (**Fig. 6G-J**). However, despite a diminished ability to form large clusters, both PacL1 deletion variants remained stable and continued to produce small fluorescent foci in the mycobacterial membrane (**Fig. 6G**). Overall, these results indicate that the cytoplasmic domain of PacL proteins, including the AE repeats, is not the primary determinant of PacL cluster formation, although it plays a crucial role in stabilizing interactions with P-ATPase pumps and ensuring proper protein function.

Finally, I have a bit of a conceptual problem: why would a single-celled organism cluster efflux pumps? In order to efficiently move toxic compounds across the cell membrane, isn't it more sensible to distribute all exporters relatively evenly across the membrane surface in order to decrease the time transporters interact with their substrates? I find it counterintuitive to generate large clusters that compete for substrate binding at few sites in the cell. Even if PacL proteins bind to metals, would this really increase the local concentration of a metal? Hard to believe.

As reported in the manuscript, the G17G20 and K9 mutants are severely impaired in both their ability to confer cadmium tolerance and to form mobile clusters. Nevertheless, they still assemble into large immobile clusters and are capable of stabilizing CtpG. We interpret these results as strong evidence that the formation of mobile PacL2/CtpG clusters is essential for P-type ATPase activity. We fully agree with the Referee that this finding is unexpected yet highly intriguing, and we plan to further explore the molecular mechanisms that govern effluxosome functionality in future studies. We find it particularly compelling to hypothesize that localized metal enrichment, driven by metal binding to multiple PacL proteins within effluxosomes, could play a key role. For instance, heterogeneous zinc distribution has been reported in *E. coli* cells (DOI: 10.1038/s41598-018-31461-y). Whether PacL proteins mediate a similar heterogeneous metal distribution in mycobacteria remains a fascinating open question.

I am also puzzled by the statement in line 405: Here, our findings indicate that the clustering of P-ATPase pumps by PacL proteins not only shields them from degradation but is also essential for their functionality." I was unable to find any Western blot in this study, so how can the authors make claims about stability?

We added western blot showing the CtpG protein is destabilized in both the absence of PacL2 or when PacL2 carries the 3EA mutation impacting the PacL2/CtpG interaction. The following sentences were added to the main text:

- “Consistent with this observation, Western blot analyses detected CtpG protein in bacterial lysates co-expressing PacL2, but not in those expressing CtpG alone (**Fig. 2I**).”
- “In addition, although the E59A or E71A mutants did not prevent the formation of PacL2-mTurquoise clusters, they resulted in destabilization of CtpG-mVenus (**Fig. 2K-N**) and the triple substitution E55A, E59A, and E71A completely abolished the presence of CtpG in bacterial lysates, as determined by Western blot analysis.”.

Additional information

Accession number for the primary proximity data:

Project accession: PXD070766

Token: CywLzIZeCkFt

Alternatively, you can access the dataset by logging in to the PRIDE website using the following account details:

Username: reviewer_pxd070766@ebi.ac.uk

Password: 7TuLMIMIDb10

Dear Olivier,

Thank you for submitting a revised version of your manuscript. We have now received input from all original reviewers, who now find that their main concerns have been addressed satisfactorily and recommend acceptance of the manuscript. There now remain only a few editorial and formatting points that need to be addressed before I can extend official acceptance of the manuscript:

1. Please submit keywords for your manuscript.
2. Email to one of the co-authors, Bertille Voisin (Bertille.Voisin@ipbs.fr), bounced, please check for correctness.
3. CRediT has replaced the traditional author contributions section because it offers a systematic, machine-readable author contributions format that allows for more effective research assessment. Please remove the Authors Contributions from the manuscript and use the free text boxes beneath each contributing author's name in our online submission system to add specific details on the author's contribution. More information is available in our guide to authors.
4. Please move "References" section before "Figure legends".
5. There is a reference to "data not shown" on page 11, row 320. "According to our policy, which does not permit references to "data not shown", please include this information in the Appendix.
6. Please rename the movies into Movie EV1-EV8 and update the callouts accordingly. The legends should be removed from the manuscript text file and zipped with each movie file.
7. Please remove EV table legends from the manuscript text and add each legend to the corresponding table.
8. In the "Data availability" section, please add a resolvable link to the PXD070766 dataset. More information about the format of this section can be found here: <https://link.springer.com/partners/embo-press/editorial-policies#Data%20availability%20statement>.
9. During our standard source data check, I noticed an unusually high number of numerical repetitions in the source data for figures 5G and 5H. I have attached the corresponding files with the detected duplications labelled in colour. Please take a look and correct if needed. A brief explanation would be very helpful - I appreciate that these duplications can also occur due to specific measurement or calculation methods used.
10. Our data editors have flagged the following issues in figure legends that need correcting:
 - Please note that the legends for figures 2; 4; 6; EV 1 is not provided in the sequential manner. Please check and correct.
 - Please define the annotated p values ****/****/**/* and provide the exact p-values in the legend of figure 3c-e.
 - Please provide the exact p values in the legends of figures 1b-d; 2a, e, h, j, l, m, o; 3c-e; 4a, c, d, f-i, k-l; 6c-f, h, i; 7b, d, e, g-i; EV 1a, b, j-l; EV 5c, d.
 - Please indicate the statistical test used for data analysis in the legends of figures 1b-d; 2a, e, h, j, l, m, o; 4a, c, d, f-i, k-l; 6c-f, h, i; 7b, d, e, g; EV 1a, b, j-l; EV 5c, d.
 - Please provide information on the number and nature of replicates in the legends of figures 1b-d; 2a, e, h, j, l, m, o; 3c-e; 4a, c, d, f-i, k-l; 6c-f, h, i; 7b, d, e, g-i; EV 1a-l; EV 5c, d.
 - Please define the white arrows and box in the legend of figure 7c.
11. Papers published in The EMBO Journal are accompanied online by a 'Synopsis' to enhance discoverability of the manuscript. It consists of A) a short (1-2 sentences) summary of the findings and their significance, B) 3-4 bullet points highlighting key results and C) a synopsis image that is 550x300-600 pixels large (width x height, jpeg or png format). You can either show a model or key data in the synopsis image. Please note that the image size is rather small and that text needs to be readable at the final size. Please send us this information together with the revised manuscript.

With best wishes,

Ieva

We realize that it is difficult to revise to a specific deadline. In the interest of protecting the conceptual advance provided by the work, we recommend a revision within 3 months (11th Mar 2026). Please discuss the revision progress ahead of this time with the editor if you require more time to complete the revisions.

Referee #1:

The authors have addressed all my concerns. Their additional mass spectrometry results are very insightful and further strengthen the study. I wish to congratulate the authors for this nicely conducted work.

Referee #2:

This is a revised manuscript, and my comments on the significance of the work can be taken from the first round reviews. The authors have provided a clear and comprehensive response to the points I raised in the previous review, and in my view, the manuscript is acceptable for publication in its current form. It reads well, and the additional clarification, as well as an additional experiment on the metal binding properties of PacL1, are greatly appreciated. In addition, I wish to thank the authors for making the work more accessible to a general audience.

Referee #3:

The authors have addressed all my concerns in a satisfactory manner. I can see much clearer now that the work is a wide importance, and I am convinced that efflux pump clustering has a true importance for the physiology of the cells.

The authors addressed the remaining editorial issues.

Dear Olivier,

Thank you for addressing the final editorial requests for your manuscript. I am now pleased to inform you that your manuscript has been accepted for publication. Congratulations with a nice study!

Before we forward your manuscript to our publishers, we would like to propose some edits in the manuscript abstract and the synopsis - please see in the attached file. I have also written a short blurb that will accompany the title of your manuscript in our online system. Please take a look and let me know if any corrections are needed.

You may qualify for financial assistance for your publication charges - either via a Springer Nature fully open access agreement or an EMBO initiative. Check your eligibility: <https://link.springer.com/journal/44318/how-to-publish-with-us>

If you have any questions, please do not hesitate to contact the Editorial Office or me directly. Thank you for your contribution to The EMBO Journal!

With best wishes,

Ieva

Please note that it is The EMBO Journal policy for the transcript of the editorial process (containing referee reports and your response letters) to be published as an online supplement to each paper. If you should prefer removal of any referee-only figures included in the point-by-point response(s), e.g. because they may still be used for future publication or because they have been reproduced from published work by others, please do let us know immediately via response email.

More information is available here: <https://link.springer.com/partners/embo-press/editorial-policies#Peer%20review>